# Glutamine deprivation alters the origin and function of cancer cell exosomes

Shih-Jung Fan[1,†], Benjamin Kroeger[1,†] (iD), Pauline P Marie[1], Esther M Bridges[2], John D Mason[1], Kristie McCormick[1], Christos E Zois[2], Helen Sheldon[2], Nasullah Khalid Alham[3,4], Errin Johnson[5], Matthew Ellis[1], Maria Irina Stefana[1], Cláudia C Mendes[1], Stephen Mark Wainwright[1], Christopher Cunningham[6], Freddie C Hamdy[6], John F Morris[1], Adrian L Harris[2], Clive Wilson[1] (iD) & Deborah CI Goberdhan[1,*] (iD)

## Abstract

Exosomes are secreted extracellular vesicles carrying diverse molecular cargos, which can modulate recipient cell behaviour. They are thought to derive from intraluminal vesicles formed in late endosomal multivesicular bodies (MVBs). An alternate exosome formation mechanism, which is conserved from fly to human, is described here, with exosomes carrying unique cargos, including the GTPase Rab11, generated in Rab11-positive recycling endosomal MVBs. Release of Rab11-positive exosomes from cancer cells is increased relative to late endosomal exosomes by reducing growth regulatory Akt/mechanistic Target of Rapamycin Complex 1 (mTORC1) signalling or depleting the key metabolic substrate glutamine, which diverts membrane flux through recycling endosomes. Vesicles produced under these conditions promote tumour cell proliferation and turnover and modulate blood vessel networks in xenograft mouse models *in vivo*. Their growth-promoting activity, which is also observed *in vitro*, is Rab11a-dependent, involves ERK-MAPK-signalling and is inhibited by antibodies against amphiregulin, an EGFR ligand concentrated on these vesicles. Therefore, glutamine depletion or mTORC1 inhibition stimulates release from Rab11a compartments of exosomes with pro-tumorigenic functions, which we propose promote stress-induced tumour adaptation.

**Keywords** exosome; extracellular vesicle; mechanistic Target of Rapamycin; multivesicular body; Rab11(a)
**Subject Categories** Cancer; Membranes & Trafficking; Metabolism
**The EMBO Journal (2020) 39: e103009**

See also: **G van Niel & C Théry** (August 2020)

## Introduction

Extracellular vesicles (EVs), produced in intracellular compartments or by plasma membrane shedding events, have emerged as critical players in cell–cell communication (Tkach & Théry, 2016; Maas *et al*, 2017). They deliver specific combinations of proteins, nucleic acids and lipids to recipient cells. EVs function in normal physiological processes, such as reproduction (Corrigan *et al*, 2014), immune responses (Bruno *et al*, 2015), neural development and maintenance (Krämer-Albers & Hill, 2016), and metabolism (Thomou *et al*, 2017). They also have roles in pathological events (Huang-Doran *et al*, 2017; Maas *et al*, 2017; Veerman *et al*, 2019) with much focus on the role of EVs in promoting tumour growth, survival and metastasis (Becker *et al*, 2016). These effects involve interactions of tumour cells with each other and surrounding stromal cells (Wendler *et al*, 2017). EVs can alter the tumour microenvironment, for example, by promoting endothelial network formation (Sheldon *et al*, 2010). In turn, microenvironmental stresses, such as hypoxia (Kucharzewska *et al*, 2013), can affect this signalling by driving changes in the tumour EV profile. The functional relevance of stress-induced EV signalling has, however, not been extensively characterised. This is of particular interest in cancer, where depletion of oxygen and key metabolites, such as glutamine (Zhang *et al*, 2017), is an inevitable outcome of rapid tumour growth.

Preparations of secreted EVs from cell lines and primary cell cultures include microvesicles derived from the plasma membrane and exosomes made inside cells (Théry *et al*, 2018). Exosomes are EVs of about 30–150 nm in diameter, formed by the inward budding of the limiting membrane of intracellular compartments, widely thought to be late endosomes. Exosome secretion results from fusion of the resulting multivesicular bodies (MVBs) with the plasma membrane (Maas *et al*, 2017). The transmembrane tetraspanins CD63 and CD81 are often used to identify exosomes (Kowal

1 Department of Physiology, Anatomy and Genetics, University of Oxford, Oxford, UK
2 Department of Oncology, Weatherall Institute of Molecular Medicine, University of Oxford, Oxford, UK
3 Institute of Biomedical Engineering, Department of Engineering Science, University of Oxford, Oxford, UK
4 Nuffield Department of Surgical Sciences, Oxford NIHR Biomedical Research Centre (BRC), John Radcliffe Hospital, University of Oxford, Oxford, UK
5 Sir William Dunn School of Pathology, University of Oxford, Oxford, UK
6 Nuffield Department of Surgical Sciences, John Radcliffe Hospital, University of Oxford, Oxford, UK
*Corresponding author (lead contact). Tel: +44 1865 282836; E-mail: deborah.goberdhan@dpag.ox.ac.uk
†These authors contributed equally to this work

*et al*, 2016; Mateescu *et al*, 2017). Members of the Endosomal Sorting Complexes Required for Transport (ESCRT) family (Colombo *et al*, 2013) and ceramides (Trajkovic *et al*, 2008) regulate two proposed exosome biogenesis pathways. Several endosomal Rab GTPases, which promote trafficking between specific intracellular compartments, also play important roles (Ostrowski *et al*, 2010). Whether these mechanisms contribute to the heterogeneity observed in exosome preparations (Zhang *et al*, 2018) remains largely undetermined.

We have developed a *Drosophila* model to investigate exosome biogenesis *in vivo* (Corrigan *et al*, 2014; reviewed in Wilson *et al*, 2017). This uses the prostate-like secondary cells (SCs) of the fly male accessory glands (AGs), which secrete exosomes in to the AG lumen and have unusually large intracellular membrane-bound compartments (Fig 1A), including Rab11-positive endosomes and Rab7-positive lysosomes.

Here, we present evidence that specific exosome subtypes are made in SC and human cancer cell recycling endosomal MVBs labelled by Rab11 family members. Furthermore, preferential release of Rab11a-marked exosomes from these compartments is triggered by depleting cancer cells of exogenous glutamine. This Rab11a-secretory switch is reproduced by reducing growth factor-regulated Akt and amino acid-sensitive mechanistic (formerly mammalian) Target of Rapamycin Complex 1 (mTORC1; Dibble & Cantley, 2015) signalling. We show that these exosomes have distinct cargos and unique *in vitro* activities. Since they also alter tumour cell growth and vessel formation in a human tumour xenograft mouse model, we propose that they contribute to adaptive responses to metabolic stress.

## Results

### Rab11-labelled multivesicular bodies make exosomes via an ESCRT-dependent mechanism in *Drosophila* secondary cells

To study exosome biogenesis in *Drosophila* SCs, we overexpressed the human exosome marker CD63-GFP. It labels the limiting membranes of large Rab7-positive acidic late endosomes and lysosomes (LELs) in SCs and of approximately 10 non-acidic compartments, containing a central, protein-rich, dense-core granule (DCG; Fig 1A–C), which are Rab11-positive (Redhai *et al*, 2016; Prince *et al*, 2019). Using super-resolution 3D-SIM, a few fluorescent intraluminal vesicles (ILVs) were observed inside LELs, although most GFP fluorescence is quenched by the acidic microenvironment (Redhai *et al*, 2016). Clusters of CD63-GFP-positive ILVs were also observed within the Rab11-positive compartments (Fig 1C and Movie EV1), the majority of which were of typical exosome size (Fig EV1A). ILVs were also seen in EM micrographs of the non-acidic and acidic compartments in non-transgenic flies (Fig EV1E).

To analyse ILV formation in these Rab11 compartments further, we studied SCs from flies expressing YFP-Rab11 from the endogenous *Rab11* locus, a so-called "gene trap" (Dunst *et al*, 2015). YFP-Rab11's fluorescence intensity was much lower than CD63-GFP and so not amenable to super-resolution imaging. Therefore, we imaged this marker using wide-field deconvolution fluorescence microscopy, which resolves membrane-bound ILVs as puncta (Fig 1D). The compartmental organisation of SCs expressing *YFP-Rab11* is not

significantly perturbed, in contrast to SCs expressing CD63-GFP, which appear to increase trafficking to acidic compartments (Corrigan *et al*, 2014; Redhai *et al*, 2016). For example, the numbers of large non-acidic compartments are unaffected compared to wild-type flies, whereas CD63-GFP expression roughly doubles their number and produces enlarged acidic compartments (Fig EV1B–D; Redhai *et al*, 2016). Fluorescent puncta were seen inside YFP-Rab11-positive SC compartments (Fig 1D), but fewer than with CD63-GFP (Fig 1C). On rare occasions (< 0.5% of cells), YFP-Rab11-labelled structures were sufficiently large that they could be resolved as ILVs (Fig EV1F). Unlike CD63-GFP (Fig EV1G), the Rab11 fusion protein did not traffic to the plasma membrane (Fig EV1H; Movie EV2). Sporadic YFP-Rab11-positive puncta were observed in the AG lumen, both in the gene trap line (Fig EV1H) and when YFP-Rab11 was specifically overexpressed in SCs (Fig EV1I). In contrast, a YFP-Rab7 gene trap fusion protein (Dunst *et al*, 2015) primarily trafficked to acidic LELs, which were abnormally enlarged (Fig 1E), and marked very few puncta in the AG lumen. To further investigate whether Rab11 is inside ILVs in Rab11 compartments, we co-expressed the YFP-Rab11 gene trap with a CD63-mCherry construct. CD63-mCherry trafficked to the limiting membrane of a subset of large non-acidic, DCG-containing compartments, which often excluded YFP-Rab11 at their surface (Fig EV1J). Inside these compartments, however, the two markers partially co-localised. This suggests that Rab11 is intravesicular, but it is not always associated with tetraspanin-labelled ILVs, perhaps because these molecules must co-localise at the endosomal limiting membrane to permit this. Taken together, we conclude that Rab11-labelled exosomes are formed at low levels in Rab11 SC compartments and secreted.

In searching for other markers of these alternative exosomes, we found using 3D-SIM that an overexpressed GFP-tagged form of Breathless (Btl; a fly homologue of the human transmembrane FGF receptor), which is normally expressed in SCs (Fig EV1K), trafficked on to ILVs in DCG-containing, Rab11 compartments (Fig 1F). Its expression did not affect the large non-acidic compartments in SCs and had only a minor effect on acidic compartment number (Fig EV1B–D). In some SCs, Btl-GFP, like YFP-Rab11, was found in the lumen of an LEL, but, unlike CD63-GFP, these markers were rarely, if ever, observed on the LEL limiting membrane (Appendix Fig S1), presumably because they reach this destination by sporadic fusion of LELs with Rab11 compartments (Corrigan *et al*, 2014) and not by endocytic trafficking. As with YFP-Rab11, transmembrane Btl-GFP protein was secreted in puncta into the AG lumen (Fig 2A). We, therefore, conclude that Rab11 and Btl are selective membrane-associated markers for at least some of the exosomes generated in Rab11 compartments of SCs, which we collectively term "Rab11-exosomes".

To test whether biogenesis of these exosomes is ESCRT-dependent, the temperature-inducible GAL4/GAL80[ts]/UAS system was used to knock down three *ESCRT*s implicated in mammalian and *Drosophila* exosome secretion (McCullough *et al*, 2013; Matusek *et al*, 2014), namely *Stam* (an ESCRT-0 subunit), *Vps28* (an ESCRT-I subunit) and *shrub* (*shrb*; the fly orthologue of mammalian *Chmp4a*-c, encoding an ESCRT-III subunit) in adult SCs. All treatments affected the number and size of large non-acidic compartments in SCs expressing Btl-GFP (Fig 2), YFP-Rab11 (Fig EV2A–F) and CD63-GFP (Appendix Fig S2). In Btl-GFP- and

CD63-GFP-expressing SCs, the number of non-acidic compartments increased in all three knockdowns (Fig 2A–E; Appendix Fig S2A–E), but analysis of *Rab11* gene trap males indicated that although drastically reduced in size, many of these non-acidic compartments retained Rab11 identity (Fig EV2A–E). For all three *ESCRT* knockdowns, the proportion of these compartments containing

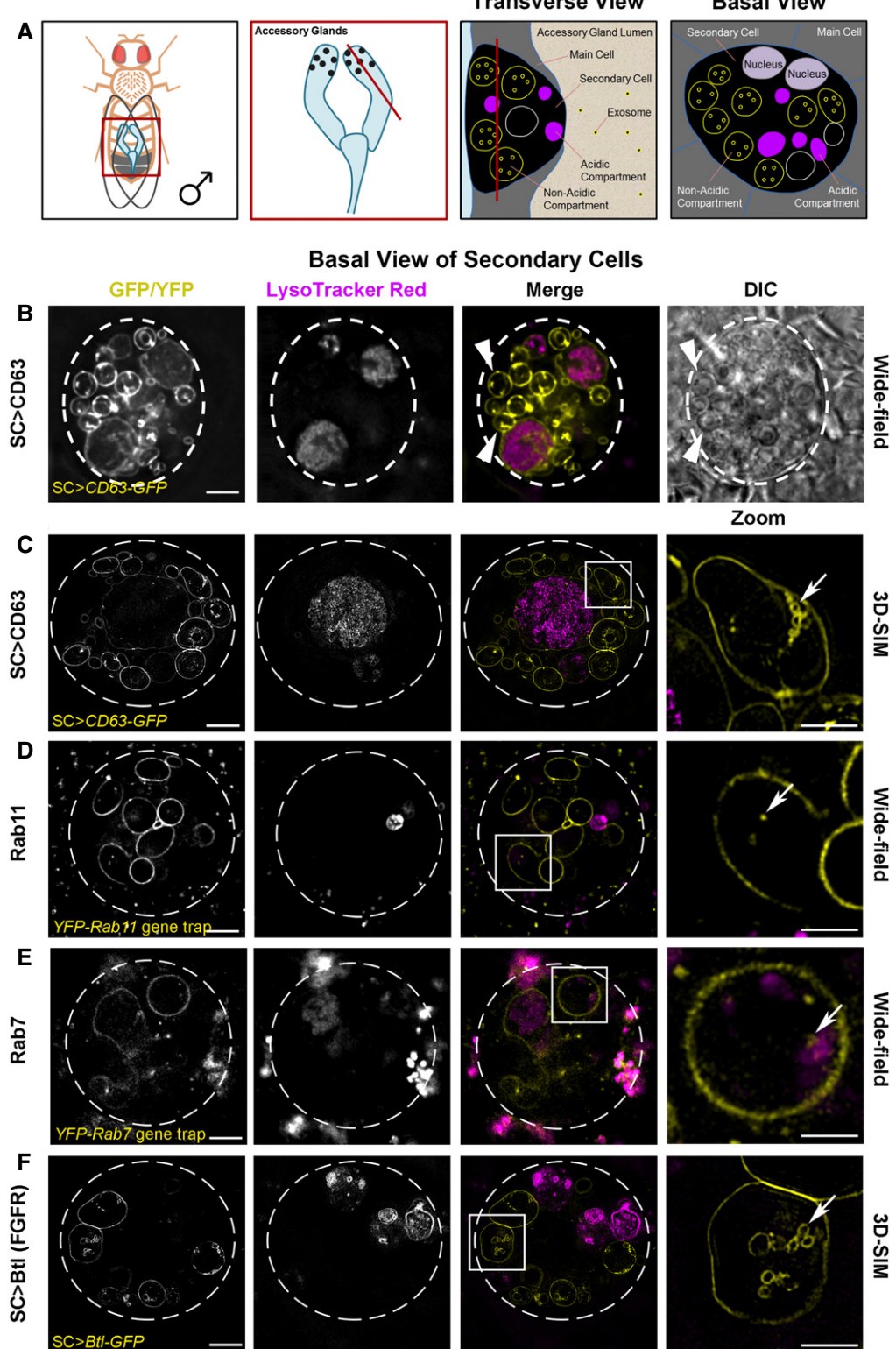

Figure 1.

◄

**Figure 1.   Rab11 compartments of *Drosophila* secondary cells contain intraluminal vesicles with specific cargos.**

A   Schematics illustrate the fly accessory glands and associated secondary cells, highlighting their acidic and non-acidic exosome-forming compartments. First panel shows male fruit fly and its accessory glands. Boxed region is enlarged in second panel, revealing secondary cells (SCs; black dots), with red line indicating the plane of section through the AG lumen, which is used to generate the transverse view through a SC and surrounding main cells within the epithelial layer in the third panel. The unusually large, acidic late endosomes and lysosomes (Rab7-positive; magenta) and non-acidic compartments, characteristic of SCs, which we demonstrate here contain intraluminal vesicles (ILVs, some of which are Rab11-positive, yellow), are labelled. The red line in the third panel shows the basal SC plane of section in the fourth panel and images in this figure.

B–F   Basal views through living SCs, with dashed white circles approximating the outline of a single SC, and acidic compartments marked by the vital dye LysoTracker® Red (magenta). In merge images, a single non-acidic (C, D and F) and acidic (E) compartment containing intraluminal vesicles (ILVs) is boxed and magnified in the right panel (Zoom). ILVs appear as membrane-delineated vesicles, using super-resolution 3D-structured illumination (3D-SIM) microscopy for the brighter overexpressed GFP-tagged constructs (yellow; C and F). However, ILVs appear only as puncta, using lower resolution wide-field microscopy for the fainter endogenously expressed YFP-tagged Rab GTPases (yellow; D and E). (B) Wide-field fluorescence image, including differential interference contrast (DIC), of SC expressing a GFP-tagged version of human CD63 (CD63-GFP). CD63-GFP expression is apparent on the limiting membranes of non-acidic compartments and their ILVs and also on the limiting membranes of the enlarged acidic compartments. Most large non-acidic compartments are Rab11-positive (D) and contain dense-core granules, which have a "fried egg" appearance with DIC (arrowheads) (Corrigan *et al*, 2014; Redhai *et al*, 2016). (C) 3D-SIM image of CD63-GFP-expressing SC. Arrow highlights CD63-GFP-marked ILVs (Zoom). Many more ILVs are apparent in non-acidic compartments in a complete Z-stack of a non-acidic compartment (Movie EV1). (D) Wide-field fluorescence image of an SC expressing a YFP-Rab11 gene trap. YFP-Rab11 marks the limiting membranes of most non-acidic compartments and internal puncta (arrow in Zoom), but not the surface of acidic compartments (Appendix Fig S1B). (E) Wide-field fluorescence image of SC expressing a YFP-Rab7 gene trap. YFP-Rab7 marks the limiting membranes of acidic compartments and internal puncta (arrow in Zoom). Enlarged acidic compartments are also present in adjacent main cells. (F) 3D-SIM image of SC expressing a GFP-tagged version of Breathless (Btl-GFP). Btl-GFP marks the limiting membranes of non-acidic compartments and their ILVs (arrow in Zoom), but not the surface of acidic compartments (Appendix Fig S1C). Images from 6-day-old male flies shifted to 29°C at eclosion. This induces GAL4/UAS-dependent SC transgene expression in (B, C and F). The genotypes of flies carrying multiple transgenes are as follows: *w; P[w+, UAS-CD63-GFP] P[w+, tub-GAL80ts]/+; dsx-GAL4/+* (B and C); *w; P[w+, tub-GAL80ts]/+; dsx-GAL4/P[w+, UAS-btl-GFP]* (F).

Data information: Scale bar in B–F (5 μm) and in C–F, Zoom (2 μm).

fluorescent puncta was significantly decreased (Figs 2F and EV2F; Appendix Fig S2F) and the number of secreted fluorescent puncta in the AG lumen was also greatly reduced (Fig 2G; Appendix Fig S2G). To test whether ESCRT proteins associate with Rab11 compartments, a Shrb-GFP fusion protein (Sweeney *et al*, 2006) was transiently expressed in SCs. It accumulated in subdomains and puncta at the surface of large non-acidic and acidic compartments (Fig EV2G; Movie EV3), consistent with our finding that *shrb* plays a role in generating ILVs in both Rab11 and LEL compartments. We conclude that the ESCRTs, including the ESCRT-0 Stam, which unlike other classes of ESCRT, is not thought to be involved in microvesicle biogenesis (McCullough *et al*, 2013), are required for ILV formation in Rab11 compartments *in vivo*. Furthermore, these ILVs are secreted as exosomes loaded with specific cargos.

### Rab11a-labelled multivesicular bodies also generate intraluminal vesicles in HCT116 colorectal cancer cells

To investigate whether human Rab11-labelled compartments also generate exosomes, we initially analysed HCT116 colorectal cancer (CRC) cells, which have clusters of perinuclear endosomal and Golgi compartments (Fan *et al*, 2016). In human cells, Rab11a, one of the two Rab11 isoforms, which primarily associates with recycling endosomes, has been reported to be associated with EVs (Keerthikumar *et al*, 2016) and to regulate exosome secretion (Savina *et al*, 2002). Using pan-Rab11, Rab11a- and Rab11b-specific antibodies, we found that Rab11a is the predominant Rab11 isoform in HCT116 cells (Appendix Fig S3A). In contrast to CD63, which co-localises with the LEL marker, LAMP1, Rab11a compartments are distinct from LELs, marked with LAMP2 (Fig 3A and B), and have very limited overlap with CD63 (Fig 3C). This differs from overexpression of fluorescent CD63 in SCs, where some CD63 enters Rab11 compartments (Fig 1B).

By overexpressing GFP-Rab11a in HCT116 cells, which induces clustering of Rab11a-positive compartments, we were able to use super-resolution 3D-SIM microscopy to detect internalised GFP in these compartments (Fig 3E), suggesting that Rab11a is incorporated into ILVs. To enlarge endosomal compartments, a GFP-tagged, constitutively active form of early endosomal Rab5 was expressed, which inhibits maturation of recycling and late endosomes, leading to the formation of enlarged immature endosomal compartments (Fig 3F). The endosomal Rab GTPases, Rab11a and Rab7, were both observed in subdomains at the surface of these Rab5-positive compartments and also on spatially distinct internal puncta (Fig 3F), demonstrating that the endosomal system can generate ILVs carrying either of these Rab signatures. We conclude that HCT116 cells contain both late and recycling endosomal MVBs and, as apparent in *Drosophila* SCs, the ILVs produced within them carry different cargos, including specific Rab GTPases.

### Glutamine depletion of HCT116 Cells induces a switch to secretion of Rab11a-exosomes with distinct cargos

To explore whether HCT116 CRC cells secrete Rab11a-positive exosomes, we collected small EVs for 24 h from cells cultured in serum-free conditions, but supplemented with insulin. This maintained growth factor signalling, as assessed by phosphorylation of mTORC1 downstream readouts, 4E-Binding Protein 1 (4E-BP1) and S6 (Appendix Fig S4A; 2.00 mM glutamine in Fig 4A). EVs isolated by ultracentrifugation were enriched for proteins that preferentially associate with exosomes (Figs 4B and EV3D), namely, the cytosolic adaptor protein Syntenin-1 (Syn-1), the ESCRT-I component Tsg101 and the tetraspanins CD81 and CD63 (Kowal *et al*, 2016). Low levels of Rab11a were also present, but not ER, Golgi or early endosome markers.

Glutamine is a major metabolic substrate in HCT116 cells (Jiang *et al*, 2013). We have previously shown that nutrient stress induced

by glutamine depletion inhibits a rapamycin-resistant form of nutrient-sensitive mTORC1, without complete suppression of S6 phosphorylation (Fan *et al*, 2016). We investigated whether cells

respond to such treatment by altering their EV secretion, as they do in hypoxic stress (Kucharzewska *et al*, 2013). As expected, glutamine depletion reduced cellular levels of hyper-phosphorylated 4E-

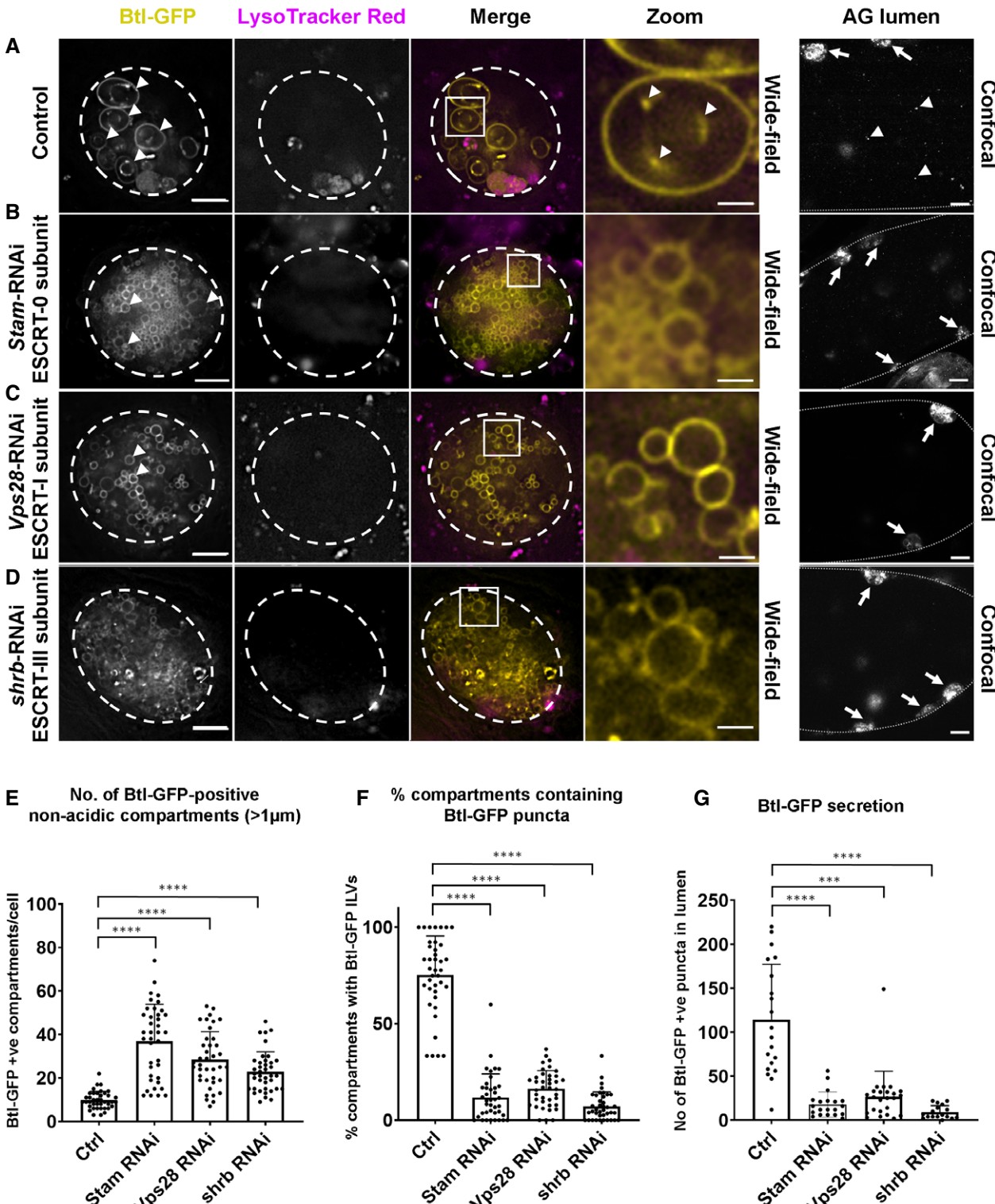

**Figure 2.**

**Figure 2. Exosome biogenesis in Rab11 compartments of *Drosophila* secondary cells is ESCRT-dependent.**

A–D  Wide-field fluorescence images of basal views through living SCs expressing a GFP-tagged form of Btl-GFP (yellow) in non-acidic compartments, with SC outline approximated by dashed white circles. Acidic compartments are marked by LysoTracker Red® (magenta). Boxed non-acidic compartments in merge images are magnified in A–D, Zoom. On the right, lower magnification confocal transverse images of fixed accessory gland (AG) lumens are shown with dotted lines indicating the basal side of the AG epithelial layer, which contains several fluorescent SCs (highlighted by arrows). (A) SC with no RNAi construct expressed (control) and AG lumen from same genotype. Btl-GFP-positive compartments containing fluorescent ILVs (in Btl-GFP panel), ILV membranes inside compartments (in Zoom panel) and secreted fluorescent puncta (AG lumen panel) are marked by arrowheads. (B) SC also expressing RNAi construct targeting ESCRT-0 subunit, *Stam*, and AG lumen from same genotype. Btl-GFP-positive ILVs (Zoom) and secreted puncta (AG lumen) are strongly reduced. (C) SC also expressing RNAi construct targeting ESCRT-I subunit, *Vps28*, and AG lumen from same genotype. Btl-GFP-positive ILVs (Zoom) and secreted puncta (AG lumen) are strongly reduced. (D) SC also expressing RNAi construct targeting ESCRT-III subunit, *shrb*, and AG lumen from same genotype. Btl-GFP-positive ILVs (Zoom) and secreted puncta are strongly reduced.

E  Bar chart showing the number of large (diameter greater than one micrometre) non-acidic Btl-GFP-positive compartments per SC. Data from 39 SCs (three per gland) are shown.

F  Bar chart showing percentage of these large Btl-GFP compartments containing Btl-GFP-positive ILVs in control and *ESCRT* knockdown SCs. Data from 39 SCs (three per gland) are shown.

G  Bar chart showing the total number of Btl-GFP fluorescent puncta in 10 Z-planes from the AG lumen following *ESCRT* knockdown in SCs, compared to controls without knockdown. Data from at least 17 AG lumens per condition are shown.

Data information: All data are from 6-day-old male flies shifted to 29°C at eclosion to induce expression of transgenes. Genotypes are as follows: *w; P[w⁺, tub-GAL80ᵗˢ]/+; dsx-GAL4/P[w⁺, UAS-btl-GFP]* with no knockdown construct (A), UAS-*Stam*-RNAi (HMS01429; B), UAS-*Vps28*-RNAi (v31894; C) or UAS-*shrb*-RNAi (v106823; D). Scale bars in A–D (5 μm); in A–D Zoom (1 μm); in A–D, AG lumen (20 μm). Data were analysed by one-way ANOVA. ****$P < 0.0001$, ***$P < 0.001$ relative to control. Bars and error bars in E–G denote mean ± SD.

Source data are available online for this figure.

BP1 and phosphorylated S6 (Figs 4A, EV3A and EV4E) over the 24-h EV collection period. Cellular expression of all exosome proteins was unaffected by glutamine depletion, except for CD63, which was reduced (Fig 4A). Blocking lysosomal function by buffering lysosomal protons with chloroquine over 24 h strongly increased CD63 levels under both conditions and suppressed the difference between conditions (Appendix Fig S4B), suggesting that slightly more CD63 is degraded when glutamine levels are depleted. There were comparably low levels of cell death in secreting cells under both glutamine-depleted ($10.2 ± 1.5\%$) and glutamine-replete ($11.5 ± 2.5\%; n = 3$) conditions, as measured by Trypan blue staining.

Nanoparticle Tracking Analysis (NTA) of EV preparations (Dragovic *et al*, 2011) revealed that EV numbers (when normalised to cell lysate protein levels) were not significantly changed compared to control values after glutamine depletion, and EV size distribution was unaltered (Figs 4B, EV3B and EV4E), a finding supported by transmission electron microscopic analysis (Fig EV3C).

However, by analysing EVs generated by the same protein mass of cells under these two conditions (a proxy for cell number; Baietti *et al*, 2012), we observed that glutamine depletion of HCT116 cells altered the secretion of specific exosome markers (Fig 4B). While exosome-associated Syn-1, Tsg101 and CD81 were only slightly decreased after glutamine depletion, when normalised to cell lysate protein levels, secretion of late endosomal CD63, which is reported to mark only a subset of exosomes in small EV preparations (Kowal *et al*, 2016), was more strongly reduced (Figs 4B and 5G). By contrast, Rab11a was increased by several fold (Figs 4B and 5G). Hypoxic stress increases levels of cytosolic lipid raft-associated caveolin-1 (Cav-1; Kucharzewska *et al*, 2013) in EVs. This protein was detected in EV preparations from HCT116 cells and its secretion was also enhanced by glutamine depletion (Figs 4B and 5G).

No marker is known to distribute evenly across all exosome or EV subtypes, so it was not possible to use a specific protein on the Western to reliably normalise signals on EV Western blots. We therefore analysed our data using a number of different variables for normalisation to confirm our findings (Fig EV3E). EV preparations contain mixtures of exosomes, microvesicles and protein aggregates

(Jeppesen *et al*, 2019); we found particle number measured by NTA to vary considerably relative to secreting cell mass. However, by including more biological replicates, we confirmed that levels of CD63 per EV fell more than other exosome markers following glutamine depletion, which may be partly explained by the reduction in cellular CD63 (Fig 4A), and Rab11a and Cav-1 levels were strongly increased (Fig EV3E). The same pattern was observed when CD81, Tsg101 or Syn-1 was employed to normalise Western blots of EVs (Fig EV3E).

To standardise our analysis in different cells and conditions in this study, we loaded EV preparations from cells of an equivalent protein mass under different conditions and directly compared the Western blot signals. However, normalising signals to the levels of CD81, which is thought to mark a broad subset of tetraspanin-labelled exosomes (Kowal *et al*, 2016; Jeppesen *et al*, 2019), led to the same conclusions (Appendix Fig S5). Furthermore, increased Rab11a and Cav-1 and decreased CD63 were also observed when Western signals were normalised to the change in mean particle number across three or more biological repeats, as measured by NTA (Fig EV4E).

Importantly, a similar switch in Rab11a secretion was demonstrated in EVs isolated by size-exclusion chromatography (SEC; Fig EV3F and F′), an alternative method for EV isolation based on particle size, which we found more consistent in comparative EV isolation experiments. We employed this method for most of our subsequent studies, but confirmed key findings, such as intravesicular localisation of Rab11a and functional activity using both approaches. Since we also observed that Rab11a and CD63 remain in separate compartments in HCT116 cells after glutamine depletion (Fig 3D), our data suggest that the endosomal origin of exosomes secreted from CRC cells is altered under these conditions, so that more exosomes are released from Rab11a compartments.

One alternative explanation is that Rab11a is secreted in microvesicles shed from the surface of HCT116 cells under stress. Although Rab11a-positive recycling endosomal compartments are observed below the plasma membrane, very little, if any, Rab11a is trafficked to the surface in either glutamine-replete or

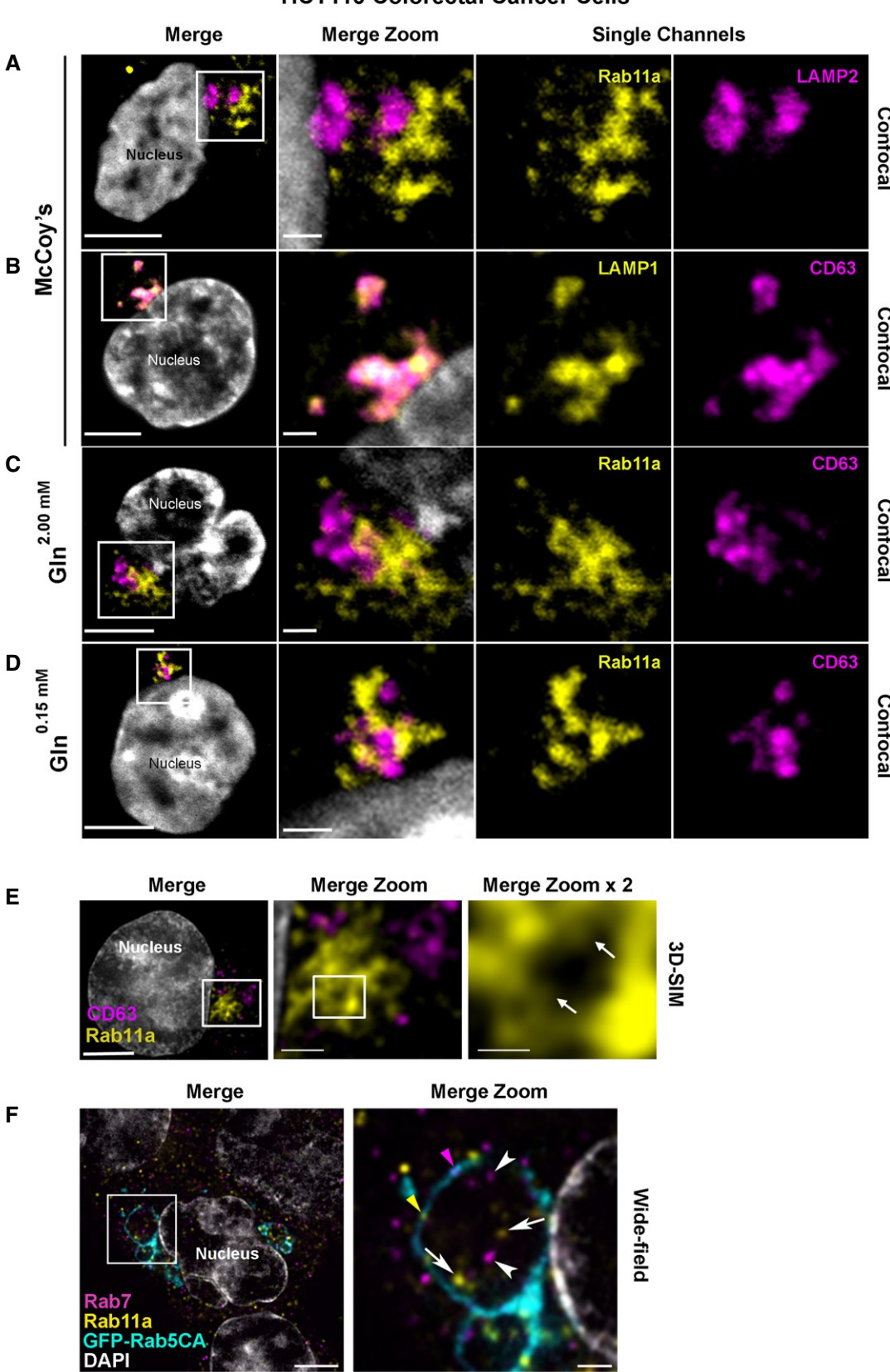

**Figure 3.**

**Figure 3. Rab11a labels a distinct subset of multivesicular bodies and their intraluminal vesicles in HCT116 colorectal cancer cells.**

A–D    Confocal images of fixed HCT116 cells, with boxed regions enlarged to the right. DAPI (grey) marks nucleus. Rab11a antibody is isoform-specific except in (F). (A) Rab11a (yellow) is located in compartments distinct from the late endosomal and lysosomal marker, LAMP2 (magenta). (B) CD63 (magenta) predominantly co-localises with the late endosomal and lysosomal marker, LAMP1 (yellow). (C) Rab11a (yellow) is located in compartments distinct from CD63 (magenta) under glutamine-replete conditions. (D) Rab11a (yellow) is located in compartments distinct from CD63 (magenta) under glutamine-depleted conditions.

E    Super-resolution 3D-SIM image of fixed HCT116 cell expressing GFP-Rab11a (yellow), and stained with CD63 (magenta). DAPI (grey) marks nucleus. Boxed Rab11a-positive compartments, which frequently cluster, are magnified in Merge Zoom. This panel is further magnified in Merge Zoom × 2, revealing GFP-Rab11a (arrows in right panel) inside compartments.

F    Wide-field fluorescence image of fixed HCT116 cells, stained with Rab11a (yellow) and Rab7 (magenta), expressing constitutively active GFP-tagged Rab5 (GFP-Rab5CA; cyan), which stalls endosomal maturation and produces enlarged Rab5-positive endosomes. One of these is boxed in the Merge and magnified in Merge Zoom, revealing internal puncta marked by Rab11a (arrows) and Rab7 (arrowheads) and limiting membrane subdomains of Rab11a (yellow arrowhead) and Rab7 (magenta arrowhead). DAPI (grey) marks nuclei.

Data information: Scale bars in A–F (5 μm), in Merge Zoom (1 μm) and in Merge Zoom × 2 (0.5 μm).

glutamine-depleted conditions (Appendix Fig S4C and D). This is in sharp contrast to CD81, which concentrates at the plasma membrane. To further demonstrate that Rab11a is associated with exosomes, we separated EV preparations using high-resolution iodixanol-PBS density gradients, on which exosomes, microvesicles and protein aggregates fractionate differently, though not entirely separately (Jeppesen *et al*, 2019). Although Rab11a levels were too low in EVs from glutamine-replete cells to draw firm conclusions (Appendix Fig S6A), Rab11a was distributed in the same way as exosomal CD63 and CD81 in EVs from glutamine-depleted cells, but did not fully overlap with the microvesicle marker, Annexin A1 (AnxA1; Fig 4C).

To demonstrate that endosomal Rab11a is a *bona fide* internal exosome marker and not an associated contaminant in the isolation procedure, EVs induced by glutamine depletion were subjected to proteinase K digestion in the absence and presence of the detergent Triton X-100. While most proteins partially exposed on the exosome surface like CD81 were destroyed by both treatments, Rab11a behaved like other internal markers, such as Syn-1, and was only fully digested in the presence of detergent, either in EV preparations isolated by ultracentrifugation (Fig 4D) or by SEC (Appendix Fig S6C). CD63 was not digested even in the presence of Triton X-100, but it was degraded in the presence of a more potent detergent, lithium dodecyl sulphate (Appendix Fig S6B), suggesting that Triton either does not fully disrupt the CD63/lipid interactions or allow sufficient unfolding of exosomal CD63 to permit digestion.

To confirm that CD63 and Rab11a mark different exosome subtypes, CD63-positive vesicles were immunocaptured and analysed by Western blot. Both in glutamine-replete and in glutamine-depleted conditions, while CD81 and other standard exosome markers were pulled down in roughly equivalent proportions, little, if any, specific Rab11a binding was observed (Fig 4E, Appendix Fig S6D). In fact, under glutamine-depleted conditions, only a small fraction of Rab11a was immunocaptured by CD81 pull-down (Fig 4F), consistent with the finding that these markers are largely separated inside cells (Appendix Fig S4C and D), and supporting our conclusion that Rab11a marks an alternative exosome subtype.

### Glutamine depletion and inhibition of Akt/mTORC1 signalling induces an "exosome switch" in cancer cell lines of different origins

We tested the effect of glutamine depletion on EV secretion by other cancer cell lines, selecting a glutamine concentration that

suppressed, but did not completely block S6 phosphorylation. Culturing HeLa cervical cancer cells in glutamine-depleted medium, which reduced phosphorylation of S6 and 4E-BP1, slightly increased EV number (Appendix Fig S7A, A', and A''', Fig EV4E) and increased Rab11a secretion relative to other exosome markers (Fig 5A and G, Appendix Figs S3B, S5B and C, and S7A''), while cellular levels of Rab11a were unaffected (Appendix Fig S7A). In the prostate cancer cell line, LNCaP, depletion of extracellular glutamine, which also reduced S6 and 4E-BP1 phosphorylation (Appendix Fig S7B, Fig EV4E), led to an increase in EV levels of Rab11a relative to CD81 (Appendix Figs S5D and H), while EV number and CD63 remained unchanged, but this increase was not significant relative to cell lysate protein mass (Figs 5B and G, and EV4E, Appendix Fig S7B'). Cav-1 was not detectable in this cell line. We conclude that HeLa and LNCaP cells, like HCT116 cells, respond to glutamine depletion by increasing secretion of exosome-associated Rab11a (though in LNCaP cells, only relative to other exosome markers), presumably by releasing more Rab11a-exosomes.

Since glutamine depletion reduces mTORC1 activity, we tested whether blocking mTORC1 with the mTOR-specific inhibitor Torin 1 (Thoreen *et al*, 2009) could also induce an exosome switch. In LNCaP cells, Torin 1 treatment slightly elevated the number of EVs produced, but reduced CD63 and strongly increased Rab11a in EVs relative to cell lysate protein and CD81 (Figs 5C and G, Appendix Figs S5E and H), without affecting cellular levels of these proteins (Appendix Fig S7C and C', Fig EV4E). However, in other cell lines like HCT116, the mTORC1 inhibitors, Torin 1 and rapamycin, had a much stronger inhibitory effect on mTORC1 activity and particularly S6 phosphorylation when compared to glutamine depletion (Fig EV4A, B and E), and this was associated with an approximately 50% reduction in EV particle numbers (Fig EV4A'', B'' and E). Secretion of all exosome markers, and particularly CD63, was significantly reduced (Fig EV4A' and B'), suggesting a more general suppression of exosome release. For rapamycin, we found that a very similar result was obtained using SEC for EV isolation, confirming that these findings are independent of EV isolation technique (Appendix Fig S7E).

A more modest mTORC1 inhibition in HCT116 cells was induced by knocking down *raptor*, which encodes a core mTORC1 component. EVs were collected for 24 h after 3 days of knockdown. Under these conditions, cellular levels of Syn-1, Tsg101 and CD63 were all decreased (Fig EV4C) and EV number was also reduced (Fig EV4C' and E), but residual phosphorylation of S6 was more variably maintained (Fig EV4C and E). This treatment induced a reduction in most standard secreted exosome

proteins, with a stronger decrease in CD63 (Fig 5D). However, Rab11a and Cav-1 secretion per unit cell protein mass was maintained, and indeed, it was increased relative to CD81 and EV number (Fig 5D, Appendix Figs S5F and H). These findings are again consistent with induction of an mTORC1-regulated switch in the balance of exosome secretion from late endosomal to Rab11a compartments.

Since mTORC1 activity is controlled by growth factor signalling, we also tested the effect of the Akt inhibitor AZD5363 on exosome secretion by HCT116 cells, which carry an activating mutation in

upstream PI3KCA (Ahmed *et al*, 2013). A switch in exosome secretion was again observed (Fig 5E and G, Appendix Fig S5G), most notably involving an increase in Rab11a and Cav-1 in EVs relative to cell lysate protein or CD81, in the absence of similar changes in cellular levels of these proteins or EV number (Fig EV4D, D′ and E). In addition to blocking phosphorylation of the Akt target PRAS40, this treatment partially inhibited phosphorylation of mTORC1 targets, S6 and 4E-BP1 (Fig EV4D and E), which might account for the switch. HCT116 cells also harbour a mutation in KRAS, which activates the ERK MAP kinase signalling cascade downstream of

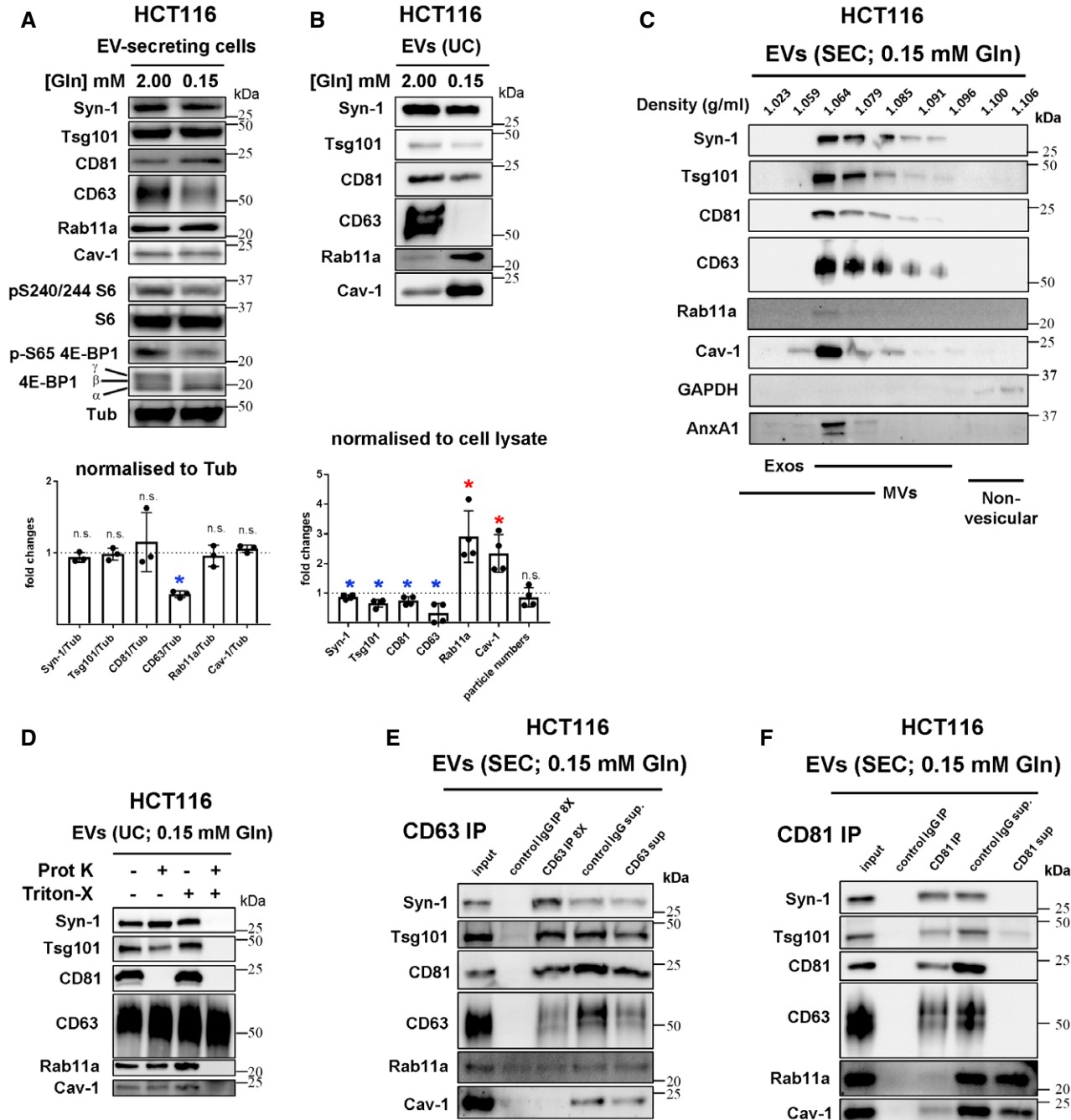

**Figure 4.**

**Figure 4.  Reduction in extracellular glutamine induces a switch to Rab11a-positive exosome secretion in HCT116 colorectal cancer cells.**

A   Western blot analysis of putative exosome markers in lysates from HCT116 cells cultured in glutamine-replete (2.00 mM) and glutamine-depleted (0.15 mM) medium for 24 h. Gel was loaded with equal amounts of protein (total cell lysate protein was reduced by 19 ± 4% after glutamine depletion). The activity of mTORC1 was assessed via phosphorylation of S6 and 4E-BP1, using phospho-specific antibodies and a pan-4E-BP1 antibody, where the hyper-phosphorylated form of 4E-BP1 produced by mTORC1 gives the slowest migrating γ band. Bar chart shows the abundance of putative exosome proteins relative to tubulin in these lysates.

B–F   Western blot analyses of EV preparations. (B) EVs isolated by ultracentrifugation (UC) of medium from HCT116 cells cultured in glutamine-replete and glutamine-depleted conditions for 24 h and loaded according to cell lysate protein levels to compare secretion on a per cell basis, as shown in bar chart from four biological replicates. (C) EV preparation isolated by size-exclusion chromatography (SEC) from glutamine-depleted HCT116 cells and then separated by high-resolution iodixanol-PBS density centrifugation. Note that exosomes (exos) and microvesicles (MVs) are found in only partially overlapping fractions and the pattern of Rab11a separation mirrors that of exosome markers (primarily 1.064–1.085 g/ml). (D) EVs isolated by UC under glutamine-depleted conditions as in (B) and then subjected to proteinase K (Prot K) digestion in the presence or absence of Triton® X-100 (Triton X). Only membrane-associated CD81 is digested in the absence of Triton® X-100, while CD63 is resistant to digestion, even in the presence of Triton® X-100. SEC-isolated EVs also behave similarly (Appendix Fig S6C). (E) EVs isolated under glutamine-depleted conditions were immunocaptured with anti-CD63 antibodies coupled to magnetic beads. This method only pulls down a fraction of CD63, so the protein from pull-down of eight times the input is loaded for comparison. Approximately the same low levels of Rab11a are captured by control IgG and anti-CD63 beads. (F) EVs isolated under glutamine-depleted conditions were immunocaptured with anti-CD81 antibodies coupled to magnetic beads. All or most CD63 and CD81 appears to be pulled down, but < 10% of Rab11a and Cav-1 is captured. NTA analysis suggests about 10% of all particles are pulled down with this approach.

Data information: Bar charts derived from at least three independent experiments and analysed by the Kruskal–Wallis test: *P < 0.05; n.s. = not significant. Bars and error bars denote mean ± SD. Significantly decreased levels are in blue and increased levels are in red.
Source data are available online for this figure.

growth factor receptors and has been implicated in regulating the proteome of CRC cell exosomes (Demory Beckler *et al*, 2013). Cells treated with a specific inhibitor that can block activated KRAS, BI-2852 (Kessler *et al*, 2019), did not show significant changes in mTORC1 activity (Appendix Fig S7D) and did not alter their EV secretion (Fig 5F, Appendix Fig S7D′), suggesting that the effect of KRAS on exosomes does not involve a switch to Rab11a-exosome secretion. Overall, our data from CRC, cervical and prostate cancer cell lines lead us to conclude that glutamine depletion or inhibition of Akt/mTORC1 signalling induces a switch in the balance of exosome secretion towards vesicles from recycling endosomal compartments, increasing exosome levels of Rab11a, as well as Cav-1, by several fold. Stronger mTORC1 blockade suppresses general exosome secretion, so although the balance of exosome secretion may still be shifted towards recycling endosomes, total Rab11a secretion is not increased.

**Glutamine depletion-induced extracellular vesicles have altered activities that are dependent on Rab11a**

Since EVs secreted by glutamine-depleted HCT116 cells have altered cargos, we tested whether this affected their bioactivity. When EVs from glutamine-replete and glutamine-depleted cells were prepared by ultracentrifugation, then mixed with naïve recipient HCT116 cells, 30 min prior to plating, the stress-induced EVs selectively stimulated cell growth under low serum conditions (Fig 6A). Cell proliferation was enhanced by this treatment (Fig EV5A), but rates of apoptosis were unaffected (Fig EV5B). The same activity was reproduced with EVs isolated by SEC and shown to be dose-dependent (Figs 6A and EV5C), so both of these methods were employed in subsequent functional assays.

To test for effects of glutamine depletion-induced EVs on stromal cells, EV preparations were also added to human umbilical vein endothelial cells (HUVECs), which were then plated in Matrigel® to assess their ability to form and maintain a tubular network. EVs isolated under glutamine-replete and glutamine-depleted conditions increased network formation compared to PBS-treated controls

(Fig 6B). Network maintenance was, however, enhanced by EVs from glutamine-depleted cells, suggesting that these EVs are better at supporting a stable capillary network. Therefore, following glutamine depletion, HCT116 cells secrete EVs enriched in exosomes from Rab11a-positive endosomes that display novel or enhanced activities.

To test whether trafficking through the Rab11a-dependent recycling endosomal pathway was essential for the increased growth induced by EVs from glutamine-depleted cells, *Rab11a* was knocked down in secreting cells. This suppressed Rab11a levels in cells (Fig EV5D) and EVs (Fig 6C), reduced the number of EVs secreted by approximately 40% (Fig EV5D′), but had no effect on mTORC1 activity or cellular levels of exosome proteins (Fig EV5D). Exosome markers Syn-1, Tsg101 and CD63 were not significantly altered in EVs compared to controls and Cav-1 secretion was increased (Fig 6C), suggesting that Rab11a compartments are not the only route for secretion of these molecules. Knockdown did, however, strongly suppress the enhanced growth-promoting activity of EVs produced by glutamine-depleted cells (Fig 6C′). We conclude that this activity is likely to involve Rab11a-dependent exosome secretion.

**Altered endosomal pathway trafficking induces an exosome switch**

We reasoned that changes in endosomal trafficking might be involved in switching the balance of exosome secretion to favour Rab11a-exosomes following glutamine depletion. To test this, the internalisation of molecules that specifically traffic from the plasma membrane either to LELs or to Rab11a compartments was assessed under both glutamine-replete and glutamine-depleted conditions. While CD63 accumulates in LEL compartments (Fig 3B), transferrin (Tfn) is returned to the plasma membrane after endocytosis, via the rapid recycling pathway and Rab11a-positive recycling endosomes (Mayle *et al*, 2012). Consistent with this, internalised Alexa-488-conjugated Tfn partly co-localised with Rab11a in HCT116 cells (Appendix Fig S8A), but not with the LEL marker LAMP2 (Appendix Fig S8B). When HCT116 cells were depleted of

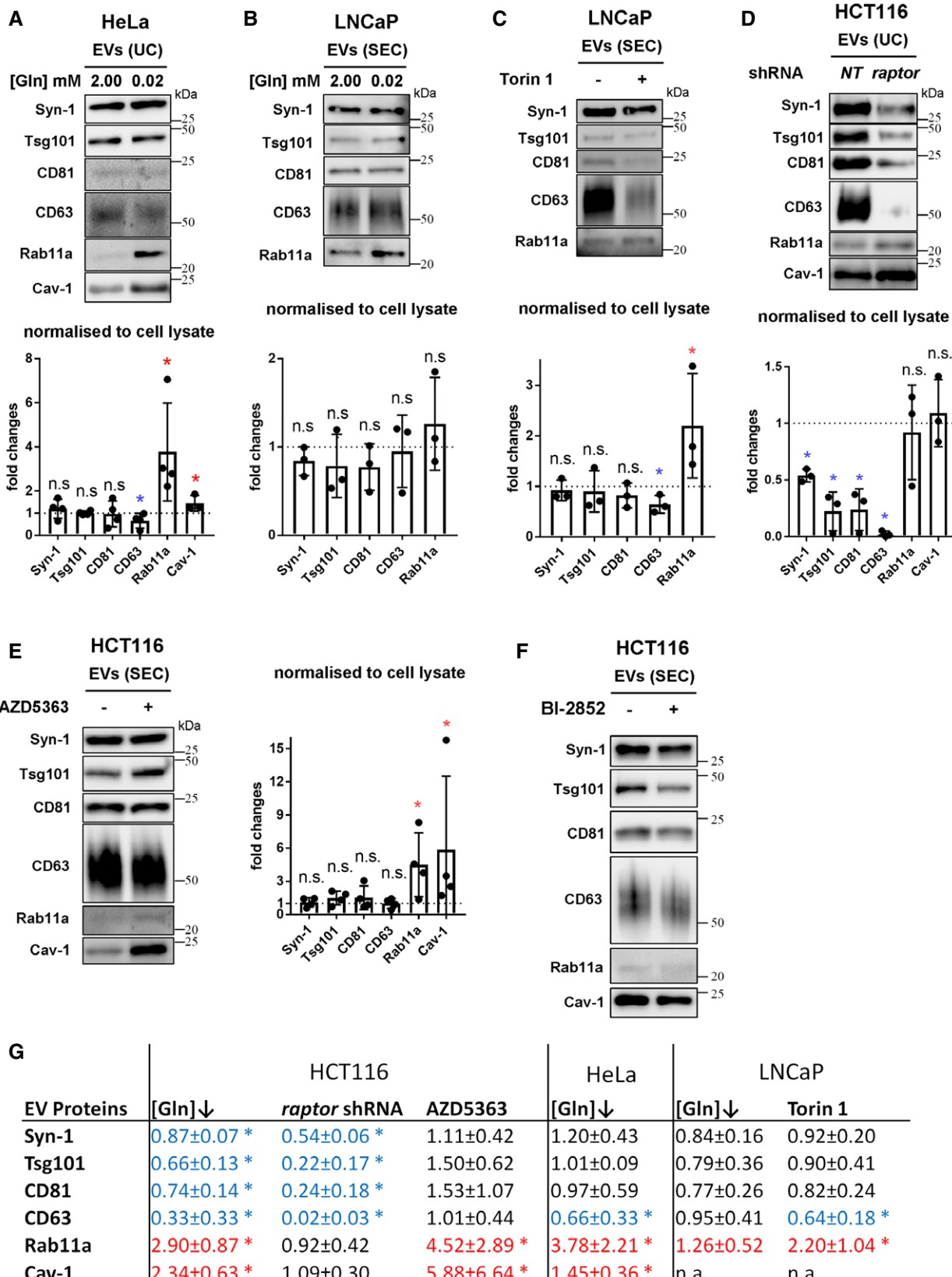

**Figure 5.**

| EV Proteins | HCT116 | | | HeLa | LNCaP | |
|---|---|---|---|---|---|---|
| | [Gln]↓ | *raptor* shRNA | AZD5363 | [Gln]↓ | [Gln]↓ | Torin 1 |
| Syn-1 | 0.87±0.07 * | 0.54±0.06 * | 1.11±0.42 | 1.20±0.43 | 0.84±0.16 | 0.92±0.20 |
| Tsg101 | 0.66±0.13 * | 0.22±0.17 * | 1.50±0.62 | 1.01±0.09 | 0.79±0.36 | 0.90±0.41 |
| CD81 | 0.74±0.14 * | 0.24±0.18 * | 1.53±1.07 | 0.97±0.59 | 0.77±0.26 | 0.82±0.24 |
| CD63 | 0.33±0.33 * | 0.02±0.03 * | 1.01±0.44 | 0.66±0.33 * | 0.95±0.41 | 0.64±0.18 * |
| Rab11a | 2.90±0.87 * | 0.92±0.42 | 4.52±2.89 * | 3.78±2.21 * | 1.26±0.52 | 2.20±1.04 * |
| Cav-1 | 2.34±0.63 * | 1.09±0.30 | 5.88±6.64 * | 1.45±0.36 * | n.a. | n.a. |

◄ **Figure 5. Reduction in extracellular glutamine or Akt/mTORC1 signalling induces a switch to Rab11a-positive exosome secretion in three different cancer cell lines.**

Western blot analyses of EV preparations. Gel loading is normalised to cell lysate protein levels. Bar charts present changes in levels of putative exosome proteins relative to cell lysate protein.

A  EVs isolated by UC of medium from HeLa cells cultured in glutamine-replete (2.00 mM) and -depleted (0.02 mM) conditions for 24 h (see Appendix Fig S7A″ for analysis of SEC-isolated EVs).

B  EVs isolated by size-exclusion chromatography (SEC; fractions two to seven) from LNCaP cells cultured in glutamine-replete (2.00 mM) and -depleted (0.02 mM) conditions for 24 h.

C  EVs isolated by SEC from LNCaP cells cultured in the presence or absence of 120 nM Torin 1 for 24 h.

D  EVs isolated by UC from HCT116 cells cultured for 24 h following 3 days of *raptor* or non-targeting (*NT*) shRNA knockdown.

E  EVs isolated by SEC from HCT116 cells cultured in the presence or absence of 3 μM Akt inhibitor AZD5363 for 24 h.

F  EVs isolated by SEC from HCT116 cells cultured under glutamine-replete conditions in the presence or absence of 10 μM KRAS inhibitor BI-2852 (or control compound BI2853) for 24 h.

G  Table summarising the EV protein analyses in Figs 4 and 5 (normalised to cell lysate protein levels). Data analysed by the Kruskal–Wallis test: *$P < 0.05$. Significantly decreased levels are in blue and increased levels are in red.

Data information: Bar charts derived from at least three independent experiments and analysed by the Kruskal–Wallis test: *$P < 0.05$; n.s. = not significant. Bars and error bars denote mean ± SD. Significantly decreased levels are in blue and increased levels are in red.

Source data are available online for this figure.

glutamine, uptake of an anti-CD63 antibody, assessed by confocal imaging and Western analysis, was significantly reduced (Fig 6D and E), consistent with the reduced levels and increased degradation of CD63 in these cells (Appendix Fig S4B). By contrast, uptake of fluorescent (Fig 6D) and biotin-conjugated (Fig 6E) Tfn were significantly elevated, demonstrating that flux through the recycling endosomal pathway is increased in response to glutamine depletion.

Since altered endosomal trafficking is associated with a glutamine-regulated switch in exosome secretion, we hypothesised that a similar switch might be induced in HCT116 cells by inhibiting traffic through the late endosomal pathway. Knockdown of *Rab7* substantially reduced EV secretion, decreased CD63 in EV preparations under glutamine-replete conditions, but increased Rab11a levels, without having a major effect on marker expression levels in cells (Figs 6F and EV5E and E′; cellular levels of Rab11a were in fact slightly reduced). The resulting EVs stimulated growth of serum-depleted HCT116 cells (Fig 6F′), demonstrating that altering the balance of late and recycling endosomal trafficking can induce changes analogous to the switch in exosome secretion caused by reducing glutamine.

### EVs secreted from glutamine-depleted cells induce ERK-dependent growth in HCT116 recipient cells and are blocked by an anti-amphiregulin antibody

Since EVs from glutamine-depleted HCT116 cells induce growth in recipient cells, we investigated whether growth factor signalling might be involved. Phosphorylation of the MAPK ERK was enhanced when serum-starved HCT116 cells were incubated for 30 min with EV preparations from glutamine-depleted cells (Fig 7A). Indeed, treating recipient HCT116 cells with the ERK inhibitor SCH772984 for the first 24 h of culture after EV addition completely blocked the extra growth induced by these vesicles (Fig 7B, Appendix Fig S8C), demonstrating that elevated ERK signalling is critically involved in growth stimulation.

Since epidermal growth factor receptor (EGFR) signalling plays a key role in CRC growth and the EGF ligand amphiregulin (AREG) is expressed by HCT116 cells (Nagathihall *et al*, 2014) and known to be packaged into CRC-derived exosomes (Higginbotham *et al*, 2011), we tested whether AREG was present on EVs from glutamine-depleted HCT116 cells. Glutamine depletion increased levels of a ~ 26 kDa membrane-associated form of AREG (Brown *et al*, 1998) in cell lysates and in EVs produced by SEC and UC under these conditions (Figs 7C and EV5F, Appendix Fig S8D). Treating EV preparations from glutamine-depleted and control cells with a neutralising antibody to AREG (Raimondo *et al*, 2019) partially suppressed ERK activation in target cells (Fig 7C″) and blocked the growth-promoting effect of glutamine depletion-induced EVs (Fig 7C′), suggesting that AREG-positive vesicles play a critical role in this process. Consistent with previous studies (Zhang *et al*, 2019), ELISA measurements of AREG in EV preparations from glutamine-depleted cells revealed that the minimum concentration of AREG required to promote proliferation is extremely low (1.4 pg AREG/ml; Fig EV5C). Glutamine depletion therefore induces the secretion of Rab11a-exosomes in HCT116 cells, and low levels of AREG on the EVs produced by these cells play an important role in the enhanced growth-promoting activity of these vesicles.

### Glutamine depletion-induced extracellular vesicles promote tumour cell turnover *in vivo*

To test whether the enhanced activities of EV preparations from glutamine-depleted HCT116 cells could be replicated *in vivo*, we established HCT116 xenografts in exponential growth phase and then directly injected them every 3 days over 9 days with EVs, isolated from HCT116 cells under glutamine-replete and glutamine-depleted conditions. Xenografts were analysed 24 h following the last injection of EVs. There was no change in overall tumour growth over this short time period following exposure to either type of EV preparation (Appendix Fig S9). However, there were significant histological differences within the xenografts for different treatments. As previously reported (Sheldon *et al*, 2010), injecting EVs into xenografts led to increased overall vessel number (Fig 8A); however, the effect of the two EV preparations was different, with vessel lumen size significantly increased in xenografts exposed to EVs from glutamine-depleted cells. The increased number of blood vessels for both EV treatments correlated with elevated tumour cell proliferation (Fig 8B). Again, consistent with observations *in vitro*, exposure to EV preparations from glutamine-depleted cells led to the highest levels of proliferating cells in the viable regions of the

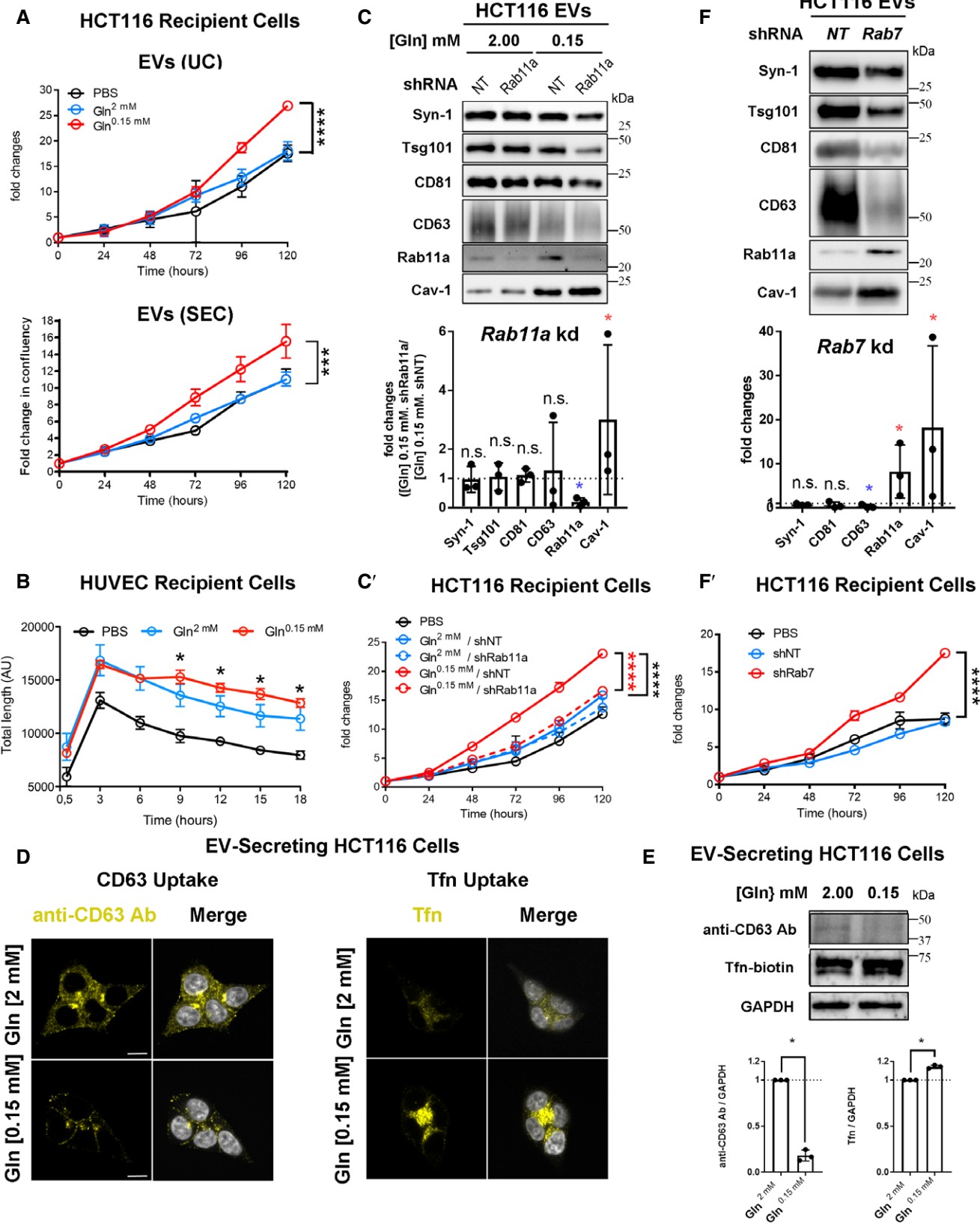

**Figure 6.**

◄

**Figure 6. Rab11a-dependent exosome secretion is induced by altered endosomal trafficking and promotes recipient tumour cell growth.**

A   Growth curves for HCT116 recipient cells in reduced (1%) serum conditions following 30-min pre-incubation with EVs isolated by UC ($10^4$ per cell; top) or by SEC ($4 \times 10^3$ per cell; bottom) from glutamine-replete and glutamine-depleted HCT116 cells or with vehicle (PBS). Fold change in confluency is a measure of cell area occupying well relative to zero time, as measured by IncuCyte software.

B   Cumulative tube length for HUVEC recipient cells following treatment with $10^4$ EVs isolated by UC from glutamine-replete and glutamine-depleted HCT116 cells or with vehicle (PBS). Both EV preparations promote tubulation, but the network is more stable with EVs from glutamine-depleted HCT116 cells.

C   Western blot analysis of EVs isolated by UC from HCT116 cells cultured in glutamine-replete and glutamine-depleted medium for 24 h, following transduction with a *Rab11a* or control non-targeting (NT) shRNA knockdown construct over previous 2 days. Bar chart shows change in putative exosome proteins in EVs secreted from *Rab11a* knockdown cells relative to NT-treated cells in glutamine-depleted conditions, following initial normalisation to cell lysate protein. (C′) Growth curves are for HCT116 recipient cells pre-treated with EVs isolated as in (C) or with vehicle (PBS). ****P colour denotes significant increase relative to EVs from glutamine-replete cells (black) and *Rab11a* knockdown cells (red).

D   Left-hand group of four images shows cells grown in glutamine-replete (2.00 mM) and glutamine-depleted (0.15 mM) conditions for 24 h, then incubated with an anti-CD63 antibody (yellow) for 30 min at 4°C, washed with PBS, chased at 37°C for 30 min, then fixed, immunostained and imaged. Right-hand group of four images shows cells grown in glutamine-replete and glutamine-depleted conditions for 24 h, incubated with Tfn-Alexa Fluor® 488 (yellow) for 30 min at 4°C, shifted to 37°C for 30 min, then washed, immediately fixed and imaged.

E   Western blot showing levels of anti-CD63 heavy-chain and biotin-conjugated Tfn in HCT116 cells cultured for 24 h in glutamine-replete or glutamine-depleted conditions, incubated for 30 min in medium containing these molecules at 4°C, then chased at 37°C for 15 min (the chase step was not performed for Tfn in D, to reduce loss due to rapid recycling of Tfn).

F   Western blot analysis of EVs isolated by UC from HCT116 cells in glutamine-replete medium, transduced with a *Rab7* or non-targeting (NT) control shRNA knockdown construct. Bar charts show changes in putative exosome proteins in EVs isolated by ultracentrifugation, following normalisation to cell lysate protein. (F′) Growth curves are for HCT116 recipient cells pre-treated with EVs isolated as in (F) or with vehicle (PBS).

Data information: Scale bars in D (5 μm) apply to all panels. Growth curves were reproduced in three independent experiments and analysed by two-way ANOVA. Bar charts derived from three independent experiments and analysed by the Kruskal–Wallis test: ****$P < 0.0001$, ***$P < 0.001$, *$P < 0.05$. Bars and error bars denote mean ± SD. Significantly decreased levels are in blue and increased levels are in red in panels C and F.

Source data are available online for this figure.

tumour (Fig 8B). However, an overall increase in necrosis (Fig 8C) and in hypoxia, as indicated by expression of carbonic anhydrase (CA9) as a downstream hypoxia marker (Fig 8D; McIntyre *et al*, 2016), was also induced by EV treatment and particularly by EV preparations from glutamine-depleted cells. Overall, we conclude that changes in EV production driven by glutamine depletion induce increased cell turnover *in vivo*, which may contribute to adaptive changes under these metabolic stress conditions.

## Discussion

Exosomes are important mediators of signalling between cells, particularly in cancer, but the mechanisms by which exosome signals might change in response to microenvironmental stress have remained largely unexplored. Here, using human CRC, cervical and prostate cancer cell lines, and supported by a fly *in vivo* exosome biogenesis model, we provide evidence that exosomes are not only formed in late endosomal MVBs, but also in Rab11/11a-positive recycling endosomal MVBs. These latter exosomes carry distinct cargos, including Rab11/11a, providing a diagnostic signature for their compartment of origin. Furthermore, they appear to be preferentially released by cancer cells following glutamine depletion or Akt/mTORC1 inhibition, generating unique biological effects *in vitro* and *in vivo* (shown schematically in Fig 8E). We propose that this previously undescribed exosome subtype plays a unique role in tumour adaptation to metabolic stresses.

### Rab11/11a-positive compartments are novel sites of exosome biogenesis

Knockdown studies in human cells and *Drosophila* have highlighted a role for Rab11 family members in exosome secretion (Savina *et al*, 2002; Koles *et al*, 2012; Beckett *et al*, 2013; Messenger *et al*, 2018).

In fact, one report previously identified Rab11a-positive MVBs in human cells (Savina *et al*, 2005). These experiments were, however, taken as evidence for Rab11a facilitating MVB trafficking to the cell surface and/or docking MVBs there. Our *in vivo* analysis in *Drosophila*, together with imaging studies and EV analysis in human cell lines, indicates that Rab11 compartments are conserved sites of exosome biogenesis. Rab11 family members are classically associated with recycling endosomes, but are also implicated in regulating secretory traffic from the Golgi (Welz *et al*, 2014). Human Rab11a-positive exosomes, however, appear to be generated in recycling endosomes, because their secretion is upregulated when traffic through the recycling pathway is increased, and Rab11a-positive ILVs can form in enlarged Rab5CA-induced endosomes. High-resolution imaging in the fly system has allowed us to demonstrate that production of these vesicles is ESCRT-dependent, suggesting parallels with late endosomal exosomes in their biogenesis mechanisms. Regulators include the ESCRT-0 Stam, which, unlike other classes of ESCRT, is not thought to be involved in microvesicle biogenesis (McCullough *et al*, 2013), providing further support that vesicles secreted into the AG lumen are an alternative form of exosomes.

Glutamine depletion induces a switch in the balance of exosome marker secretion in cancer cells from CD63 to Rab11a, a result observed with two very different EV isolation methods. Using density gradient separation, we have shown that most of Rab11a in EV preparations from glutamine-depleted cells co-fractionates with exosome markers. However, immunocapture experiments reveal that CD63- and Rab11a-containing exosomes are essentially distinct. A small proportion of Rab11a is carried on exosomes marked by CD81, which is one of the best-established markers for "classical" exosomes. Currently, we cannot determine whether significant numbers of Rab11a-negative, CD81-positive exosomes are made in Rab11a compartments, or whether they are primarily formed in other endosomal compartments (Fig 8E). Interestingly, in *Drosophila* SCs, human CD63 can be incorporated into ILVs in Rab11

**A   HCT116 recipient cells**

**B   HCT116 Recipient Cells**

**C   HCT116 Cells**

normalised to cell lysate

**C′   HCT116 Recipient Cells**

**C″   HCT116 Recipient Cells**

Figure 7.

**Figure 7. EVs induced by glutamine depletion promote amphiregulin- and ERK-dependent growth in HCT116 recipient cells.**

A   Western blot analysis of ERK phosphorylation (p-ERK) in recipient serum-deprived HCT116 cells pre-treated with EVs isolated by SEC from glutamine-replete or glutamine-depleted HCT116 cells or vehicle (PBS). Bar chart shows ratio of p-ERK to ERK levels from triplicate independent experiments.

B   Bar chart shows HCT116 recipient cell growth over 120 h, after pre-treatment with EV preparations isolated as in (A) or with PBS, and then incubation for the first 24 h of culture in the presence or absence of the ERK inhibitor SCH772984 (1.00 μM). Growth curves are shown in Appendix Fig S8C.

C   Western blot analysis showing levels of the EGFR ligand, amphiregulin (AREG) in EVs isolated by SEC of medium from HCT116 cells cultured in glutamine-replete (2.00 mM) or glutamine-depleted (0.15 mM) conditions for 24 h. Gel loading was normalised to cell lysate protein levels. AREG's molecular weight (approximately 26–28 kDa) suggests it is in its membrane-associated form (see Appendix Fig S8D). In the bar chart, the levels of putative exosome proteins were normalised to cell lysate protein levels. Significantly decreased levels are in blue and increased levels are in red. (C') Growth curves in low (1%) serum are for HCT116 recipient cells pre-treated with the EV preparations [isolated as in (C)], which had themselves been pre-treated with and without anti-AREG neutralising antibody or with a control IgG. Solid red line shows growth-promoting effect of EVs isolated under glutamine depletion, which is blocked by anti-AREG antibody (red dashed line). ***P colour denotes significant increase relative to EVs from glutamine-replete cells (black) and after anti-AREG treatment of EVs (red). (C'') Western blot analysis of ERK phosphorylation (p-ERK) in recipient serum-deprived HCT116 cells pre-treated with EVs isolated by SEC from glutamine-replete (2.00 mM) or glutamine-depleted (0.15 mM) HCT116 cells or vehicle (PBS), which were pre-treated with anti-AREG (AREGab) or a control immunoglobulin (Ig). Note increase in ERK phosphorylation using EVs from glutamine-depleted cells, which is blunted by the addition of the anti-AREG antibody.

Data information; Growth curves were reproduced in three independent experiments and analysed by two-way ANOVA. Bar charts derived from three independent experiments and analysed by ANOVA or the Kruskal–Wallis test: ***$P < 0.001$, **$P < 0.01$, *$P < 0.05$. Bars and error bars denote mean ± SD.
Source data are available online for this figure.

compartments, and some of these exosomes appear to contain Rab11 (Fig EV1J). However, even in this system, the vast majority of CD63 traffics to LEL compartments, suggesting that co-localisation is likely to be the result of high-level expression in these cells leading to co-trafficking of these markers to recycling endosomal compartments.

Since we do not currently have a method to isolate Rab11a-labelled exosomes, we cannot exclude the possibility that increased Rab11a secretion under stress conditions is explained by loading more Rab11a into a similar number of exosomes. However, this seems unlikely, because we have shown that these stress conditions increase secretory traffic through the recycling endosomal pathway, and that diverting traffic through this pathway by *Rab7* knockdown is sufficient to both increase Rab11a secretion and induce growth effects that mirror those seen after glutamine depletion.

Both in *Drosophila* (Corrigan *et al*, 2014) and in glutamine-depleted human cells, blocking Rab11/11a activity inhibits the biological activities of secreted EVs, suggesting a key role for exosomes generated in Rab11/11a compartments. This manipulation could have other indirect effects on secretion; for example, it has been implicated in CD63-marked exosome secretion (Messenger *et al*, 2018; van Niel *et al*, 2018). However, at least in HCT116 cells, *Rab11a* knockdown leads to only modest or no reduction in several other exosome markers in EV preparations and increases Cav-1 secretion. The treatment therefore appears to have relatively specific effects on Rab11a-exosome secretion in this cell type and suggests these exosomes have important disease-relevant activities.

The continued release of Cav-1, which partially co-fractionates with exosome markers, following *Rab11a* knockdown in HCT116 cells suggests that its secretion under stress conditions involves another EV biogenesis pathway, within unidentified intracellular compartments and/or from the cell surface. Secretion of Cav-1-containing small EVs of unknown subcellular origin has also recently been shown to be under metabolic (Crewe *et al*, 2018), as well as hypoxic (Kucharzewska *et al*, 2013) control. In the latter case, this may involve reduced constitutive endocytosis (Bourseau-Guilmain *et al*, 2016), but a role for different endosomal pathways has not been investigated. Distinguishing the different classes of tumour EV that are produced under metabolic stress should assist in determining what cancer-relevant functions are associated with each specific EV subtype, as we have done for Rab11a-exosomes.

Although not major constituents of exosomes, the identification of Rab11/11a and potentially Rab7 (Fig 3F) as signatures for exosome origin suggests a new approach for distinguishing different exosome subtypes in EV preparations. In flies, the selective labelling by Rab11 of smaller numbers of vesicles in Rab11 compartments than CD63-GFP or Btl-GFP suggests the existence of subpopulations of vesicles in a single compartment, which we may have partly distinguished by co-expressing YFP-Rab11 with CD63-mCherry (Fig EV1J). Unlike transmembrane proteins present on the limiting membrane of secretory compartments, Rabs are thought to disengage from the lipid bilayer before or during plasma membrane fusion, as evidenced by our fly (Fig EV1G and H) and human (Appendix Fig S4C and D) cell data, so they are unlikely to be incorporated into microvesicles shed from the cell surface. Multiple Rabs have been reported to be present in EV preparations (Keerthikumar *et al*, 2016) and several are only partially pulled down by anti-tetra-spanin EV immunocapture (Kowal *et al*, 2016), suggesting that other exosome subtypes are yet to be identified. Of particular interest are Rab35 and Rab27 family members, which, like Rab11, have been implicated in exosome release (Hsu *et al*, 2010; Ostrowski *et al*, 2010). Whether they mark additional exosome biogenesis pathways or primarily promote trafficking from other pathways remains to be determined.

### Tumour exosome signalling is regulated by metabolic stress and Akt/mTORC1 signalling

Our finding that several cancer cell types alter their exosome secretion and signalling following glutamine depletion highlights a novel function for the recycling endosomal exosome biogenesis pathway in a tumour's response to its microenvironment. In contrast to cancer cells, the highly secretory SCs of the fly AG release exosomes from Rab11-positive compartments containing DCGs in a regulated fashion under normal physiological conditions (Corrigan *et al*, 2014; Redhai *et al*, 2016). Consequently, we have not been able to assess whether there is a switch in exosome secretion by these cells under nutrient stress. Rab11 family members have also been implicated in granule secretion from pancreatic beta cells (Sugawara *et al*, 2009) and *Drosophila* salivary glands (Farkaš *et al*, 2015), although to date, ILVs have not

been reported in the much smaller compartments involved. It will be interesting to investigate whether highly secretory cells use an entirely different mechanism to control Rab11-exosome secretion or whether there is an unappreciated link to cellular stress, perhaps involving the ER stress that is likely to be induced in these cells.

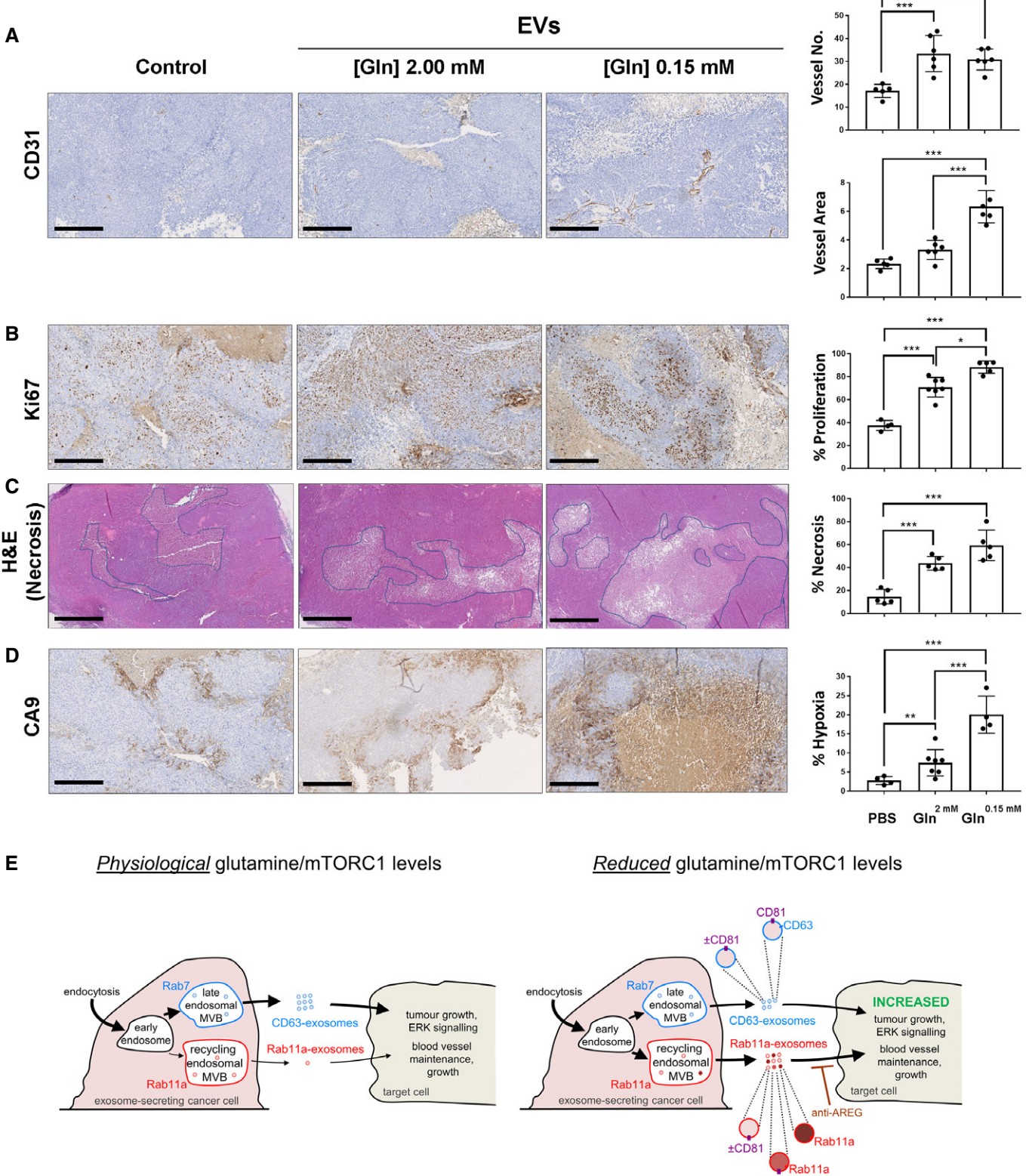

**Figure 8.**

**Figure 8. EVs from glutamine-depleted HCT116 cells increase tumour cell turnover and vessel growth in mouse HCT116 xenografts.**

HCT116 flank tumours produced by subcutaneous injection were established for 20 days before injection at 3-day intervals with vehicle (PBS), or EVs isolated by UC from glutamine-replete or glutamine-depleted HCT116 cells. Tumours were excised 24 h after last of four injections for analysis. Panels show representative immunostained histological sections of tumour tissue quantified using the Visiopharm Integrator System.

A  Sections immunostained for CD31, which labels endothelial cells and blood vessels, with blood vessel number (upper) and total area (lower) represented in bar charts.
B  Sections immunostained for Ki67, which stains proliferative cells. Proportion of tumour cells with Ki69 staining is represented in bar chart.
C  Sections stained with haematoxylin and eosin, which highlights necrotic regions (pale staining). Proportion of tumour area that is necrotic is represented in bar chart.
D  Sections immunostained for CA9, which is expressed in hypoxic regions. Proportion of tumour cells with CA9 staining is represented in bar chart.
E  Schematic model showing how in cancer cells, regulation of endosomal trafficking by depletion of exogenous glutamine or reduced Akt/mTORC1 signalling can induce a change in the balance of exosome production. Relative levels of a mixed population of exosomes from the established Rab7-late endosomal multivesicular bodies (MVBs), termed "CD63-exosomes", are reduced relative to the mixed exosome population from Rab11a-labelled recycling endosomal MVBs, termed "Rab11a-exosomes", a subset of which appear to be marked with Rab11a. Although CD63 does not appear to be trafficked through these latter compartments, the "classical" exosome marker CD81 appears to be present at undefined levels and marks some Rab11a-containing exosomes. The resulting vesicles can increase ERK signalling and cell growth in recipient tumour cells and enhance growth and stability of network formation in endothelial cells. The growth effects are Rab11a-dependent and can be blocked by anti-AREG antibodies. Thickness of arrows indicates relative levels of membrane flux through the two exosome-generating endosomal routes.

Data information: Scale bar is 250 μm, except for (C), which is 500 μm. Data were analysed by one-way ANOVA; $n \geq 4$; *$P < 0.05$, **$P < 0.01$, ***$P < 0.001$. Bars and error bars denote mean ± SD.
Source data are available online for this figure.

Growth factor- and nutrient-dependent regulation of many cellular functions, including proliferation, apoptosis, autophagy and metabolism, involves mTORC1 (Goberdhan *et al*, 2016). The roles of mTORC1 in secretion, particularly exosome release, have been less extensively studied. A recent report suggested that exosome secretion is increased after mTORC1 inhibition in mouse embryonic fibroblasts, but markers like Rab11a and Cav-1 were not analysed in this study (Zou *et al*, 2018). Our findings suggest that therapeutic blockade of PI3K/Akt or mTORC1, but not KRAS, signalling in cancer can induce a switch in secreted exosome subtype, although in some cells, complete mTORC1 inhibition appears to strongly reduce all exosome secretion. In addition to highlighting a new role for growth factor signalling and glutamine in exosome regulation, and its potential role in adaptation mechanisms, our data indicate the importance of monitoring mTORC1 activity during EV isolation procedures.

**Exosome signalling in tumour adaptation**

We were initially surprised to find that the release of recycling endosomal exosomes from glutamine-depleted HCT116 cells enhances tumour cell growth, in addition to promoting blood vessel maintenance *in vitro*. Our subsequent xenograft analysis has allowed us to detect similar activities *in vivo* and develop a model that may explain these findings.

Consistent with previous reports (Sheldon *et al*, 2010), we observed that injecting EVs from glutamine-replete or glutamine-depleted cells increased the number of blood vessels within tumours. However, EVs from glutamine-depleted cells also preferentially increased proliferation in the tumour, which likely contributed to the elevated hypoxia and necrosis within the xenografts, despite the increased number of blood vessels. Tumour cells under nutrient and hypoxic stress are known to adapt to such metabolic stress conditions to survive and grow. For example, cancer cells can become "glutamine addicted", whereby glutamine metabolism is able to sustain proliferation (Tardito *et al*, 2015). These cells often upregulate their growth regulatory pathways to achieve this. For instance, oncogenic KRAS drives glutamine metabolism in lung adenocarcinoma (Romero *et al*, 2017), while in lung squamous cell carcinoma, where mTORC1 is frequently hyper-activated, it has been observed that tumours that are highly resistant to conventional chemotherapy respond to inhibitors targeting the glutamine pathway (Lukey *et al*, 2018). Therefore, although we show an important role for amphiregulin in glutamine depletion-induced, EV-mediated growth, the HCT116 cell line already has activating PI3KCA and KRAS mutations, which may allow it to respond differently to the exosomes generated under such circumstances.

Release of Rab11a-exosomes under metabolic stress provides a mechanism for generating the heterogeneity in tumours that might drive adaptive changes. The secretion of growth-promoting molecules associated with Rab11a-exosomes from cells might promote their quiescence or even apoptosis. However, it could also stimulate a subpopulation of target cells that are more resistant to metabolic stress to activate ERK and other growth factor cascades, proliferate, alter their metabolism and potentially outcompete their neighbours.

Our *in vitro* studies suggest that low levels of membrane-associated AREG and the ERK signalling it induces may be involved in this process, although we cannot eliminate the possibility that binding of the anti-AREG antibody to AREG-containing exosomes also indirectly interferes with other functions of growth-promoting exosomes.

Since glutamine is now recognised as a key metabolite in several cancers (Zhang *et al*, 2017), glutamine depletion is likely to be a common event in growing tumours, which would trigger a switch to Rab11a-exosome secretion. Reducing growth factor and associated mTORC1 signalling could have similar effects. Therefore, blocking this form of Rab11a-exosome communication may have therapeutic value in suppressing adaptation and/or resistance mechanisms, a concept that needs to be explored with further *in vivo* experiments. In addition, the use of exosomes as biomarkers for cancer and other diseases has considerable clinical potential (Barile & Vassalli, 2017; Shah *et al*, 2018). Our data suggest that the tumour exosome profile can indicate the response of cancer cells to microenvironmental stresses, anti-angiogenic drugs and inhibitors of Akt/mTORC1 signalling. A better understanding of metabolically regulated exosome secretion should therefore lead to exosome profiling being applied with increasing precision in future diagnosis and in prognostic applications involving such treatments.

# Materials and Methods

### *Drosophila* stocks and genetics

The following fly stocks, acquired from the Bloomington *Drosophila* Stock Centre, unless otherwise stated, were used: *UAS-CD63-GFP* (Panáková *et al*, 2005; a gift from S. Eaton, Max Planck Institute of Molecular Cell Biology and Genetics, Dresden, Germany), *dsx-GAL4* (Rideout *et al*, 2010; a gift from S. Goodwin, University of Oxford, UK), *tubulin-GAL80^ts^*; *UAS-Btl-GFP* (Sato & Kornberg, 2002), *YFP^MYC^-Rab11* and *YFP^MYC^-Rab7* (Dunst *et al*, 2015, a gift from F. Karch and E. Prince, Department of Genetics and Evolution, University of Geneva, Switzerland), *UAS-Stam-RNAi* (TRiP.HMS01429: Ni *et al*, 2011), *UAS-shrb-RNAi* (v106823: Dietzl *et al*, 2007; Matusek *et al*, 2014; Vienna *Drosophila* Resource Centre [VDRC]); *UAS-Vps28-RNAi* (v31894; Dietzl *et al*, 2007; Matusek *et al*, 2014, VDRC), *btl-Gal4* (*btl^NP6593^*; Hayashi *et al*, 2002; Kyoto *Drosophila* Stock Centre); *UAS-GFP^NLS^*; *UAS-YFP-Rab11* (Zhang *et al*, 2007), *UAS-shrb-GFP* (Sweeney *et al*, 2006); and *w^1118^* (a gift from L. Partridge, UCL, UK). The C-terminal end of the human CD63 coding sequence was fused in frame to mCherry, and the resulting gene cloned downstream of UAS sequences in the vector pUAST (Brand & Perrimon, 1993), which was then used to generate transgenic lines.

Flies were reared at 18°C in vials with standard cornmeal agar medium containing, per litre, 12.5 g agar, 75 g cornmeal, 93 g glucose, 31.5 g inactivated yeast, 8.6 g potassium sodium tartrate, 0.7 g calcium and 2.5 g nipagen (dissolved in 12 ml ethanol). They were transferred onto fresh food every 3–4 days. No additional dried yeast was added to the vials.

Temperature-controlled, SC-specific expression of *UAS-CD63-GFP*, *UAS-Btl-GFP* and *UAS-Rab11-YFP* was achieved by combining these transgenes with the specific driver, *dsx-GAL4* and the temperature-sensitive, ubiquitously expressed repressor *tubulin-GAL80^ts^*. Newly eclosed virgin adult males were transferred to 28.5°C to induce post-developmental SC-specific expression. For knockdown experiments, the same strategy was employed, but in the presence of a UAS-RNAi transgene. Six-day-old adult virgin males were used throughout this study to ensure that age- and mating-dependent effects on SC biology were mitigated (Leiblich *et al*, 2012; Redhai *et al*, 2016).

To produce a pulse of Shrb-GFP expression, newly eclosed virgin males were kept at 25°C (permissive for GAL80^ts^ activity) for 6 days, before being transferred to 28.5°C for a 4 h "pulse" of transgene expression under *dsx-GAL4* control, then returned to 25°C for a further 4 h before imaging.

### Preparation of accessory glands for live imaging

Adult male flies were anaesthetised using $CO_2$. The anaesthetised flies were submerged in ice-cold PBS (Gibco®) during the micro-dissection procedure. The whole male reproductive tract was carefully pulled out of the body cavity. The testes were then removed by scission close to the seminal vesicles, as they often folded over the AGs, obscuring imaging. Care was taken to keep the seminal vesicles and vas deferens intact in order to prevent damage to the papilla of the anterior ejaculatory duct. The AG epithelium can be stressed by damage to this valve-like papilla, through which the spermatozoa and AG contents are transferred during mating.

Finally, the remaining reproductive tissues, including the AGs, were isolated by separation of the ejaculatory bulb from the external genitalia, fat tissues and gut.

The isolated AGs were kept in PBS on ice until sufficient numbers had been obtained (typically for approximately 15 min). They were then stained with ice-cold 500 nM LysoTracker® Red DND-99 (Invitrogen®) in $1 \times$ PBS for 5 min. Finally, the glands were washed in ice-cold PBS for 1 min before being mounted onto High Precision microscope cover glasses (thickness No. 1.5H, Marienfeld-Superior). A custom-built holder secured the cover glasses in place during imaging with both the super-resolution 3D-SIM and wide-field systems. To avoid dehydration and hypoxia, the glands were carefully maintained in a small drop of PBS, surrounded by 10S Voltalef® (VWR Chemicals), an oxygen-diffusible hydrocarbon oil (Parton *et al*, 2010) and kept stably in place by the application of a small cover glass (VWR).

### Imaging

#### *Confocal imaging*
Confocal images of cultured cells and AGs were acquired either by using a Zeiss LSM 510 Meta [Axioplan 2], an Olympus FV1000 microscope set-up with 100× NA1.4 oil objective or an LSM 880 laser scanning confocal microscope equipped with a 63×, NA 1.4 Plan APO oil DIC objective (Carl Zeiss). RI 1.514 objective immersion oil (Carl Zeiss) was employed.

#### *Super-resolution 3D-SIM microscopy*
For super-resolution 3D-SIM, fixed human cells or freshly prepared living *Drosophila* AGs were imaged on a DeltaVision OMX V3 system (GE Healthcare Life Sciences) equipped with a 60×, NA 1.42 Plan Apo oil objective (Olympus), sCMOS cameras (PCO), and 405, 488, 593 and 633 nm lasers. Objective immersion oil of RI 1.516 (Cargille Labs) was used to produce optimal reconstruction for a few μm along the z-axis of the sample.

The microscope was kept at approximately 20°C throughout experiments. Images were acquired in sequential imaging mode, with Z-stacks, typically spanning 8–12 μm, being acquired with 15 images per plane (five phases, three angles) and a z-distance of 0.125 μm. The raw data were computationally reconstructed in the softWoRx 6.0 software package (GE Healthcare Life Sciences), using a Weiner filter of 0.002 and wavelength-specific optical transfer functions (OTFs) to generate super-resolution images with twofold improved resolution in all three axes. The ImageJ2 plugin "SIMcheck" (Ball *et al*, 2015) was used to analyse the quality of the reconstructed images. As the OMX system uses separate cameras for each colour, the originally produced images required careful re-alignment. To make the appropriate adjustments, reference Z-stacks of 200 nm TetraSpeck Microspheres (Invitrogen®, T7280) were aligned using custom-automated software (OMX Editor) that corrected for X, Y and Z shifts, magnification and rotation differences between channels. The resulting transformations were then applied to align the images of the test samples.

#### *Wide-field microscopy*
For wide-field imaging, living SCs were imaged using a DeltaVision Elite wide-field fluorescence deconvolution microscope (GE Healthcare Life Sciences) equipped with both a 100×, NA 1.4 UPlanSApo oil objective (Olympus) and a 60×, NA 1.42

UPlanSApo oil objective (Olympus), as well as a CoolSNAP HQ2 CCD camera (Photometrics®) and a seven-colour illumination system (SPECTRA light engine®, Lumencor). A 1.6× auxiliary magnification lens and objective immersion oil with RI 1.514 (Cargille Labs) were used throughout. The microscope room was temperature-controlled at ~ 18°C. The images acquired were typically Z-stacks spanning 8–12 μm depth with a *z*-distance of 0.2 μm. Images were subsequently deconvolved using the Resolve 3D-constrained iterative deconvolution algorithm within the softWoRx 5.5 software package (GE Healthcare Life Sciences).

### Electron microscopy

To image isolated EVs, 10 μl of a purified EV suspension was incubated on a glow discharged 300 mesh copper grid coated with a carbon film for 2 min, blotted with filter paper and stained for 10 s with 2% uranyl acetate. The grid was then blotted and air-dried. Grids were imaged in a Tecnai FEI T12 transmission electron microscope (TEM) operated at 120 kV with a Gatan Oneview digital camera.

For SC studies, a high-pressure freezing (HPF) and freeze substitution method was employed. AGs from 6-day-old virgin $w^{1118}$ males were carefully dissected in ice-cold PBS under a dissection microscope. The glands were then removed from the PBS and coated in yeast paste (dry baker's yeast mixed with 10% methanol), which acts as a cryo-protectant. Four or five glands were transferred using forceps into the well of a membrane carrier (1.5 mm diameter, 100 μm deep, Leica). The carriers were then loaded into bayonet pods (Leica) and transferred using a manual loading device to the high-pressure freezer (EM PACT2, Leica). Samples were frozen immediately at 2,000 bar.

The carriers were carefully moved into pre-cooled 2-ml screw-top cryotubes containing pre-cooled freeze substitution medium (0.1% uranyl acetate and 1% osmium tetroxide in dry acetone) within the sample chamber of an automatic freeze substitution unit (EM AFS2, Leica) held at −130°C. Over the following 40 h, the samples were gradually brought up to room temperature to avoid formation of ice crystals.

Following freeze substitution, the glands were then moved using a plastic Pasteur pipette into fresh 2-ml tubes of 100% acetone. Samples were then gradually infiltrated with Agar 100 Hard resin (Agar Scientific) at room temperature over several days. Samples were first incubated with a 3:1 mix of 100% dry acetone:resin (without accelerator) for 2 h, then a 1:1 mix for 3 h, then a 1:3 mix for 2 h. The samples were left overnight in 100% resin (with accelerator from then on), which was replaced with fresh 100% resin four times over 32 h. The glands were carefully transferred to and positioned in embedding capsules, and the resin was polymerised at 60°C in an oven over 40 h. Ultrathin (90 nm) sections were taken with a Diatome diamond knife on a Leica UC7 ultramicrotome and transferred to 200 mesh copper grids, which were then post-stained for 5 min in uranyl acetate, washed 8 times in water and then stained for 5 min in lead citrate. The grids were washed seven times in water, and allowed to dry. Samples were imaged as described for EV preparations above.

### Drosophila exosome secretion assay

Control virgin 6-day-old SC>CD63-GFP- or SC>Btl-GFP-expressing males and virgin CD63-GFP/Btl-GFP males also expressing an RNAi

against ESCRT components *Stam*, *shrb* or *Vps28* in SCs were dissected in 4% paraformaldehyde (Sigma-Aldrich) dissolved in PBS. The glands were left in 4% paraformaldehyde for 20 min to preserve the luminal contents before being washed in PBS for 5 min. Glands were then mounted onto SuperFrost® Plus glass slides (VWR), removing excess liquid using a tissue and finally immersed in a drop of Vectashield® with DAPI (Vector Laboratories) for imaging by confocal microscopy.

Exosome secretion was measured using a modification of a previously described approach (Corrigan *et al*, 2014), in which multiple areas of luminal fluorescent puncta (exosomes or exosome aggregates) were sampled within the central third of each gland. Identical microscope settings and equipment were used throughout. At each sampling location, three different Z-planes were analysed, spaced by 5 μm for CD63-GFP, and 10 different Z-planes were analysed, spaced by 1 μm for Btl-GFP, with the first image in the Z-stack located exactly 5 μm apical to the main cell nuclei for both markers. This helped to ensure similar fluorescence levels between glands of the same genotype.

The automated analysis of exosome secretions by SCs was performed using ImageJ2 (Schindelin *et al*, 2015), distributed by Fiji. The raw data from each lumenal area were thresholded using the Kapur–Sahoo–Wong (Maximum Entropy) method (Kapur *et al*, 1985) to determine the outlines of fluorescent particles. The "Analyse Particles" function was then used to determine the total number of fluorescent puncta.

### Analysis of ILV content in large non-acidic compartments

Living SCs from 6-day-old SC>Btl-GFP-, SC>CD63-GFP-, or SC>YFP-Rab11-expressing flies (with or without RNAi expression) were imaged using identical settings by wide-field microscopy. The total number of fluorescently labelled non-acidic compartments and the total number of these compartments that contained fluorescently labelled puncta was counted in each cell, using a full Z-stack of the epithelium. Three individual SCs were analysed from each of 13 glands.

### SC compartment analysis

Secondary cell compartment numbers were manually analysed in ImageJ2 using the "Cell Counter" plugin. Complete Z-stacks of individual cells were scored for the number of large non-acidic compartments (LysoTracker Red®-negative) and the number of large acidic late endosomal and lysosomal (LEL) compartments (LysoTracker Red®-positive). The diameter of the largest LEL was measured manually at the Z-plane where the diameter was at its greatest using the line tool (see Corrigan *et al*, 2014; Redhai *et al*, 2016). Both fluorescently labelled and non-transgenic $w^{1118}$ flies were analysed using a combination of fluorescence and DIC microscopy on the wide-field microscope. Three individual SCs per gland from each of 10 glands were used to calculate the mean compartment number in each genotype.

### Measurement of ILV diameter

Super-resolution 3D-SIM images of living SCs expressing CD63-GFP were used to determine the diameter of individual ILVs within non-

acidic compartments. Careful manual measurements were taken of the diameters of ILVs within individual compartments using the line tool in ImageJ2. Many ILVs were smaller than the resolution limits of super-resolution 3D-SIM and were visualised as individual bright puncta (< 100 nm diameter). The three largest compartments per SC were analysed by this method, and the mean was calculated from three separate SCs in each of three glands.

## Cell culture

HCT116, HeLa and LNCaP cells were purchased from ATCC to ensure authenticity and tested regularly for mycoplasma. Cells were grown in McCoy's 5A modified medium (Life Technologies; HCT116), DMEM (Life Technologies; HeLa) or RPMI-1640 medium (Life Technologies; LNCaP), supplemented with 10% heat-inactivated foetal calf serum (FBS), 100 U/ml penicillin–streptomycin (Life Technologies) and incubated in 5% $CO_2$ at 37°C prior to EV collection.

## Extracellular vesicle isolation

During EV isolation, HCT116 and HeLa cells were maintained in serum-free basal medium (DMEM/F12) supplemented with 1% ITS (insulin–transferrin–selenium; #41400045 Life Technologies) (Tauro *et al*, 2013) for 24 h. LNCaP cells were cultured in RPMI-1640 supplemented with 1% ITS for 24 h. For glutamine depletion experiments, cells were grown for 24 h in DMEM/F12 medium without L-glutamine (#21331046; Life Technologies) supplemented with 1% ITS and 2.00 mM or low glutamine (0.15 mM for HCT116; 0.02 mM for HeLa and LNCaP; Life Technologies). For each cell line, a dose–response curve with different glutamine concentrations was produced, and a glutamine concentration that significantly decreased 4E-BP1 phosphorylation without strongly inhibiting S6 phosphorylation was selected. Torin 1 was added at 100 nM (HCT116) or 120 nM (LNCaP), rapamycin at 10 nM, and AZD5363 at 3 μM. Generally, about $8–9 \times 10^6$ HCT116 cells, $4 \times 10^6$ HeLa cells, or $12 \times 10^6$ LNCaP cells were seeded per 15-cm cell culture plate (typically 10–15 plates used per condition) and allowed to settle for 16–24 h in complete medium before a 24-h EV collection. Cells were typically 80–90% confluent at the end of the collection period. The culture medium was pre-cleared by centrifuging at 500 *g* for 10 min at 4°C and 2,000 *g* for another 10 min at 4°C to remove cells, debris and large vesicles. The supernatant was filtered using 0.22-μm filters (Milex®).

For EV isolation by differential ultracentrifugation (UC), large volumes of the EV-containing filtrate were concentrated using a tangential flow filter (TFF) set-up with a 100-kDa membrane (Vivaflow 50R, Sartorius) using a 230V pump (Masterflex). The EV suspension was typically centrifuged at 100,000–108,000 *g* for 70–120 min in a Beckman 55.2 Ti or 70 Ti rotor at 4°C (Beckman Coulter). For activity assays and some experiments involving Western analysis, the EV pellet was resuspended in 25 ml PBS and the ultracentrifugation step repeated to minimise soluble protein contaminants. Finally, the EV pellet was resuspended in 70–100 μl PBS for subsequent experiments. For activity, proteinase K and most NTA assays, EVs were used within 24 h, but for other analyses, including EVs used for xenograft injections, they were stored at −80°C prior to thawing and use on the day of the injection experiment.

To isolate EVs by SEC, EV-containing filtrate from a 2,000 *g* centrifugation was concentrated using a 100-kDa TFF filter to a volume of approximately 30 ml and then further concentrated in 100 kDa Amicon filters by sequential centrifugation at 4,000 *g* for 10 min at 4°C to a final volume of 1 ml. This was injected into a size-exclusion column (column size 24 cm × 1 cm containing Sepharose 4B, 84 nm pore size) set-up in an AKTA start system (GE Healthcare Life Science) and eluted with PBS, collecting 30 × 1 ml fractions. Fractions corresponding to the initial "EV peak" (fractions two to five for HCT116 and HeLa cells and fractions two to seven for LNCaP cells, which appeared before the large protein peak) were pooled in 100-kDa Amicon tubes to a final volume of approximately 100 μl for analysis.

Initially, EV preparations for Western and functional analysis were assessed after normalisation based on the mass of protein in EV-secreting cells and on EV number, determined by NTA. To determine the mass of protein in EV-secreting cells, cells were washed in PBS twice and then lysed on ice in cold RIPA buffer supplemented with protease and phosphatase inhibitors (Sigma). Cell debris was pelleted by centrifugation at 13,000 *g* for 10 min at 4°C. Protein concentration in the supernatant was measured by bicinchoninic acid (BCA) assay (Pierce, Thermo Scientific).

To test whether EV proteins were located inside vesicles, EV samples were resuspended in Dulbecco's PBS (DPBS; Gibco®, #14040-083), then incubated with or without proteinase K at 25 μg/ml (Roche, #03115828001), and with or without 0.1% Triton X (Sigma, #T8787) or with 1.1% LDS (lithium dodecyl sulphate) at 37°C for 30 min. Proteinase K digestion was terminated using 5 mM PMSF (Cell Signaling Technology, #8553S), and the samples were subjected to Western blot analysis.

For EV immunocapture, EV samples were isolated from fifty 15-cm cell culture plates using SEC and then resuspended in 500 μl PBS supplemented with protease inhibitors (Sigma; P8340). For pull-down, 150 μl of each sample was incubated at 4°C overnight with 150 μl of magnetic beads conjugated with CD63 or CD81 antibodies (ThermoFisher; 10606D and 10616D, respectively) or with the equivalent amount of Protein G beads (ThermoFisher; 10612D), which were covalently linked with mouse control IgGs (Cell Signaling Technology; 5415S). To reduce non-specific binding, the beads were pre-washed three times with PBS containing 0.5% BSA. The captured proteins were removed from the beads by incubating with 2× Laemmli buffer at 37°C for 60 min and then at 95°C for 10 min, before analysing by Western blot.

## Nanoparticle tracker analysis (NanoSight®)

The NS500 NanoSight® was used. Thirty-second videos were captured between three and five times per EV sample at known dilution (normalised to protein mass of secreting cells). Particle concentrations were measured within the linear range of the NS500 between about $2–10 \times 10^8$ particles per ml. Particle movement was analysed by NTA software 2.3 (NanoSight Ltd.) to obtain particle size and concentration.

## EV separation by high-resolution iodixanol-PBS density gradients

Extracellular vesicles were isolated using SEC from conditioned medium produced by fifty plates of HCT116 cells grown in either

glutamine-replete or glutamine-depleted conditions for 24 h and concentrated using ultrafiltration at 4,000 *g* for 8 min at 4°C with 100 kDa Amicon Ultra-15 filter units (Milex®) to a final volume of 750 µl. Iodixanol (Optiprep™, Sigma-Aldrich) density medium was prepared with PBS to generate discontinuous gradients (12–36%). A bottom loading method for EV analysis was utilised (Jeppesen *et al*, 2019), whereby EVs were mixed with the 36% iodixanol step and pipetted at the bottom of an ultracentrifuge tube, with subsequent decreasing concentrations of iodixanol carefully layered in four steps (30, 24, 18, 12%) to complete the gradient. The tubes were then ultracentrifuged at 120,000 *g* for 15 h at 4°C using a SW41 Ti swinging-bucket rotor (Beckman Coulter). Twelve 1 ml fractions were then collected from the top of the gradient and weighed to calculate each fraction density. Each fraction then underwent ultra-filtration using 100 kDa Amicon Ultra-4 filter units (4,000 *g* at 4°C for 4–10 min) to give a final volume of approximately 70 µl. The samples were analysed by Western blotting.

## Genetic manipulation of cultured cells

For shRNA knockdown, the following conditions were used. For *Rab11a*, cells were cultured for 2 days following transfection with *Rab11a* shRNA lentivirus (GTTGTCCTTATTGGAGATTC; TRCN0000381243; Sigma) before EV collection. For *raptor*, cells were cultured for 3 days following transfection with *raptor* shRNA lentivirus (Addgene #1858; Sarbassov *et al*, 2005) before EV collection. For *Rab7a*, cells were cultured for 3 days following transfection with *Rab7* shRNA lentivirus (GGCTAGTCACAATGCA-GATAT; in pHR-U6 vector) before EV collection. For NT shRNA, the shRNA sequence was CAACAAGATGAAGAGCACCAA (SHC 202, Sigma).

Transient transfections of fusion protein constructs were carried out when the cells reached 80% confluency with the transfection reagent Lipofectamine® 2000 in serum-reduced OptiMEM® medium (Gibco®, Life Technologies), and following manufacturers' instructions to obtain maximum transfection rates. Constructs were GFP-Rab5CA (Q79L; Addgene #35140) and GFP-Rab11 WT (Addgene #12674).

## Western analysis

Both cell lysates and EV preparations were electrophoretically separated using 10% mini-PROTEAN® precast gels (Bio-Rad). We loaded gels for Western analysis with EV lysates extracted from the same protein mass of secreting cells, so that changes in band intensity on the blots with glutamine depletion reflected a net change in secretion of the marker on a per cell basis. Glutamine depletion did not significantly affect EV number. Furthermore, the normalisation method for analysing EV proteins on Western blots did not affect the overall conclusions from our analysis (Figs 4B and EV4E, Appendix Fig S5). However, after some drug and knockdown treatments, EV production was reduced by up to 50% (Fig EV4E). We continued to load EV protein gels for Westerns based on EV-secreting cell protein mass to ensure that any changes in protein levels did not result from testing EVs produced by different numbers of cells.

Extracellular vesicle preparations were lysed in RIPA or 1× sample buffer. Protein preparations were ultimately dissolved in either reducing (containing 5% β-mercaptoethanol) or non-reducing (for CD63 and CD81 detection) sample buffer and were heated to 90–100°C for 10 min before loading with a pre-stained protein ladder (Bio-Rad). Proteins were wet-transferred to polyvinylidene difluoride (PVDF) membranes at 100 V for 1 h using a Mini Trans-Blot Cell (Bio-Rad). The membranes were then blocked with either 5% milk (optimal for CD63 detection) or 5% BSA in TBS buffer with Tween-20 (TBST) for 30 min and probed overnight at 4°C with primary antibody diluted in blocking buffer. After 3 × 10 min washing steps with TBST, the membranes were probed with the relevant secondary antibodies for 1 h at 22°C. Following 3 × 10 min wash steps, the signals were detected using the enhanced chemiluminescent detection reagent (Clarity®, Bio-Rad) and the Touch Imaging System (Bio-Rad). Relative band intensities were quantified by ChemiDoc® software (Bio-Rad) or ImageJ. Signals were normalised to tubulin (cell lysates), or cell lysate protein or CD81 (HCT116, HeLa and LNCaP EVs). Note, as shown in Figs 4B and EV3E, that if EV protein signals are normalised to cell lysate levels, EV particle number, CD81, Syn-1 or Tsg101, relative levels of Rab11a (and Cav-1) are generally increased after glutamine depletion or partial blockade of mTORC1 signalling, while CD63 is typically reduced.

Antibody suppliers, catalogue numbers and concentrations used were as follows: rabbit anti-4E-BP1 (Cell Signaling Technology #9644, 1:1,000), rabbit anti-p-4E-BP1-Ser65 (Cell Signaling Technology #9456, 1:1,000), rabbit anti-S6 (Cell Signaling Technology #2217, 1:4,000), rabbit anti-p-S6-Ser240/244 S6 (Cell Signaling Technology #5364, 1:4,000), rabbit anti-Raptor (Cell Signaling Technology #2280, 1:1,000), rabbit anti-caveolin-1 (Cell Signaling Technology #3238, 1:500), goat anti-AREG (R&D Systems #AF262, 1:200), mouse anti-Tubulin (Sigma #T8328, 1:4,000), rabbit anti-GAPDH (Cell Signaling Technology #2118, 1:2,000), mouse anti-CD81 (Santa Cruz #23962, 1:500), mouse anti-CD63 (BD Biosciences # 556019, 1:500), rabbit anti-Syntenin-1 antibody (Abcam ab133267, 1:500), rabbit anti-Tsg101 (Abcam ab125011, 1:500), mouse anti-Rab11 (BD Biosciences #610657, 1:500), rabbit anti-Rab11a (Cell Signaling Technology; 2413S, 1:500), rabbit anti-Rab11b (Cell Signaling Technology; 2414S, 1:500), rabbit anti-p44/42 MAPK (ERK; Cell Signaling Technology #4695, 1:1,000), rabbit anti-p-p44/42 MAPK (Cell Signaling Technology #4370, 1:1,000), rabbit anti-PRAS40 (Cell Signaling Technology #2691, 1:1,000), rabbit anti-p-PRAS40-Thr246 (Cell Signaling Technology #2997, 1:1,000), rabbit anti-Rab7 (Cell Signaling Technology #9367, 1:1,000), rabbit anti-calnexin (Abcam #ab213243, 1:1,000), sheep anti-TGN46 (Bio-Rad; AHP500G, 1:1,000), rabbit anti-Annexin A1 (Cell Signaling Technology; 32934, 1:500), anti-mouse IgG (H+L) HRP conjugate (Promega #W4021, 1:10,000), anti-rabbit IgG (H+L) HRP conjugate (Promega #W4011, 1:10,000), anti-goat IgG (H+L) HRP conjugate (R&D Systems #HAF109, 1:100).

## Measurement of AREG concentration by ELISA

Extracellular vesicle samples were lysed in Cell Lysis Buffer 2 (R&D SYSTEMS; 895347) supplemented with protease inhibitors (Sigma; P8340) at room temperature for 30 min. The amount of AREG protein in the EV samples was measured using Quantikine ELISA—Human amphiregulin Immunoassay (R&D SYSTEMS; DAR00) according to the manufacturer's instructions.

## Immunostaining of cultured human cells

For immunofluorescence studies, cultured cells on coverslips were fixed in 4% paraformaldehyde for 15 min at ~ 23°C, washed 3 × 10 min in PBS, and permeabilised with 0.05% Triton X-100/PBS for 30 s. After three PBS washes, cells were incubated with the primary antibody prepared in 5% donkey serum/PBS in a humid chamber at 4°C overnight. The coverslips were washed 4 × 10 min in PBS and then incubated with the appropriate secondary antibodies diluted in 5% donkey serum/PBS for 1 h at ~ 23°C. Following 4 × 10 min PBS washes, the coverslip was mounted on a glass slide with DAPI-containing mounting medium (VectorShield; Vector Laboratories) and sealed with nail polish. Antibody concentrations were as follows: mouse anti-Rab11 (detects both isoforms; BD Biosciences #610657, 1:50), mouse anti-CD63 (BD Biosciences #556019, 1:100), rabbit anti-Rab7 (Cell Signaling Technology #9367, 1:50), mouse anti-CD81 (Santa Cruz; SC-23962, 1:100), rabbit anti-Rab11a (Cell Signaling Technology; 2413S, 1:50), rabbit anti-Lamp1 (Cell Signaling Technology; 9091S, 1:100); mouse anti-Lamp2 (Abcam; ab25631, 1:100), GFP-Booster ATTO 488 (Chromotek gba488-100, 1:500), donkey anti-rabbit IgG (H+L) (DyLight® 550; Abcam ab96892, 1:500), donkey anti-mouse IgG (H+L) (Alexa Fluor® 488 AffiniPure; Jackson ImmunoResearch #715-545-151, 1:500).

Cell viability was measured by Trypan Blue staining. HCT116 cells were trypsinised and then resuspended in McCoy's 5A medium (Gibco®, #26600080). The cell suspension was mixed with Trypan blue solution (Sigma, #T8154) at 1:1 ratio and the number of blue staining (non-viable) cells and total cell number were assessed using a haemocytometer.

## Growth, proliferation, apoptosis and neutralising antibody assays

For growth assays, $2 \times 10^3$ HCT116 cells per well were seeded in a 96-well plate in 100 μl of medium, following pre-treatment with freshly prepared EVs for 30 min in PBS at 37°C (up to $10^4$ EVs [isolated by UC or SEC] per cell for controls, which is roughly the number of EVs secreted by each HCT116 cell in 24 h). The cells were maintained in McCoy's 5A medium in reduced serum (1% FBS) conditions. Growth was followed over a period of 5 days by live image acquisition using the IncuCyte ZOOM® analysis system (10× magnification; Essen Bioscience) to automatically detect cell edges and to generate a confluence mask for cell coverage area calculation, which was normalised to confluence at zero time (giving fold change in confluency). Each biological replicate is represented as mean of at least eight technical replicates.

For ERK inhibition experiments, SCH772984 (S7101; SelleckChem) was added to the cell suspension along with the exosomes to a final concentration of 1 μM. The medium was changed 24 h after cell plating.

To assess cell proliferation, cells were fixed with methanol and stained with DAPI. The number of nuclei/field for three fields per well was counted manually from fluorescence microscope images. To measure levels of apoptosis, caspase-3 and caspase-7 activities were assayed using 1 μM of Cell Player® caspase-3/7 reagent (Essen Bioscience), monitored with the IncuCyte ZOOM® system.

AREG-neutralising antibody experiments were performed in a 96-well plate using $2 \times 10^3$ HCT116 cells per well treated with 4,000 SEC-isolated EVs per recipient cell, using goat anti-AREG (R&D

Systems #AF262, 1:200), and goat IgG (R&D Systems, AB-108-C) as a control antibody treatment. Prior to mixing with recipient cells, EVs were pre-incubated with 4 μg/ml anti-AREG or control antibody in McCoy's 5A medium with 1% FBS for 2 h at 37°C. The mixture was then added to the recipient cells (also in reduced serum media) and allowed to incubate for a further 30 min before plating. Growth was monitored using the IncuCyte ZOOM® analysis system in the same fashion as standard growth assays described above.

All assays were repeated on at least three independent occasions.

## Tubulation assay with HUVEC cells

Human umbilical vein endothelial cells were suspended in LONZA medium (endothelial basal medium-2—EBM2) containing freshly prepared exosomes ($10^4$ UC-isolated EVs per cell). After 10-min incubation, $6 \times 10^3$ cells were seeded onto a solidified thin layer of BD phenol red-free Matrigel (BD Bioscience) in a 96-well plate and incubated at 37°C. Images of vessel growth were captured at 3-h intervals for 18 h by live image acquisition using the IncuCyte ZOOM® microscope (10× lens). Quantification of tubule length was performed using ImageJ. Each biological replicate is represented as mean of at least eight technical replicates.

## Assessment of EV-induced ERK activation

To assess ERK activation upon EV treatment, HCT116 cells were plated in six-well plates. After 24 h, the cells were serum-starved for an additional 18 h. At this point, freshly prepared exosomes ($10^4$ UC-isolated EVs per cell) were added and the mixture incubated for 1 h at room temperature. Cell lysates were generated and analysed by Western blot.

## Cellular uptake assays

For confocal analysis of Tfn uptake, $5 \times 10^4$ HCT116 cells were plated on a coverslip in McCoy's 5A medium supplemented with 10% FBS and cultured overnight, followed by a 24-h incubation in DMEM/F-12, 1% ITS, with 2.00 mM or 0.15 mM glutamine at 37°C. The cells were then washed using cold PBS and cultured in DMEM/F-12, 1% ITS, containing transferrin-Alexa Fluor® 488 (Thermo-Fisher Scientific #T13342, 1:200) with the appropriate glutamine concentration at 4°C for 30 min, followed by a 30-min incubation at 37°C. The cells were fixed for confocal imaging. For the anti-CD63 antibody uptake experiments, cells washed with cold PBS were cultured in DMEM/F-12, 1% ITS, containing mouse anti-CD63 antibodies (BD Biosciences # 556019, 1:50) and the appropriate glutamine concentration at 4°C for 30 min. The cells were then washed with PBS, prior to 30-min incubation at 37°C and a subsequent 30-min chase in DMEM/F-12, 1% ITS with the appropriate glutamine concentration. The cells were fixed, incubated with secondary antibody for the anti-CD63 uptake experiments and then imaged by confocal microscopy.

For Western analysis, $2 \times 10^5$ HCT116 cells were plated in each well of a six-well plate in serum-supplemented McCoy's 5A medium. After 20 h, the culture medium was replaced by DMEM/F-12, 1% ITS containing 2.00 mM or 0.15 mM glutamine. After 24 h, the cells were shifted to 4°C for 10 min to stop intracellular vesicular trafficking and then incubated in DMEM/F-12, 1% ITS, containing

mouse anti-CD63 antibodies (1:50) and 50 μg/ml transferrin-biotin (Sigma #T3915) with appropriate glutamine concentration (2.00 or 0.15 mM) at 4°C for 30 min. After washing with cold PBS, the cells were incubated in DMEM/F-12, 1% ITS, with appropriate glutamine concentration at 37°C for 15 min and then lysed using RIPA buffer.

## Generation and histological analysis of HCT116 tumours in xenograft mouse models

All procedures were carried out in accordance with Home Office licence 30/3197. HCT116 xenografts were established by subcutaneous injection of $2.5 \times 10^6$ cells into the flank of seven female CD1 nude mice (aged 5–6 weeks; Charles River) per group. When tumours had grown to an average group tumour volume of 100 mm³, then mice were treated with four intratumoral injections ($1 \times 10^8$ EVs/100 μl exosome suspension in PBS or PBS vehicle) at 3-day intervals in a semi-blind study, as previously described (Sheldon *et al*, 2010). Tumour growth was measured three times a week using callipers and volume calculated using the formula $1/6$ $\pi \times$ length $\times$ width $\times$ height. Twenty-four hours after the last injection, tumours were excised and half of the tumour fixed in 4% formalin overnight at 4°C before being processed and embedded in paraffin wax and sectioned for histological evaluation.

A standard haematoxylin and eosin protocol was followed to assess the morphology and the amount of necrosis in xenografts. For immunohistochemistry staining, slides were dewaxed and antigen retrieval performed at pH 6. Slides were stained using the FLEX staining kit (Agilent). Endogenous peroxidase activity was blocked before slides were stained with antibodies diluted 1:100, namely rabbit anti-Ki67 (M7240; Dako, Glostrup, Denmark), rabbit anti-CA9 (M75; BD Biosciences), and mouse anti-CD31 (JC70, Dako) for 1 h at 22°C. Slides were incubated with the appropriate anti-rabbit/anti-mouse HRP-coupled secondary antibody (Dako) for 30 min at room temperature and washed in Flex buffer and 3,3′-diaminobenzidine (Flex-DAB; Dako) was applied to the sections for 10 min. The slides were counterstained by immersing in Flex-haematoxylin solution (Dako) for 5 min, washed and air-dried before mounting with mounting medium (Sigma). Secondary-only control staining was done routinely, but was consistently negative.

Expression of CA9 and Ki67, as well as vessel number and size, and viable/necrotic areas were quantified on whole sections using the Visiopharm Integrator System. The HDAB-DAB colour deconvolution band was used to detect positively stained cells. Threshold classification was employed to identify necrotic and living regions and then identify number of positive-staining cells within the latter regions. Threshold levels were checked against control xenograft staining before being set and the xenografts from all groups were then analysed in the same way. Vessel number and size were determined as part of a post-processing step, where areas surrounded by CD31-positive endothelial cells were automatically filled in and the areas generated were counted and quantified.

## Statistical analysis

Before parametric tests were employed, the Shapiro–Wilk test was used to confirm normality of the datasets. For larger datasets, Levene's test was used to evaluate equality of variances. For Western blots, relative signal intensities were analysed using the Kruskal–Wallis test. For the growth and tubulation assays, data were analysed by two-way ANOVA. For SC analysis, a one-way ANOVA test was employed for normally distributed data, with *post hoc* Dunnett's two-tailed *t*-tests used to directly compare individual control and experimental datasets. Non-parametric data were analysed using a Kruskal–Wallis test. We employed power calculations based on previous experiments to ensure the appropriate numbers of animals were used. In order to provide a statistical power of at least 80% to detect a twofold difference in the mean tumour volume between groups, with tumour volume variation of up to 50% within groups, and a statistical difference level of $\alpha = 0.05$, a sample size of seven mice per group is required.

**Expanded View** for this article is available online.

## Acknowledgements

We thank C. P. Alves for his contributions to the work presented. We are grateful to I. Dobbie and to A. Pielach for technical expertise and support with microscopy, which was undertaken in the Wellcome Trust-funded MICRON Oxford Advanced Bioimaging Unit and Dunn School EM Facility. We thank S. Eaton, S. Goodwin, E. Prince and F. Karch, as well as the Bloomington, Vienna and Kyoto Stock Centres for *Drosophila* stocks; J. Whitburn for cells; and the Developmental Studies Hybridoma Bank (Iowa) for antibodies. We are particularly grateful to K. E. Carr for comments on the manuscript. We acknowledge the support of Cancer Research UK (C19591/A19076, C602/A18974), the Cancer Research UK Oxford Centre Development Fund (C38302/A12278), the BBSRC (BB/K017462/1, BB/L007096/1, BB/N016300/1, BB/R004862/1), the Breast Cancer Research Foundation (ANR 00162), the Wellcome Trust (Strategic Awards #091911, #107457 and 102347/Z/13/Z), the National Institute for Health Research (NIHR) Oxford Biomedical Research Centre (BRC) and the John Fell Fund, Oxford (141/063).

## Author contributions

Experimental design conceptualisation, S-JF, BK, PPM, EMB, JDM, KM, CEZ, HS, NKA, EJ, ME, MIS, CCM, SMW, CC, FCH, JFM, ALH, CW, DCIG; experimental work, S-JF, BK, PPM, EMB, JDM, KM, CEZ, EJ, JFM; data analysis, S-JF, BK, PPM, EMB, JDM, KM, CEZ, HS, NKA, EJ, ME, MIS, JFM; writing and editing manuscript, CW, DCIG; reviewing and editing manuscript, S-JF, BK, PPM, EMB, JDM, KM, CEZ, HS, NKA, EJ, ME, MIS, CCM, SMW, CC, FCH, JFM, ALH; and funding acquisition, ALH, CW, DCIG.

## Conflict of interest

The authors declare that they have no conflict of interest.

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
