## [Review Process File · The EMBO Journal]

Glutamine deprivation alters the origin and function of cancer cell exosomes

Shih-Jung Fan, Benjamin Kroeger, Pauline P. Marie, Esther M. Bridges, John D. Mason, Kristie McCormick, Christos E. Zois, Helen Sheldon, Nasullah Khalid Alham, Errin Johnson, Matthew Ellis, M. Irina Stefana, Cláudia C. Mendes, S. Mark Wainwright, Christopher Cunningham, Freddie C. Hamdy, John F. Morris, Adrian L. Harris, Clive Wilson, Deborah C. I. Goberdhan

Review timeline:

Submission date:	21st Jul 2019
Editorial Decision:	9th Sep 2019
Revision received:	23rd Dec 2019
Editorial Decision:	31st Jan 2020
Revision received:	9th Feb 2020
Accepted:	10th Feb 2020

Editor: Daniel Klimmeck

Transaction Report:

1st Editorial Decision

9th Sep 2019

Thank you for the submission of your manuscript (EMBOJ-2019-103009) to The EMBO Journal. Your manuscript has been sent to three reviewers, and we have received reports from all of them, which I enclose below.

As you will see, the referees acknowledge the potential interest and novelty of your results, although they also express a number of major issues. In more detail, referee #3 states that the characterization of the Rab11-positive compartment as exosomes and the details of Rab11's functional involvement are not at this stage, which in his-her view undermines the impact of the work (ref#3, pts standfirst, pts. 1,2). Further, this referee asks you to test the Kras-dependence of your results (ref#3, pt.3) and clarify consistency of experimental settings (ref#3, pt.10; see also ref#2, pt.1-1). Referee #2 agrees in that the distinctive nature of CD63-positive versus Rab11-positive exosomes has not been sufficiently demonstrated and points to inconsistencies in the model proposed (ref#2, pts.1,2). Referee #1 states that the subcellular origin of Rab11-positive EVs has to be explored in more detail (ref#1, pt.1). In addition, the reviewers raise a number of issues related to methods annotation, data representation, statistics and appropriate citation of literature references as well as clarity and flow of the overall manuscript that would need to be conclusively addressed to achieve the level of robustness and clarity needed for The EMBO Journal.

I judge the comments of the referees to be generally reasonable and given their overall interest, we are happy to invite you to revise your manuscript experimentally to address the referees' comments. Given the multitude of matters brought-up and the open outcome of the revisional work, I need to stress though that we would need strong support from the referees on a revised version of the study in order to move on to publication of the work.

REFEREE REPORTS:

Referee #1:

The article by Fan et al demonstrates a novel pathway of secretion of extracellular vesicles induced upon glutamine deprivation of the cells. These vesicles form intracellularly inside multivesicular compartments, thus are called exosomes, but these compartments are not classical CD63+ late endosomes, but rather Rab11+ recycling endosomes. Furthermore, the resulting EVs display novel functional properties, such as enhanced proliferation of tumor or endothelial cells, which are due to overexpression of an EGFR ligand, the transmembrane protein AREG, on the glutamine deprivation-induced exosomes.

These results are novel and very interesting, especially in light of the growing understanding of the heterogeneity of extracellular vesicles, and their different intracellular origins: classically MVB and plasma membrane are described as the places of vesicle budding and secretion, but here the authors show that other intracellular compartments can be a source of EVs.

The authors also provide functional evaluation of the EVs in vitro and in vivo.

All EV experiments are performed in a manner that properly follows the ISEV guidelines (quoted here: They et al 2018), in terms of reporting, isolation, characterization.

I only have minor comments:

1) can the authors really exclude the possibility that part or all of the Rab11/Cav1/AREG-containing EVs could form by direct budding at the plasma membrane, rather than inside internal recycling endosomes? Additional IF images of HCT116 cells could possibly be shown in figure 3, displaying localization of Rab11a in the plasma membrane area, with CD81 as a marker of this area, and possibly AREG. Of note, CD81 is not an endosomal marker, as wrongly suggested in the introduction p4, it is in most cells expressed at the PM or possibly sub-PM compartments. the quality of IF images in figure 3 is not very satisfying

2) the decrease of global CD63 expression in the cells upon glutamine deprivation is not really discussed or explained: is CD63 degraded in lysosomes? is it localized in different intracellular compartments than in control cells? This decrease makes it difficult to interpret the anti-CD63 uptake experiment shown in figure 5D: is uptake really compromised, or is it impossible to do the experiment in the glutamine-deprivation condition because the Ab will not bind any CD63 at the cell surface?

3) The first two figures showing intracellular compartments in the Drosophila accessory gland are interesting and nice in suggesting the existence of the Rab11+ compartments with internal vesicles and of Rab11+ EVs and Btl secretion, but maybe a bit difficult to follow for someone who is not used to look at images of this experimental model.

For instance, Figure S1 may be more informative than current main figure 1.

The images do not show the plasma membrane pattern of the various molecules in secondary cells, thus again, do not make it possible to determine whether PM could contribute to the EVs found in the accessory gland lumen. Of note, ESCRT-I and -III are known to be also involved in budding and release of EVs from the PM.

It is surprising to see such heterogeneity in the LysoTrackerRed pattern in the different panels: can the authors explain?

In figure 1E, it would have been interesting to show if Btl and Rab11 colocalize, at the PM or in internal compartments, to strengthen the message that Btl+ EVs come from this pathway,

Finally, for the link with the rest of the article, is there any glutamine-deprivation situation in the accessory gland of drosophila?

Referee #2:

This is an interesting study identifying an alternate class of exosomes that are marked by Rab11 and have particular functional properties in terms of promoting cell proliferation. This study will be of interest to a broad audience studying exosomes, growth factor signaling, and cell proliferation.

There are two major issues which need clarification to make the main messages of this manuscript solid:

1. What is the relationship between CD63+ and Rab11+ exosomes. Are they the same particles or different particles? There are several statements in the manuscript indicating that the two classes of exosomes have different origins and carry distinct cargo, and many experiments in the manuscript showing that when Rab11a+ exosomes increase CD63+ exosomes decrease (e.g. Fig. 4B) yet some of the data do not fit this model:

1-1 In Fig 4D comparing the two glutamine conditions, there are equal levels of Syn-1, Tsg101 and CD81 hence presumably the same number of exosomes particles in the two samples. In the low-glutamine sample there is more Rab11a, but not less CD63 compared to the high-glutamine sample. This does not fit the simple model that there are more Rab11a+ exosome particles and less CD63+ particles. Some possible options are that in the low-glutamine sample
-there is more Rab11a protein per Rab11a+ exosome (ie not more Rab11a+ particles, but more Rab11a protein per particle)
-Rab11a now also gets loaded on CD63+ particles
-Rab11a+ particles also contain CD63

Which of these scenarios is the explanation? One way to look at this would be to IP CD63+ exosomes and check if they contain Rab11a in the two conditions. Additionally, one could perform microscopy as in Fig 3D to check Rab11/CD63 colocalization, but in the low-glutamine condition.

1-2 The same issue arises in Fig 4G upon Akt inhibition. There is more Rab11a but not less CD63.

1-3 Regarding the Drosophila data:

-The Drosophila data seem to indicate the opposite - that CD63 and Rab11 colocalize. Does this suggest they are not mutually exclusive in the fly?
-The authors claim that the vesicles in the SCs marked by CD63-GFP are Rab11-positive. Co-staining would be necessary to clarify if this is indeed the case.
-The same is true for Rab11 / Btl colocalization. The authors write "we, therefore, conclude that Rab11 and Btl are selective membrane-associated markers for exosomes generated in Rab11-compartments of SCs, which we describe as 'Rab11-exosomes'." but Rab11 / Btl colocalization is not shown.
-Same is true for Shrb and Rab11

1-4 Much of the data (e.g. Fig 4E-F) showing that TOR inhibition leads to increased Rab11a and decreased CD63 come from normalizations relative to CD81. Is it possible instead to normalize to EV particle amounts? Does this show the same result?

2. The link between pERK and increased proliferation caused by the low-glutamine derived exosomes is not completely solid.

-In Fig 6B are the cells still proliferating in the presence of ERK inhibitor? Or does it completely shut down proliferation, in which case it would show an epistatic effect no matter what stimulus is given to the cells?

- The AREGAb nicely blunts the proliferative effect of the low-glutamine derived EVs. Does it also blunt the difference between high-glutamine and low-glutamine derived EVs on pErk, as in Fig 6A?

Minor Issues:

1. Is Rab11 just a cargo/marker for this class of exosomes, or required for Rab11+/Cav-1+ exosomes? Fig 5C shows that glutamine removal increases the amount of Rab11+/Cav-1+ exosomes (both markers increase). Upon Rab11 KD, there is still more Cav-1+ in the EVs. So does this suggest that Rab11 is a marker for this class of exosomes, but not required for their biogenesis?

2. A question for the discussion: In the fly, from the following text:

"In contrast, a YFP-Rab7 gene trap fusion protein (Dunst et al., 2015) primarily trafficked to acidic LELs (Figure 1D) and marked very few puncta in the AG lumen." it appears there are mainly rab11 exosomes and not rab7 exosomes. In humans, instead, the authors suggest it's the other way around and the Rab11 exosomes are only stress induced. Is this a difference between the two systems? Or do SCs have some basal stress levels because, perhaps, they are highly secretory?

3. Page 6 - the abbreviation "ILV" should be spelled out the first time it is used.

4. Fig 4A - the authors conclude that the HCT116 cells "maintain growth factor signaling" when transferred from complete medium to serum free medium supplemented with insulin, however they do not show pS6 or p4EBP levels in cells in complete medium as a comparison. This is needed to claim the cells maintain growth factor signaling. (Otherwise, the levels of pS6 or p4EBP shown in Fig 4A could be only 1% of complete medium, but a long enough exposure of the blot will give a signal.)

5. Relating to the torin treatment of HCT116, the authors conclude that "Secretion of all exosome markers was significantly reduced (Figure S5D', S5E'), suggesting a general shut-down in exosome release."

Indeed the EV number drops from 0.89 to 0.5, which is a reduction but not a general shut-down. Furthermore, in Fig S5D' one sees that Syn1 and Tsg101 levels in the EV isolation are ok. (Presumably because this is what is being used to normalize the EV loading) But nonetheless, these exosomal markers are present, also suggesting there is not a general shut-down in exosome release. Instead, the exosomal cargo seems to change because CD81, CD63 and Rab11a levels are dropping. I wonder whether the interpretation of the data is correct ?

6. Fig 5a - why are the error bars on the lower graph larger, yet the significance of the difference is *** compared to the * in the upper graph?

7. Fig 5a, upper panel - the lower part of the error bars appears to be missing. Or, if the error bars are only shown in one direction, this inconsistent across panels.

8. X-axis labeling Fig 5B has an error (the 2nd to last set should be without ERKi).

9. Fig 4F quantification - data points (dots) on the Rab11a and Cav-1 bars must be missing because all the dots shown are lower than the average.

10. The order in which the panels are referenced in the text does not always match the order in the figures. This makes it a bit difficult to follow.

11. Some main-figure bar graphs are missing the overlaid individual data dots (e.g. Fig 5E, 6A)

Referee #3:

In this manuscript by Fan et al., the authors propose the existence of two distinct populations of exosomes based on cell state-regulated biogenesis. One type would be considered classical CD-63-containing endosome-derived exosomes and the other is a new population of exosomes derived from Rab11a-positive recycling endosomes. This latter population of exosomes are enriched by glutamine deprivation and mTORC1 inhibition. These Rab11a-specific vesicles are growth promoting and exert their EGFR inducing activity, at least in part, through secretion of AREG. The release of this putative new class of exosomes is thought to be an adaptive response to metabolic cell stress. Utilizing a Drosophila model they previously developed and a limited number of human cell lines, the authors present a considerable amount of interesting data. However, there is insufficient characterization of this population of exosomes to call them a unique exosomal vesicle. High-speed UC-based methods are used here isolate an EV pellet containing a heterogenous population of

vesicles, including exosomes as well as non-vesicular components. Additional purification and characterization are needed to determine whether these "new" EVs depend on Rab11a for their biogenesis and are, in fact, exosomes. EV pellets have been shown in multiple studies to contain Rab11 but Rab11a and Rab11b can only be detected by antibodies directed to the variable regions between these two isoforms and it is unclear if the antibody used here distinguishes these isoforms. Although there are several papers that describe the importance of Rab11 in exosome biogenesis, the evidence presented here is insufficient to infer that this constitutes a new class of "recycling exosomes". The authors do present substantial evidence that glutamine depletion and mTORC1 inhibition may influence secretion of exosomes - actually the preferred term at this point would be small extracellular vesicles (sEVs). Based on the evidence presented, the authors cannot rule out that canonical exosomes, or other type of sEVs, are not also carriers for Rab11a. The authors should note that there is evidence that Rab11a is involved in secretion of canonical CD63-positive exosomes (van Niel et al. 2018, Nat Rev Mol Cell Biol).

Major concerns

1. For the authors to claim that they have discovered a novel type of distinct Rab11a-positive exosomes, they must do definitive and comprehensive characterization of this new type of vesicle using a variety of different techniques and must use a validated Rab11a antibody. A few approaches are listed here.
2. Super resolution 3D SIM imaging can be used to examine whether Rab11 is in the ILV of recycling MVBs rather than simply being present in MVBs. Gradient fractionation could be used to determine which fractions contain Rab11 and other exosomal markers (Kowal et al 2016, PNAS). In addition, magnetic beads conjugated to a CD63 antibody or an antibody against a transmembrane protein found in the putative new population of exosomes could be used to capture these two populations to provide a fuller characterization of the composition of this new population and contrast it to classical exosomes (see Kowal et al 2016, PNAS; Jeppesen et al 2019, Cell). Crucially, the authors must be able to demonstrate that CD63-positive and Rab11-positive exosomes are distinct entities and not simply the same exosomes but with increased or decreased CD63 and Rab11a (and other cargoes) due to glutamine depletion or mTORC inhibition.
3. The rationale for the choice of the different cell lines is not clear. There is substantial evidence that mutant KRAS alters cancer cell dependence on glutamine. The HCT116 human colorectal cancer cell line has mutant KRAS. It might be instructive to use isogenic cell lines that differ only in their KRAS status. It is also not clear why HCT116 cells were chosen as recipient cells. Perhaps KRAS wild-type colorectal cancer cells might be a better choice.
4. The manuscript has validated limited cargos by western blot under glutamine-depleted conditions. Global proteomic analysis may lead to identification of additional cargos differentially altered due to glutamine depletion or mTORC1 inhibition.
5. Authors state, "However, there was an increase in Rab11a and Cav-1 relative to CD81 (Figure 4F, I), consistent with induction of an mTORC1-regulated switch in the balance of exosome secretion from late endosomal to Rab11a-compartments." Because this treatment also reduces the number to total EVs released it may not be a switch in secretion but a loss of the late endosomal pathway with the "Rab11a" pathway simply being maintained. One way to resolve this issue that is also raised in the model proposed in Figure 7E, is to determine whether CD63 knockdown affects Rab11-derived EVs with and without glutamine depletion.
6. Figure 1 C stained for Rab11, the authors stated that due to the low fluorescence intensity relative to CD63-GFP, they could not image YFP-Rab11 using super-resolution microscopy (Figure 1C). However, for Figure S1D, the authors use wide field to image "Rab11a" when YFP-Rab11a was overexpressed? Without high resolution imaging, the author cannot conclude that Rab11 is inside of Rab11-positive ILV only based on the presence of Rab11a inside the MVBs. The same argument holds for Btl. The authors state that "Rab11 and Btl are selective membrane-associated markers for exosomes generated in Rab11-compartments of SCs, which we describe as "Rab-11-exosomes". The data presented, in our judgement, do not support this conclusion.

7. There are no high-resolution images to show Rab11 localization when the cells are exposed to glutamine depletion or mTORC1 pathway inhibition. How did the author ascertain that Rab11 only localizes to recycling exosomes under these conditions? How did the authors know that the sEV pellets obtained by UC consist purely of "Rab11-recycling exosomes"? As mentioned above, in order to claim a separate Rab-11 recycling exosome, they should combine beads capture, gradient purification and high-resolution images to further characterize these Rab-11-containing compartments and EVs.

8. Different ways of normalizing proteins in various figures is confusing. In some cases, proteins are normalized to total cell lysate proteins. On Page 33, authors state " we elected to analyze EVs based on EV-secreting cell mass to ensure that any changes in protein levels did not result from testing EVs produced by altered number of cells." What does this sentence mean? The author needs to explain the normalization in detail and why this makes sense. Also, for all the EV western blot quantification, levels were normalized to CD81. Why is CD81 chosen to be the standard for normalization? In general, protein levels can be normalized to total EV number or total protein level in EVs. Cell lysate normalization is an apples to oranges comparison. CD81 presence is associated with subtypes of EV in a heterogeneous mixture of EVs and can't be used as a normalization control. Ideally, a normalization control would be a common protein used in all EV biogenesis; due to the heterogeneous nature of EVs this is hard to determine.

9. Different non-comparable EV isolation methods are used (some use SE, others UC method). Each method has to have purified fractions fully characterized to allow comparison. Without a common method of isolation, such differently purified EVs are not comparable. For instance, in Figure 4 E and G, Torin1 and AKT inhibitor AZD5363 treatment, EVs are isolated using SEC method. However, for Figure 4 F, in the Raptor knockdown experiment, EVs were isolated by UC. This is also the case for Figure 5D' and 5E'.

10. Different glutamine concentrations were used for glutamine depletion in different cells, from 0.15mM to 0.02 mM or 0. Why? Different cells may respond to different levels of Gln depletion but there needs to be some metric for comparison or a discussion of why these are comparable.

11. All Western blots must have molecular weight markers to allow the reader to assess if the correct protein is present.

12. In Figure 6C, there is a Western for AREG. There are several isoforms of AREG differing in size in EVs. One can't tell which isoforms are present in this Western blot. Multiple publications have shown that AREG in EVs is very potent and these should be cited. The authors should consider looking at the effect of AREG neutralizing antibody on p-EGFR in recipient cells. What population of exosomes produces AREG - classical exosomes or their putative new class of exosomes? Authors might consider quantitating the amount of AREG per EV and do a concentration curve when adding these to recipient cells.

13. For Figure 4 H, it is surprising that CD63 would not be digested by both proteinase K and Triton X-100 treatment while another tetraspanin, CD81, appears to be digested. The authors need to provide an explanation for their findings.

14. The blots of Figure 4E, 4F and 4G clearly showed that Torin1 treatment or raptor knockdown or AKT inhibitor treatment did not change the Rab11 levels in those EVs. However, the authors conclude that Rab11a level strongly increased under these conditions when the cargo levels are normalized to CD81. This comparison is not an appropriate way to normalize the level of cargos. Equal amounts of proteins or equal number of vesicles should be loaded for each sample compared.

Minor concerns

1. Some of the blots are not publication quality. For example, figure S5D, CD63 and 4E-BP1; Figure S6D, for S6; Figure 6A.

2. Figure S5 F, based on the western blot, it is hard to tell that CD63 levels are strongly reduced when Raptor was knocked down even though the quantification reflected this. Quantification from three separate biological replicates is needed.
3. Figure 6B, the last two bars appear to be incorrectly labeled.
4. For Figure S3D, E and F, why are only the bar charts shown? The imaging associated with these should be shown.
5. Figure 7C, Y-axis is not labeled. What does fold change in confluency mean? Can the authors provide an explanation for why the Gln 0.15m M AREG Ab treatment has the highest fold change?
6. Figure S4E has a bar chart showing changes in levels of putative exosome proteins in EVs purified by UC. What is the Western blot corresponding to in this bar chart? As noted above, we don't think cell lysates serve as the appropriate control. normalized to cell lysates?
7. For Figure S5E, which method was used for EV isolation?
8. For Figure S5G, when the AKT inhibitor AZD5363 was added, what is the level of phospho-AKT?
9. For figure S5E, why are the levels of phospho-S6 and 4E-Bp1 not examined following exposure to rapamycin?

1st Revision - authors' response

23rd Dec 2019

Point-by-point response to the editorial decision letter

1. *'referee #3 states that the characterization of the Rab11-positive compartment as exosomes and the details of Rab11's functional involvement are not at this stage, which in his-her view undermines the impact of the work (ref#3, pts standfirst, pts. 1,2).'*

Response: Using density gradients, we show that Rab11a has the same fractionation pattern as exosome, and not microvesicle, markers (**ref#3, pt.1,8**). We have used immunocapture methodologies to show that CD63-positive exosomes carry negligible, if any, Rab11a. Rab11a is indeed, mainly associated with vesicles that also lack the 'classical' exosome marker, CD81 (**ref#3, pt.1,2**). Technical challenges in the EV field limit the resolution of different exosome populations inside human cells and microvesicles at the cell surface. We have, however, performed some additional imaging experiments with our fly model and in human cells to provide further evidence for the presence of Rab11-positive vesicles in Rab11-compartments (**ref#3, pt.6,7**) and to argue against the hypothesis that the plasma membrane is a source of these EVs (**ref#1, pt.1**).

2. *'Further, this referee asks you to test the Kras-dependence of your results (ref#3, pt.3) and clarify consistency of experimental settings (ref#3, pt.10; see also ref#2, pt.1-1).'*

Response: We have used a recently characterised KRAS inhibitor, BI-2852, to show that KRAS does not play a major role in controlling Rab11a-exosomes (**ref#3, pt.3**); repeated the experiment highlighted by **ref#3, pt.10** to confirm our findings under our standard HCT116 cell glutamine depletion conditions; explained how appropriate glutamine depletion concentrations were determined for different cell lines (**ref#3, pt.10**), and assessed the effects of

glutamine depletion on subcellular Rab11a localisation, showing that it remains distinct from CD63 (**ref#2, pt.1-1**).

3. *'Referee #2 agrees in that the distinctive nature of CD63-positive versus Rab11-positive exosomes has not been sufficiently demonstrated and points to inconsistencies in the model proposed (ref#2, pts.1,2).'*

Response: We have now shown by immunocapture that CD63- and Rab11a-positive exosomes are distinct (**ref#2, pt.1**). Furthermore, we now demonstrate that glutamine depletion-induced EVs promote an increase in ERK activation in target cells, which is selectively blunted by neutralising anti-AREG antibodies; we also include data showing that the ERK inhibitor does not completely shut down growth and proliferation in these cells (**ref#2, pt.2**).

4. *'Referee #1 states that the subcellular origin of Rab11positive EVs has to be explored in more detail (ref#1, pt.1).'*

Response: Using the approaches suggested by **ref#1**, we show most peripheral Rab11a is in compartments below the cell surface and remains separated from CD63 inside the cell (**ref#1, pt.1**; see also 1 above for further arguments that Rab11a-marked vesicles are exosomes). We agree with this referee that, as with other proteins associated with exosomes, it is technically very difficult to completely eliminate the possibility that a small subfraction of a protein marker reaches the cell surface transiently. Indeed, interpreting such experiments is also challenging: CD81 is highly concentrated at the plasma membrane, yet is thought to be an exosome-specific marker.

5. *'In addition, the reviewers raise a number of issues related to methods annotation, data representation, statistics and appropriate citation of literature references as well as clarity and flow of the overall manuscript that would need to be conclusively addressed to achieve the level of robustness and clarity needed for The EMBO Journal.'*

Response: We have addressed these issues in multiple parts of the manuscript.

Point-by-point response to the referees' comments

We are grateful to the referees for their detailed analysis of our manuscript. We address their points below, using their numbering, with other points marked by letters. For new references added to the manuscript, we give the full citation when first mentioned here. Since the figure numbering has been significantly changed as a result of the new data, we use 'new' = revised, and 'old' = original manuscript.

Referee #3 (ref#3):

- A.** *'However, there is insufficient characterization of this population of exosomes to call them a unique exosomal vesicle. High-speed UC-based methods are used here isolate an EV pellet containing a heterogenous population of vesicles, including exosomes as well as non-vesicular components. Additional purification and characterization are needed to determine whether these "new" EVs depend on Rab11a for their biogenesis and are, in fact, exosomes. EV pellets have been shown in multiple studies to contain Rab11 but Rab11a and Rab11b can only be detected by antibodies directed to the variable regions between these two isoforms and it is unclear if the antibody used here distinguishes these isoforms. Although there are several papers that describe the importance of Rab11 in exosome biogenesis, the evidence presented here is insufficient to infer that this constitutes a new class of "recycling exosomes". The authors do present substantial evidence that glutamine depletion and mTORC1 inhibition may influence secretion of exosomes - actually the preferred term at this point would be small extracellular vesicles (sEVs). Based on the evidence presented, the authors cannot rule out that canonical exosomes, or other type of sEVs, are not also carriers for Rab11a.'*

Response: Based on the very helpful suggestions of this referee below, we provide substantive further evidence that Rab11a is present on a novel form of exosome.

- B.** *'The authors should note that there is evidence that Rab11a is involved in secretion of canonical CD63-positive exosomes (van Niel et al. 2018, Nat Rev Mol Cell Biol).'*

Response: We agree that there is a published literature supporting the role of Rab11a in exosome secretion and did allude to this in the original manuscript (Savina *et al.*, 2002; Koles *et al.*, 2012; Beckett *et al.*, 2013). We are grateful to **ref#3** for highlighting further evidence, reviewed in van Niel *et al.*, 2018. This review has been added to the revised manuscript (**line 521**) together with another reference, which more specifically relates to Rab11's role in CD63-positive exosome release using breast cancer cells (Messenger *et al.*, 2018; **line 521**). Indeed, *Rab11a* knockdown in our manuscript (old **Fig 5C**, new **Fig 6C**) indicates that the levels of both CD63 and CD81 secreted from HCT116 cells may be reduced, but not strongly, in normal and glutamine-depleted conditions.

New citation: van Niel G, D'Angelo G, Raposo G (2018) Shedding light on the cell biology of extracellular vesicles. *Nat Rev Mol Cell Biol* 19: 213-228.

New citation: Messenger SW, Woo SS, Sun Z, Martin TFJ (2018) A Ca²⁺-stimulated exosome release pathway in cancer cells is regulated by Munc13-4. *J Cell Biol* 217: 2877-2890.

Major concerns of ref#3

1. *'For the authors to claim that they have discovered a novel type of distinct Rab11a-positive exosomes, they must do definitive and comprehensive characterization of this new type of vesicle using a variety of different techniques and must use a validated Rab11a antibody. A few approaches are listed here.'*

Response: Please see below for new experiments to study the Rab11a-marked exosome subtype. The reviewer has made an important point concerning antibody specificity for Rab11a. In all immunostainings except **Fig 3F**, we have employed a rabbit Rab11a-specific antibody (Cell Signaling). In our hands, this antibody works relatively poorly on western blots (**Appendix Fig S3A**). However, using *Rab11a* knockdown in HCT116 cells, we have shown that a non-isoform-specific antibody predominantly detects Rab11a in western blots of cell lysates (**Appendix Fig S3A**) and EVs (**Fig 6C** [old **Fig 5C**]) from HCT116 cells. Rab11a is frequently the predominant Rab11 protein in tissues. We also show for HeLa cells (**Appendix Fig S3B**) that Rab11a is increased in EVs following glutamine depletion.

2. *'Super resolution 3D SIM imaging can be used to examine whether Rab11 is in the ILV of recycling MVBs rather than simply being present in MVBs. Gradient fractionation could be used to determine which fractions contain Rab11 and other exosomal markers (Kowal et al 2016, PNAS). In addition, magnetic beads conjugated to a CD63 antibody or an antibody against a transmembrane protein found in the putative new population of exosomes could be used to capture these two populations to provide a fuller characterization of the composition of this new population and contrast it to classical exosomes (see Kowal et al 2016, PNAS; Jeppesen et al 2019, Cell). Crucially, the authors must be able to demonstrate that CD63-positive and Rab11-positive exosomes are distinct entities and not simply the same exosomes but with increased or decreased CD63 and Rab11a (and other cargoes) due to glutamine depletion or mTORC inhibition.'*

Response: We respond to the comment on super-resolution microscopy under **ref#3, pt.6** below.

Further characterisation of the Rab11a-positive exosomes using density gradient fractionation proved very informative. We used characterised exosome and microvesicle markers (Jeppesen *et al.*, 2019) to detect the fractions that contain these different vesicles (new **Fig 4C**), which have partially overlapping fractionation patterns in HCT116 cells. Although the Rab11a signal is too weak under normal conditions to make firm conclusions (new **Appendix Fig S6A**), in glutamine-depleted conditions, it primarily co-fractionates with the exosome proteins CD63 and CD81, and lacks the lower density profile of microvesicle marker AnnexinA1 (AnxA1; new **Fig 4C** and **lines 270-277**).

We have also performed immuno-capture of CD63-containing vesicles produced under glutamine-depleted and -replete conditions. In our hands, CD63 only immuno-isolates a small proportion of HCT116 CD63-positive vesicles. To circumvent this issue, we used a large quantity of the captured

material (relative to input) for western blot analysis (new **Fig 4E**, **Appendix Fig S6D** and **lines 289-292**). Under glutamine-depleted conditions, an equivalent small proportion of Rab11a is captured with a CD63 versus control antibody, in contrast to the much higher levels of the classical exosome markers CD81, Syn-1 and Tsg101 observed with CD63 immunocapture (new **Fig 4E**).

CD81 has been reported to mark a larger proportion of exosomes than CD63 (Kowal *et al.*, 2016). Whether the additional CD81-positive exosomes originate from the recycling endosomes, which may contain some CD81 (imaging in new **Appendix Fig S4C**, **S4D**), remains unclear. With the CD81 pull-down, only a small fraction of vesicular Rab11a is captured (new **Fig 4F**, **lines 292-5**), while no CD63, Syn-1, etc remains in the supernatant, supporting the idea that CD63- and Rab11a-positive exosomes are largely distinct.

We have not yet identified a specific EV surface marker to immunocapture a substantial proportion of Rab11a-positive exosomes or all the exosomes from Rab11a-compartments. Anti-tetraspanin antibodies are currently the most effective tools for classical exosome pull-down, perhaps because they are so highly enriched on specific vesicles, and so it is likely to be challenging to develop such a tool for Rab11a-positive exosomes. In new **Fig 8E** (supported by **lines 499-505** of Discussion), we present a schematic explaining our current model, which leaves the question of how many CD81-positive exosomes might form in Rab11a-compartments open. Whatever the balance of different ILVs in these compartments, our protease/detergent and density gradient studies demonstrate that Rab11a is inside vesicles of exosome density, which are formed separately from CD63-positive and most CD81-positive exosomes and therefore outside CD63-positive late endosomes.

New citation: Jeppesen DK, Fenix AM, Franklin JL, Higginbotham JN, Zhang Q, Zimmerman LJ, Liebler DC, Ping J, Liu Q, Evans R, Fissell WH, Patton JG, Rome LH, Burnette DT, Coffey RJ (2019) Reassessment of Exosome Composition. *Cell* 177: 428-445.

3. *'The rationale for the choice of the different cell lines is not clear. There is substantial evidence that mutant KRAS alters cancer cell dependence on glutamine. The HCT116 human colorectal cancer cell line has mutant KRAS. It might be instructive to use isogenic cell lines that differ only in their KRAS status. It is also not clear why HCT116 cells were chosen as recipient cells. Perhaps KRAS wild-type colorectal cancer cells might be a better choice.'*

Response: We selected HCT116 cells for two major reasons. *First*, we had previously characterised their glutamine dependence and its effects on mTORC1 signalling (Fan *et al.*, 2016). Depletion of exogenous glutamine is used to suppress, but not completely inhibit, mTORC1 signalling in a particularly reproducible way throughout this manuscript. *Second*, we knew that in HCT116 cells, secretory and endosomal membranes are clustered in a perinuclear region (Fan *et al.*, 2016). This was helpful in showing that CD63- and LAMP-positive late endosomes and lysosomes are largely distinct from Rab11a-positive compartments (see also **ref#2**, **pt.1-1**; **Fig 3**).

KRAS mutations are common in colorectal cancer (CRC; ~30-40% of all tumours) and cells carrying such mutations have been extensively used for exosome studies previously, for example by Demory-Beckler *et al.* (2013) (new **line 348**). We elected to screen cell lines from a range of different

tumour types, rather than different CRC cell lines, to confirm that the switch in exosome secretion induced by mTORC1 inhibition is not CRC-specific.

We used HCT116 cells as recipient cells so that we could test whether HCT116 cells can intercommunicate under stress conditions via Rab11a-exosomes. We agree that it will be interesting to test the effects on CRC cells without a KRAS mutation in the future, but since we see effects, we do not think this is essential in this current study.

Regarding the referee's comment concerning KRAS's role, our analysis included HeLa cells, which do not have mutant KRAS. These cells still demonstrate a switch in exosome production under glutamine depletion. To further address this comment, we have now included a pharmacological approach to investigate KRAS function in HCT116 cells, treating the cells with a recently identified KRAS inhibitor, BI-2852 (Kessler *et al.*, 2019), which can block the activated form of the molecule. While this inhibits ERK signalling, unlike mTORC1 blockade, it does not affect the secretion of exosome markers, including Rab11a, from HCT116 cells (new **Fig 5F** and **lines 348-352**).

New citation: Demory Beckler M, Higginbotham JN, Franklin JL, Ham AJ, Halvey PJ, Imasuen IE, Whitwell C, Li M, Liebler DC, Coffey RJ (2013) Proteomic analysis of exosomes from mutant KRAS colon cancer cells identifies intercellular transfer of mutant KRAS. *Mol Cell Proteomics* 12: 343-355.

New citation: Kessler, D, Gmachl, M, Mantoulidis, A, Martin, LJ, Zoephel, A, Mayer, M, Gollner, A, Covini, D, Fischer, S., Gerstberger, T, *et al.* (2019) Drugging an undruggable pocket on KRAS. *PNAS* 116: 15823-15829.

4. *'The manuscript has validated limited cargos by western blot under glutamine-depleted conditions. Global proteomic analysis may lead to identification of additional cargos differentially altered due to glutamine depletion or mTORC1 inhibition.'*

Response: We agree that it would be very helpful to use proteomic analysis to characterise the content of Rab11a-exosomes. However, we do not currently have an immunocapture method to isolate them, so we could only analyse mixed EV preparations (perhaps depleted of CD81-positive exosomes, which removes only about 10% of particles) from glutamine-depleted cells. Defining which proteins are located on Rab11a-exosomes and then developing antibody tools to isolate these exosomes is likely to take some significant time and we believe this is outside the scope of the current manuscript.

5. *'Authors state, "However, there was an increase in Rab11a and Cav-1 relative to CD81 (Figure 4F, I), consistent with induction of an mTORC1-regulated switch in the balance of exosome secretion from late endosomal to Rab11a-compartments." Because this treatment also reduces the number of total EVs released it may not be a switch in secretion but a loss of the late endosomal pathway with the "Rab11a" pathway simply being maintained. One way to resolve this issue that is also raised in the model proposed in Figure 7E, is to determine whether CD63 knockdown affects Rab11-derived EVs with and without glutamine depletion.'*

Response: This referee raises a very important point, which relates to the approach by which we analyse the switch in exosome production under glutamine depletion, which is considered in detail below (**ref#3, pt.8**). Based on the arguments presented therein, we have more extensively discussed the results in old **Fig 4F** (new **Fig 5D; lines 332-336**) and for other treatments.

Importantly, we did originally argue that there was a change in the balance of Rab11a to CD81 secretion after *raptor* knockdown and we present further analysis to show this is the case. However, **ref#3** is correct to flag that levels of Rab11a secreted per cell are essentially unchanged.

Since CD63 does not mark all tetraspanin-positive exosomes (Kowal et al., 2016), we do not think that a CD63 knockdown will confirm that the Rab11a pathway is independent of the late endosomal pathway. Indeed, it has been reported that *CD63* knockdown is not as effective at removing secreted exosome markers as other exosome-inhibitory manipulations (Baietti et al., 2012). We believe that the *Rab7* knockdown experiment in our original manuscript (old **Fig 5F**, new **Fig 6F**), where levels of all classical exosome markers, but not Rab11a, are reduced, provides good evidence that the switch in the balance of exosomes is not driven by secretion from late endosomes (**lines 408-414**).

New citation: Baietti MF, Zhang Z, Mortier E, Melchior A, Degeest G, Geeraerts A, Ivarsson Y, Depoortere F, Coomans C, Vermeiren E, et al. (2012) Syndecan-syntenin-ALIX regulates the biogenesis of exosomes. *Nat Cell Biol* 14: 677-685.

6. *'Figure 1 C stained for Rab11, the authors stated that due to the low fluorescence intensity relative to CD63-GFP, they could not image YFP-Rab11 using super-resolution microscopy (Figure 1C). However, for Figure S1D, the authors use wide field to image 'Rab11a' when YFP-Rab11a was overexpressed? Without high resolution imaging, the author cannot conclude that Rab11 is inside of Rab11-positive ILV only based on the presence of Rab11a inside the MVBs. The same argument holds for Btl. The authors state that "Rab11 and Btl are selective membrane-associated markers for exosomes generated in Rab11-compartments of SCs, which we describe as "Rab-11-exosomes". The data presented, in our judgement, do not support this conclusion.'*

Response: Regarding the referee's comment on Btl-GFP, which marks one of the two transmembrane FGFRs in *Drosophila*, we did analyse localisation of this marker by 3D-SIM in **Fig 1E, Zoom** (now **Fig 1F, Zoom**) and showed this protein is located at the surface of intraluminal vesicles inside the large non-acidic secondary cell compartments. We have previously shown, and also demonstrate in **Fig 1D**, that these large non-acidic compartments are Rab11-positive.

As stated in the original manuscript, in flies carrying the *YFP-Rab11* gene trap, Rab11 is expressed at endogenous levels, which are not high enough to image by 3D-SIM. We have found that 'overexpression' of YFP-Rab11 using the GAL4-UAS system does not lead to higher Rab11 expression levels, so this is also incompatible with 3D-SIM; there must be some post-transcriptional regulation of Rab11 that restricts overall protein levels. We did, however, provide evidence in the original manuscript that *ESCRT* knockdown blocks the formation of Rab11 puncta inside Rab11-positive compartments, consistent with the idea that is associated with ILVs (old **Fig S3B** and **S3C**; new **Fig EV2**). In addition, in human cells, we had shown that Rab11a is shielded by membranes from proteases in EV preparations (old **Fig 4H**, new **Fig 4D**). Taken together these findings support our proposal that Rab11 is inside ILVs. In fact, if Rab11 is not in vesicles within SC compartments, it must either be transported through a membrane from the cytosol (via a currently uncharacterised mechanism) or be released from a degraded vesicle.

To further address the referee's point for Rab11 in flies, we have presented data from two additional experiments. First, we have co-expressed the *Rab11* gene trap with a UAS-CD63-mCherry construct we have generated. We show that CD63-mCherry traffics at very high levels to the acidic LEL structures in SCs, a common observation for red fluorescent proteins in these cells. It is, however, also found at much lower levels at the surface of some non-acidic compartments, which seems to result in exclusion of YFP-Rab11 at the limiting membrane. Importantly, although CD63 and Rab11 are often observed separately inside these compartments, there is co-localisation of CD63-mCherry and YFP-Rab11 in some internal puncta (new **Fig EV1J**, lines **127-134**), which at this level of resolution, is the best additional imaging evidence we can provide for vesicular association of Rab11. We have also included a rare example of a cell in which very large vesicles have formed inside a Rab11 compartment, where it is clear that YFP-Rab11 is associated with the vesicle membranes (new **Fig EV1F**, lines **120-122**).

7. *'There are no high-resolution images to show Rab11 localization when the cells are exposed to glutamine depletion or mTORC1 pathway inhibition. How did the author ascertain that Rab11 only localizes to recycling exosomes under these conditions? How did the authors know that the sEV pellets obtained by UC consist purely of "Rab11-recycling exosomes"? As mentioned above, in order to claim a separate Rab-11 recycling exosome, they should combine beads capture, gradient purification and high-resolution images to further characterize these Rab-11-containing compartments and EVs.'*

Response: We thank the reviewer for these helpful comments. Bead capture and gradient purification experiments are discussed in **ref#3, pt.2**. We had previously undertaken confocal imaging of HCT116 cells under glutamine-replete conditions and have now repeated this work and included analysis of glutamine-depleted cells (new **Figs 3C, 3D**). We have also assessed Rab11a and CD81 staining under the two conditions (see **ref#1, pt.1**). These data indicate that following glutamine depletion, Rab11a continues to predominantly localise to perinuclear compartments distinct from CD63-positive compartments (**lines 262-5, 292-295**), consistent with our analysis of isolated EVs.

Regarding the point about sEV pellets consisting entirely of Rab11a-exosomes, we do not believe that we suggested that UC or SEC preparations of EVs isolated under glutamine-depleted conditions are exclusively Rab11a-exosomes. In fact, our western analysis data show that other vesicles are present, eg CD63-positive vesicles and Cav-1-positive vesicles. However, the preparations are enriched for these exosomes compared to EV preparations from glutamine-replete cells. Purifying novel populations of EVs is a significant issue in the EV field. Until an EV subtype-specific marker can be efficiently pulled down, the 'purity' of any EV preparation cannot be determined, cf. the analysis of non-exosome vesicle subtypes in Jeppesen *et al.*, 2019.

8. *'Different ways of normalizing proteins in various figures is confusing. In some cases, proteins are normalized to total cell lysate proteins. On Page 33, authors state "we elected to analyse EVs based on EV-secreting cell mass to ensure that any changes in protein levels did not result from testing EVs produced by altered number of cells." What does this sentence mean? The author needs to explain the normalization in detail and why this makes sense. Also, for all the EV western blot quantification, levels were normalized to CD81. Why is CD81 chosen to be the standard for normalization? In general, protein levels can be normalized to*

total EV number or total protein level in EVs. Cell lysate normalization is an apples to oranges comparison. CD81 presence is associated with subtypes of EV in a heterogeneous mixture of EVs and can't be used as a normalization control. Ideally, a normalization control would be a common protein used in all EV biogenesis; due to the heterogeneous nature of EVs this is hard to determine.'

Response: This is an important and very difficult point to address fully. As the referee indicates, there is no protein known to be common to all EVs. For western analysis, it is, therefore, not possible to normalise EVs in a similar fashion to cell lysates, where housekeeping proteins, such as actin or tubulin, can be used.

We have elected *not* to use EV number or EV protein content as a means of normalising EV loading for western analysis for three reasons:

First, we find that the measurement of EV number by NTA in repeats under the same EV collection conditions is more variable on a per cell basis than the other measures that we use for normalisation. We believe that this reflects limitations with the technology, which effectively involves a particle count rather than an EV count, using an approach based on Brownian motion. To illustrate this and partly address the reviewer's point, we have included a normalisation with EV number for glutamine-depleted HCT116 cells in **Fig EV3E**. It requires a larger number of samples to reach significance. This analysis suggests that the number of CD81- and Syn-1-labelled exosomes is reduced by about 30-40% within the mixture of particles, but not as much as the CD63-labelled exosomes, and that Rab11a-labelled exosomes are increased by nearly three-fold. In some other experiments, particularly the knockdown studies, there is even more variation in particle counts, sometimes prohibiting their use for normalisation.

Second, we believe particle counts and EV protein levels are not preferable normalisation controls for our experiments. EV preparations also contain protein aggregates (Jepperson et al., 2019; **Fig 4C**), and at least in the case of UC, some soluble proteins can be present. In our hands, this means that particle numbers and protein levels in EV preparations do not correlate well either with each other or with EV marker signals on westerns. As an illustration, when we perform anti-CD81 exosome pull-down experiments, the EV particle number only reduces by about 10%; we believe that at least some of the remaining particles are likely to be non-vesicular, in addition to the Rab11a-marked exosomes and other vesicles in the preparation.

Third, in the example given by **ref#3, pt.5** for *raptor* knockdown, using EV number affects the conclusion in the same way as using CD81 for normalisation (new **Appendix Fig S5F**), indicating an increase in Rab11a and Cav-1 secretion, when the levels secreted per cell are not sufficiently altered to produce a significant increase. In fact, cell lysate protein mass is the only variable that can be used for normalisation to highlight this point.

Importantly, several previous studies have used cell number or protein content of EV-secreting cells to determine the loading of EV proteins for western analysis, especially when comparing secretion from the same cell line. These include: (1) cell number (Ghossoub *et al.*, 2014) Zimmerman lab, KU Leuven, Belgium, (2) protein content in cell lysates, after knockdown of exosome regulators (Baietti *et al.*, 2012) Zimmerman and David labs, KU Leuven, Belgium and (3) cell number to compare the effects of serum starvation on MDA-MB-231 cells (Kowal *et al.*, 2016) Théry lab, Institut Curie, France.

Their rationale and ours is that loading gels on this basis is the most appropriate way of testing whether a specific exosome marker is secreted at different 'per cell' levels under different conditions. In most of our analyses, we consistently see an increase in Rab11a and Cav-1 levels following glutamine depletion or Akt/mTORC1 inhibition, and usually a decrease in CD63 (see new **Fig 5G**), though as we highlighted in the original manuscript (and discussed and investigated further in **ref#1, pt.2**), this is probably partly due to a reduction in CD63 expression in cells.

We selected CD81 as a normalisation control protein, because it seemed to be the best general marker for exosomes and would therefore reflect changes in the balance of Rab11a and CD63 secretion. However, our pull-down experiments suggest most Rab11a-positive exosomes are not labelled with CD81, and so CD81 levels may well also drop if the balance is altered under glutamine-depletion, etc. We have therefore responded to **ref#3, pt.8** in three ways:

- 1) included a normalisation to EV number in new **Fig EV3E** to show the greater variability in this measurement and confirm a Rab11a/Cav-1 increase and CD63 decrease for glutamine-depleted HCT116 cells (**lines 241-247**). We have also included a similar analysis for *raptor* knockdown in these cells (**Appendix Fig S5F**);
- 2) used the absolute intensity values from the western blots of EV proteins, loaded according to cell lysate protein mass, to determine the change in secreted levels of each exosome marker for all our analyses on a 'per unit cell lysate protein mass' basis;
- 3) included all the CD81 normalisation data in **Appendix Fig S5**, since it does provide an indication of the increase in Rab11a/Cav-1 and decrease in CD63 relative to a 'classical' general exosome marker;
- 4) discussed these points in **lines 238-256**, explained our approach in **lines 917-926**, and removed the sentence highlighted by **ref#3**.

New citation: Ghossoub R, Lembo F, Rubio A, Gaillard CB, Bouchet J, Vitale N, Slavík J, Machala M, Zimmermann P (2014) Syntenin-ALIX exosome biogenesis and budding into multivesicular bodies are controlled by ARF6 and PLD2. *Nat Comm* 5: 3477.

9. *'Different non-comparable EV isolation methods are used (some use SE, others UC method). Each method has to have purified fractions fully characterized to allow comparison. Without a common method of isolation, such differently purified EVs are not comparable. For instance, in Figure 4 E and G, Torin1 and AKT inhibitor AZD5363 treatment, EVs are isolated using SEC method. However, for Figure 4 F, in the Raptor knockdown experiment, EVs were isolated by UC. This is also the case for Figure 5D' and 5E'.*

Response: The referee is correct that in old **Figs 4B, 4C, 4F, 5D'** and **5E'** (new **Figs 4B, 5A, 5D, EV4A', EV4B'**), we showed western analysis of EV preparations isolated using UC. Except for **Fig 4B**, we did not show repeats using size-exclusion chromatography (SEC). All other western blot data in which we characterise the cargos of EVs produced under 'stress' versus normal culture conditions employed EV preparations isolated by SEC, which we have found to be a more consistent method for EV isolation.

Importantly, in the original manuscript, for the HCT116 glutamine depletion condition that we used throughout the study, we showed UC and SEC data and confirmed that essentially equivalent changes were observed, eg old **Fig 4B** and **Fig S4F** (new **Fig 4B** and **Fig EV3F**) for EV cargos, and old **Fig 5A** (new

Fig 6A) for EV growth-promoting function. Particularly because we wanted to assess EV function, we were concerned that using a single isolation method might produce effects on recipient cells that were not directly linked to vesicles, and reasoned that using two very different isolation approaches would control for this.

To consolidate the core messages in the manuscript, we have also now included an analysis of AREG in HCT116 EVs using UC to confirm that this is increased following glutamine depletion (**Appendix Fig S8D** versus SEC data in **Fig 7C**) and show the proteinase K digestion data for SEC (**Appendix Fig S6C**) in addition to UC (**Fig 4D**).

For the glutamine depletion of HeLa cells (old **Fig 4C**) and HCT116 rapamycin treatment (old **Fig S5E'**), we have repeated the experiment using SEC and included these data (new **Appendix Figs S7A", S7E**). Since Torin1, like rapamycin, strongly suppressed EV production in HCT116 cells (old **Fig 5D'**), and did not therefore reveal increased secretion of Rab11a, we have not repeated this experiment with SEC. We also did not think a triplicate analysis of the *raptor* knockdown using SEC would add significantly to our findings, since we have already shown Akt inhibition and mTORC1-inhibitory glutamine depletion in HCT116 cells produce a switch to more Rab11a secretion using SEC.

Overall, we think that our use of both UC and SEC is a strength of the study, given that we have shown that both isolation methods produce a similar profile of the key markers we are studying and we have confirmed functional parallels (briefly discussed in **lines 257-62**). In summary, the manuscript now contains SEC data for all 'stress' treatments except Torin1 addition and *raptor* knockdown in HCT116 cells. Importantly, we have used both isolation methods to confirm the central findings that glutamine depletion induces a switch in exosome production in HCT116 cells, leading to the production of growth-promoting, AREG-dependent exosomes.

- 10.** *'Different glutamine concentrations were used for glutamine depletion in different cells, from 0.15mM to 0.02 mM or 0. Why? Different cells may respond to different levels of Gln depletion but there needs to be some metric for comparison or a discussion of why these are comparable.'*

Response: This point is well taken. We undertook the HCT116 uptake analysis experiment (old **Fig 5D, 5E**) at an early stage of our study with 0 mM glutamine. We have now repeated this experiment with 0.15 mM glutamine, which gives a similar outcome (new **Fig 6D, 6E**).

For different cell lines, we selected an appropriate glutamine concentration using a dose-response analysis to identify a concentration at which mTORC1 signalling was not strongly inhibited, but 4E-BP1 phosphorylation was clearly reduced over 24 h, in order to mirror the conditions used for exosome analysis in HCT116 cells. We have now explained this in the Materials and Methods section (**lines 820-823**).

- 11.** *'All Western blots must have molecular weight markers to allow the reader to assess if the correct protein is present.'*

Response: We agree that this is helpful and have added the marker sizes.

- 12.** *'In Figure 6C, there is a Western for AREG. There are several isoforms of AREG differing in size in EVs. One can't tell which isoforms are present in this Western blot. Multiple publications have shown that AREG in EVs is very potent and these should be cited. The authors should consider looking at the effect of AREG*

neutralizing antibody on p-EGFR in recipient cells. What population of exosomes produces AREG – classical exosomes or their putative new class of exosomes? Authors might consider quantitating the amount of AREG per EV and do a concentration curve when adding these to recipient cells.'

Response: Thank you for highlighting this literature on AREG and its potency in EVs. We did cite Higginbotham *et al.* (2011) and have now also added Zhang *et al.* (2019), which shows that AREG on exosomes and exomeres is extremely potent.

We have included size marker positions for the AREG blot and in new **Appendix Fig S8D**, we have included a blot spanning the size range of different AREG isoforms, which demonstrates that the membrane-associated form of ~26 kDa (Brown *et al.*, 1998) predominates. We had cited this paper in the original manuscript, but without stating the size of the AREG band, so we now do this (**lines 428-431**).

Currently we cannot demonstrate whether AREG is carried on exosomes from the Rab11a compartments of HCT116 cells, because we do not have a surface marker that could be used to selectively pull down this exosome subtype.

In response to **ref#3's** helpful comment concerning AREG concentration, we have now measured the concentration of AREG by ELISA in the EV preparations from glutamine-depleted HCT116 cells. We had already performed a dose-response analysis with EVs in the manuscript (now Figs **6A, EV5C**) and use this to calculate the minimum active AREG concentration (**lines 435-438**). This equates to approximately 2 AREG molecules per EV, although we have not included this number, since we do not know what proportion of the EVs carry AREG.

New citation: Zhang Q, Higginbotham JN, Jeppesen DK, Yang YP, Li W, McKinley ET, Graves-Deal R, Ping J, Britain CM, Dorsett KA, Hartman CL, Ford DA, Allen RM, Vickers KC, Liu Q, Franklin JL, Bellis SL, Coffey RJ (2019) Transfer of Functional Cargo in Exomeres. *Cell Rep* 27: 940-954.e6.

- 13.** *'For Figure 4 H, it is surprising that CD63 would not be digested by both proteinase K and Triton X-100 treatment while another tetraspanin, CD81, appears to be digested. The authors need to provide an explanation for their findings.'*

Response: Yes, we agree that some more details concerning why CD63 might not be digested would be helpful to include. In the text we now attribute this to CD63 interacting with lipids or failing to unfold in a way that prevents access to proteases, even in the presence of low concentrations of detergent. Furthermore, we confirm this, by repeating the digestion in the presence of the detergent LDS, which presumably disrupts these interactions sufficiently to permit proteolytic digestion of CD63 (new **Appendix Fig S6B; lines 284-287**).

- 14.** *The blots of Figure 4E, 4F and 4G clearly showed that Torin1 treatment or raptor knockdown or AKT inhibitor treatment did not change the Rab11 levels in those EVs. However, the authors conclude that Rab11a level strongly increased under these conditions when the cargo levels are normalized to CD81. This comparison is not an appropriate way to normalize the level of cargos. Equal amounts of proteins or equal number of vesicles should be loaded for each sample compared.*

Response: We have discussed the issues around gel loading above and our decision to load EV preparations, so that we normalise to the protein mass of cells producing them (**ref#3, pt.8**). The effect on Rab11a on these blots following different cell treatments (LNCaP with Torin1, HCT116 *raptor* knockdown and HCT116 with AZD5363) is less pronounced than under some other conditions, but as explained above, except for *raptor* knockdown, levels of secreted Rab11a increase in triplicate experiments when normalised to cell lysate protein levels (new **Fig 5G**), unlike 'classical' exosome markers. We agree that normalising to CD81 accentuates this effect, and therefore only now include CD81 normalisation in new **Appendix Fig S5**. Furthermore, if we normalise the levels of Rab11a to the mean EV number in each case, Rab11a is increased in all three conditions, including *raptor* knockdown (**Appendix Fig S5F**).

In summary, we have shown an increase in Rab11a secretion per unit mass of cell lysate protein under all conditions except LNCaP glutamine depletion and *raptor* knockdown, which we flag up in the text (**lines 305-313, 332-337**). Importantly, the ratio of Rab11a:CD81 increases in all conditions tested, consistent with a change in the balance of Rab11a-positive to CD81/CD63-positive exosomes.

Minor concerns of ref#3

1. *'Some of the blots are not publication quality. For example, figure S5D, CD63 and 4E-BP1; Figure S6D, for S6; Figure 6A.'*

Response: We agree that these blots needed to be improved. We have replaced **Fig 6A** (now **Fig 7A**) and also the blots in **Figs S5D** and **S6D** (now **Figs EV4A** and **EV5E**).

2. *'Figure S5 F, based on the western blot, it is hard to tell that CD63 levels are strongly reduced when Raptor was knocked down even though the quantification reflected this. Quantification from three separate biological replicates is needed.'*

Response: Wherever bar charts are shown for westerns in the manuscript, including in **Fig S5F** (now **Fig EV4C**), all data were from at least three biological replicates. CD63 levels are reduced by approximately 25% in lysates after *raptor* knockdown. In the manuscript text, we had described CD63 levels as 'decreased', not 'strongly reduced'.

3. *'Figure 6B, the last two bars appear to be incorrectly labeled.'*

Response: We appreciate your alerting us to this and have now corrected this (new **Fig 7B**), in addition to changing the format of the graph (as suggested by **ref#2 pt.min11**).

4. *'For Figure S3D, E and F, why are only the bar charts shown? The imaging associated with these should be shown.'*

This is now added as a new supplementary figure (new **Appendix Fig S2**) and we have included the bar chart data for the analysis of Rab11 in SC *ESCRT* knockdowns (new **Fig EV2E, EV2F**).

5. *'Figure 7C, Y-axis is not labeled. What does fold change in confluency mean? Can the authors provide an explanation for why the Gln 0.15m M AREG Ab treatment has the highest fold change?'*

Response: We think this comment refers to old **Fig 6C'**. If so, the Y-axis was labelled 'fold change in confluency'. To improve clarity, we have now explained this phrase in the Materials and Methods (**lines 998-1002**) and when we first use it in new **Fig 6A**.

The greatest fold change in what is now **Fig 7C'** is with Gln 0.15 mM and no Ab (solid red line) rather than the antibody-treated equivalent (dashed red line), which follows the control growth curve. We have highlighted this in the figure legend.

6. *'Figure S4E has a bar chart showing changes in levels of putative exosome proteins in EVs purified by UC. What is the Western blot corresponding to in this bar chart? As noted above, we don't think cell lysates serve as the appropriate control. normalized to cell lysates?'*

Response: This bar chart corresponds to the triplicate EV data for which one blot is presented in **Fig 4B**, which we explained in the figure legend. As discussed above, we have now included an extra graph based on normalisation to EV number (**Fig EV3E**), using a larger number of samples, and this essentially shows equivalent changes for the key markers (CD63, Rab11a, Cav-1) when compared to normalisation using cell lysate (see **ref#3, pt.8** above).

7. *'For Figure S5E, which method was used for EV isolation?'*

Response: UC was used here, which we had described in the figure legend, but we have now included this in the figure panel for both **Figs S5D** and **S5E** (new **Figs EV4A', EV4B'**).

8. *'For Figure S5G, when the AKT inhibitor AZD5363 was added, what is the level of phospho-AKT?'*

Response: We did not show a phospho-Akt blot, because it proved difficult for us to reliably detect phospho-Akt in HCT116 cells. We did assess the phosphorylation of Akt's downstream target PRAS40 (old **Fig S5G**; new **Fig EV4D**) to show that the drug was working. We are not sure whether **ref#3** is interested to see the effect of AZD5363 on Akt, because it has previously been reported to increase Akt phosphorylation (Zhang *et al.*, 2016), perhaps because of the loss of a negative feedback pathway. We do not think that this point is directly relevant to the theme of the manuscript, focusing on signalling downstream of Akt.

Relevant reference: Zhang Y, Zheng Y, Faheem A, Sun T, Li C, Li Z, Zhao D, Wu C, Liu J (2016) A novel AKT inhibitor, AZD5363, inhibits phosphorylation of AKT downstream molecules, and activates phosphorylation of mTOR and SMG-1 dependent on the liver cancer cell type. *Oncol Lett* 11: 1685-1692.

9. *'For figure S5E, why are the levels of phospho-S6 and 4E-Bp1 not examined following exposure to rapamycin?'*

Response: They were in the western and bar chart in **Fig S5E** (new **Fig EV4B**), which shows a major inhibition of P-S6, also presented in **Fig EV4E**.

Referee #1 (ref#1) - minor comments:

1. *'can the authors really exclude the possibility that part or all of the Rab11/Cav1/AREG-containing EVs could form by direct budding at the plasma membrane, rather than inside internal recycling endosomes? Additional IF images of HCT116 cells could possibly be shown in figure 3, displaying localization of Rab11a in the plasma membrane area, with CD81 as a marker of this area, and possibly AREG.'*

Response: We do not think we can completely exclude the possibility that some of Rab11a-containing vesicles are formed from the cell surface. We have, however, undertaken further experiments based on **ref#1**'s suggestions, as well as more detailed characterisation of these vesicles (eg, density gradient fractionation; see **ref#3, pt.2** above), to strengthen our argument that Rab11a is packaged into exosomes.

We have included images of co-staining with anti-CD81 and Rab11a antibodies under glutamine-replete and -depleted conditions (**Appendix Fig S4C; lines 266-270**). These show some co-localisation in the clustered perinuclear recycling endosomal region, but at the cell surface, while CD81 is strongly surface-localised (as suggested by the reviewer), low-level Rab11a staining is primarily localised in puncta below the plasma membrane. There is very little co-localisation at the plasma membrane, supporting our argument that Rab11a is not a microvesicle marker. However, it is worth noting that CD81 is considered to be one of the best exosome-specific markers, despite its abundance at the cell surface, so we have not made a strong case for Rab11a as an exosome marker, based on its localisation below the plasma membrane of HCT116 cells.

- a. *'Of note, CD81 is not an endosomal marker, as wrongly suggested in the introduction p4, it is in most cells expressed at the PM or possibly sub-PM compartments.'*

Response: We thank the reviewer for pointing out this error, now corrected (**line 69**).

- b. *'the quality of IF images in figure 3 is not very satisfying.'*

Response: We have repeated the stainings in old **Figs 3A-C**, and also included a Rab11a/CD63 co-staining in glutamine-depleted conditions in response to **ref#3, pt.7** and **ref#2, pt.1-1** (new **Figs 3A-D**).

2. *'the decrease of global CD63 expression in the cells upon glutamine deprivation is not really discussed or explained: is CD63 degraded in lysosomes? is it localized in different intracellular compartments than in control cells? This decrease makes it difficult to interpret the anti-CD63 uptake experiment shown in figure 5D: is uptake really compromised, or is it impossible to do the experiment in the glutamine-deprivation condition because the Ab will not bind any CD63 at the cell surface?'*

Response: We agree that this is an important point to clarify. To determine whether lysosomal degradation is increased in glutamine-depleted conditions, we have treated cells under these and normal conditions with chloroquine to buffer the protons inside the late endosomes and lysosomes. This is now shown in **Appendix Fig S4B (lines 217-220)**, after we first show that CD63 levels are reduced in cell lysates in **Fig 4A**, and then highlighted again in the text accompanying the uptake experiment in new **Fig 6D, 6E (lines 399-402)**.

3. *'The first two figures showing intracellular compartments in the Drosophila accessory gland are interesting and nice in suggesting the existence of the Rab11+ compartments with internal vesicles and of Rab11+ EVs and Btl secretion, but maybe a bit difficult to follow for someone who is not used to look at images of this experimental model.*

For instance, Figure S1 may be more informative than current main figure 1.'

Response: We think that we need to retain the images shown in old **Fig 1**, because imaging CD63-GFP and Btl-GFP with 3D-SIM provides key evidence for the presence of vesicles in the non-acidic Rab11-compartments. This figure also presents the clearest data showing Rab11-positive puncta inside these same compartments in the *YFP-Rab11* gene trap line. In response to the reviewer's point, we have moved the first image in old **Fig S1A** into the main figures (now **Fig 1B**) to give a clearer view of the SC with DIC, which shows the dense cores in the Rab11-compartments. It also illustrates the extra resolution provided by 3D-SIM.

- a. *'The images do not show the plasma membrane pattern of the various molecules in secondary cells, thus again, do not make it possible to determine whether PM could contribute to the EVs found in the accessory gland lumen.'*

This was shown in old **Fig S1B** and **S1C** (now **Fig EV1G, EV1H**) – as we discussed in the original manuscript, it is only YFP-Rab11 that does not have clear plasma membrane staining. The presence of an exosome membrane marker at the cell surface does raise the possibility that it could be secreted by plasma membrane budding, but exosome markers do localise to the cell surface (**ref#1, pt.1**), and we now show that secretion of all the exosome markers from SCs is inhibited by knockdown of an ESCRT-0, as well as an ESCRT-I and ESCRT-III (see below).

- b. *'Of note, ESCRT-I and -III are known to be also involved in budding and release of EVs from the PM.'*

Finding a genetic manipulation that only blocks exosome formation is one of the challenges in the exosome field. We had previously shown that CD63-GFP secretion is inhibited by knockdown of a range of exosome-regulatory Rabs (including Rab11) and ESCRTs (Corrigan *et al.*, 2014), providing further support for this marker being secreted on exosomes. We have now also included the data from knockdown of an ESCRT-0, *Stam*, which is not thought to be involved in regulating shed microvesicles, because its primary role is in the sequestration of ubiquitinated cargos to endosomes (McCullough *et al.*, 2013). These data, which again show that ILV biogenesis is blocked, are presented in **Figs 2B, EV2B** and **Appendix Fig S2B**, and accompanying graphs), and discussed in **lines 166-9, 492-495**.

New citation: McCullough, J, Colf, LA, and Sundquist, WI (2013) Membrane fission reactions of the mammalian ESCRT pathway. *Annu Rev Biochem* 82: 663-692

- a. *'It is surprising to see such heterogeneity in the LysoTrackerRed pattern in the different panels: can the authors explain?'*

We have previously shown that large acidic compartments sporadically fuse to each other and with non-acidic compartments and this does lead to variability in their size (Corrigan *et al.*, 2014). The

increased size of acidic compartments in CD63-GFP-expressing cells has been highlighted previously (Corrigan *et al.*, 2014, Redhai *et al.*, 2016), a point that we have now mentioned (**lines 114-116**); we believe it is due to increased late endosomal trafficking. *Stam* knockdown also leads to the formation of large acidic compartments, as previously reported for ESCRT-0 *Hrs* knockdown (see Corrigan *et al.*, 2013), presumably because maturation of these compartments is inhibited and degradation of contents is therefore suppressed (**Figs 2B and EV2B; Appendix Fig S2B**), whereas loss of other ESCRTs appears to block acidic compartment formation.

- b. *'In figure 1E, it would have been interesting to show if Btl and Rab11 colocalize, at the PM or in internal compartments, to strengthen the message that Btl+ EVs come from this pathway.'*

We do not have a Btl transgene labelled with a fluorescent protein other than GFP, which expresses at high enough levels to undertake these experiments. Since all the dense-core granule compartments in SCs carry Rab11 (Redhai *et al.*, 2016), we expect Btl-GFP to co-localise with Rab11 at the limiting membrane of large non-acidic compartments.

To address this point further, we have made a UAS-CD63-mCherry construct and expressed this with the *YFP-Rab11* gene trap. This does affect the identity of some non-acidic compartments in the cell, but they still contain internal fluorescent puncta produced from both transgenes. The results discussed in **ref#3, pt.6** above and suggest that the two markers sometimes co-localise in regions containing ILVs (though we cannot use 3D-SIM to assess this at the single vesicle level), supporting the argument that Rab11 does traffic to ILVs, but indicating that the EVs in individual compartments are not all identical (**lines 127-134**).

- c. *'Finally, for the link with the rest of the article, is there any glutamine-deprivation situation in the accessory gland of drosophila?'*

We have not been able to show an enhancement of secretion from Rab11 compartments using mTORC1 signalling inhibition or 4E-BP overexpression, which are the simplest manipulations to test this idea. However, as we had mentioned in the Discussion (now **lines 558-568**), it appears that secretion from Rab11-compartments is high, even under physiological conditions in secondary cells, and we speculate this may be because of their high secretory activity.

Referee #2 (ref#2):

1. *'What is the relationship between CG63+ and Rab11+ exosomes. Are they the same particles or different particles?'*

Response: Please see below and also the discussion in **ref#3, pt.2** and **ref#3 pt.7** above.

- 1.1 *'In Fig 4D comparing the two glutamine conditions, there are equal levels of Syn-1, Tsg101 and CD81 hence presumably the same number of exosomes particles in the two samples. In the low-glutamine sample there is more Rab11a, but not less CD63 compared to the high-glutamine sample. This*

does not fit the simple model that there are more Rab11a+ exosome particles and less CD63+ particles. Some possible options are that in the low-glutamine

-there is more Rab11a protein per Rab11a+ exosome (ie not more Rab11a+ particles, but more Rab11a protein per particle)

-Rab11a now also gets loaded on CD63+ particles

-Rab11a+ particles also contain CD63

Which of these scenarios is the explanation? One way to look at this would be to IP CD63+ exosomes and check if they contain Rab11a in the two conditions.

Additionally, one could perform microscopy as in Fig 3D to check Rab11/CD63 colocalization, but in the low-glutamine condition.'

These are important and helpful points. In the revised manuscript (new **Figs 4E, 4F**), we now present further evidence that Rab11a and CD63 are on distinct particles, using the IP approach suggested, and indeed, that most Rab11a is not pulled down by CD81. Nevertheless, Rab11a fractionates with exosome markers and not the microvesicle marker AnxA1 on iodixanol density gradients (**Fig 4C**; see **ref#3, pt.2** for further discussion). Therefore, the levels of Rab11a in exosomes can potentially change independently of CD63 and CD81. We did flag up in the text associated with **Fig 4D** (now **Fig 5B**) that CD63 secretion is not altered by glutamine depletion and this happens for other treatments, eg Akt inhibitor treatment of HCT116 cells mentioned below, where the primary change is an increase in Rab11a (and Cav-1) secretion without a reduction in CD63, but still effectively a change in the Rab11a/CD63 balance. Our current model (new **Fig 8E**) is that some CD81 may pass through the recycling endosomes, but the majority is secreted via the late endosomes when CD63 secretion is high, though it may not all be associated with CD63-positive vesicles (Kowal et al., 2016).

In the absence of a surface marker to selectively pull down Rab11a-exosomes, it is not possible to unequivocally show that the same number of vesicles are being loaded with more Rab11a protein. This is also a problem in microvesicle studies. We discuss in **ref#3, pt.8** why total EV particle number is not appropriate for normalisation, mainly because it assesses particles of several types and is more variable than other measures. Therefore, we cannot exclude that Rab11a is loaded into exosomes at higher levels following glutamine depletion or mTORC1 inhibition, though this seems relatively unlikely to be the full explanation, given the increased flux through the recycling endosomal pathway and the fact that *Rab7* knockdown is sufficient to induce increased Rab11a secretion (**lines 510-6**). However, we can say that exosome-associated Rab11a secretion is elevated after glutamine depletion, that this seems to involve the Rab11a recycling endosomal pathway and that the change is accompanied by enhanced activity of the secreted vesicles (**Figs 4, 6, 7**).

We have also followed the helpful suggestion of **ref#2** and undertaken a CD63/Rab11a co-immunostaining in HCT116 cells under glutamine-depleted conditions (**Fig 3D**), as well as including co-staining for Rab11a and CD81 (**Appendix Fig S4; Ref#1 pt.1-1**). This has shown that Rab11a continues to be almost totally excluded from CD63-positive compartments, only partially overlaps with CD81, and is primarily found at low levels in puncta just beneath the plasma membrane (but not on it), as would be expected for a recycling endosomal marker (**lines 262-265, 292-295**).

1.2 *'The same issue arises in Fig 4G upon Akt inhibition. There is more Rab11a but not less CD63.'*

Please see explanation for **ref#2, pt.1-1** above.

1.3 *'Regarding the Drosophila data:*

- a.** *'The Drosophila data seem to indicate the opposite - that CD63 and Rab11 colocalize. Does this suggest they are not mutually exclusive in the fly?'*

For the CD63-GFP experiments in *Drosophila*, we are overexpressing a human transgene in a fly cell, and as we have previously shown (Corrigan *et al.*, 2014; Redhai *et al.*, 2016), while this increases trafficking to the late endosomes and lysosomes, a fraction of the CD63-GFP passes through the Rab11 compartments, which does not seem to happen detectably in human cells. Using a UAS-CD63-mCherry construct, we have now included co-expression data and have observed partial co-localisation (**Fig EV1J**). It appears that if tetraspanins traffic through Rab11 compartments, they will be incorporated into some, but not all, Rab11-positive ILVs, as may also happen for CD81 in human Rab11a-positive exosomes. The idea that CD63 behaves differently in fly SCs was discussed in the original manuscript, and this discussion is now expanded in **lines 505-509**.

- b.** *'The authors claim that the vesicles in the SCs marked by CD63-GFP are Rab11-positive. Co-staining would be necessary to clarify if this is indeed the case.*

-The same is true for Rab11 / Btl colocalization. The authors write "we, therefore, conclude that Rab11 and Btl are selective membrane-associated markers for exosomes generated in Rab11- compartments of SCs, which we describe as 'Rab11-exosomes'." but Rab11 / Btl colocalization is not shown.'

Response: Since we cannot be sure that any of the above markers labels all ILVs in the Rab11 compartments, we did not intend to give the impression that these markers would co-localise in all vesicles. We did point out that only a subset of ILVs in Rab11 compartments appears to be marked by Rab11 in the original manuscript. Since we cannot distinguish YFP-Rab11 from the GFP constructs, which are expressed at much higher levels, using wide-field fluorescence microscopy, and a Btl-mCherry construct that we have analysed does not express at high levels, we needed to construct and express the UAS-CD63-mCherry construct referred to in (a) (see also **ref#3, pt.6**), which shows partial co-localisation with YFP-Rab11. In discussing this result (**lines 127-135**), we explain that Rab11 is only expressed in a small number of ILVs.

- c.** *'Same is true for Shrb and Rab11'*

Response: The Shrub-GFP protein is not detectable inside Rab11 compartments and therefore is not observed in ILVs. It is only at the limiting membrane (now **Fig EV2G**), so we would not expect Rab11 and Shrb co-localisation in ILVs.

1.4 *'Much of the data (e.g. Fig 4E-F) showing that TOR inhibition leads to increased Rab11a and decreased CD63 come from normalizations relative to CD81. Is it possible instead to normalize to EV particle amounts? Does this show the same result?'*

Response: Again, this is an important point. We discuss the general problems concerning normalisation and the issues with using EV number for normalisation in **ref#3, pt.8**. We now also present the data in **Fig 4B** normalised to EV number in **Fig EV3E** to show that this method can be used, but the higher variation in EV counts means that larger numbers of replicates are required. Furthermore, EV number is actually a particle count that detects protein aggregates as well as a range of different vesicles. Nevertheless, if we normalise the levels of Rab11a to the mean EV number in each condition, Rab11a is increased following all treatments; as mentioned above and in the original manuscript, whether CD63 is reduced is dependent on the treatment employed, and even when it is reduced, this may partly reflect increased lysosomal degradation of CD63 (new **Appendix Fig S4B; lines 217-220**). Having now shown that CD81 does not mark the majority of Rab11a-positive exosomes (new **Fig 4F**), we have decided to normalise our data to cell lysate protein mass (a proxy for cell number) in the main figures and present the CD81 normalisation in new **Appendix Fig S5** (see **ref#3, pt.8**).

2. *The link between pERK and increased proliferation caused by the low-glutamine derived exosomes is not completely solid.*
 - a. *-In Fig 6B are the cells still proliferating in the presence of ERK inhibitor? Or does it completely shut down proliferation, in which case it would show an epistatic effect no matter what stimulus is given to the cells?*

Response: This is a very good point, which we should have dealt with in the original manuscript. We have now included the growth curves for this experiment in **Appendix Fig S8C**, which show that the control cells still continue to grow in the presence of ERK inhibitor, explaining why ERKi with control gives the same overall growth as control without inhibitor.

- b. *- The AREGab nicely blunts the proliferative effect of the low-glutamine derived EVs. Does it also blunt the difference between high-glutamine and low-glutamine derived EVs on pErk, as in Fig 6A?*

Response: This is a very helpful suggestion. Yes, it does reduce ERK phosphorylation, as shown in new **Fig 7C" (lines 431-435)**.

Minor issues of ref#2

1. *'Is Rab11 just a cargo/marker for this class of exosomes, or required for Rab11+/Cav-1+ exosomes? Fig 5C shows that glutamine removal increases the amount of Rab11+/Cav-1+ exosomes (both markers increase). Upon Rab11 KD, there is still more Cav-1+ in the EVs. So does this suggest that Rab11 is a marker for this class of exosomes, but not required for their biogenesis?'*

Response: This is an excellent point and one we cannot yet answer fully because we do not have a surface marker to pull down all the exosomes from Rab11a compartments. We do know that *Rab11a* knockdown blocks the growth-promoting activity of glutamine-depletion-induced EV preparations (**Fig 6C'**), suggesting that Rab11a has a role in producing key functional vesicles, not just marking them; it seems very unlikely that Rab11a itself is an essential active cargo in stimulating growth. EV levels of CD81, Syn-1 and Tsg101 seem to be modestly reduced on a per cell basis after Rab11a knockdown in glutamine-depleted conditions (**Fig 6C**), consistent with a reduction in exosomes marked by these molecules, perhaps those vesicles

that pass through the recycling endosomal pathway. By contrast, Cav-1 increases, so we believe that its secretion does not require Rab11a-dependent trafficking. We do not at present know the origin of Cav-1-positive vesicles. The Optiprep gradient suggests that they mainly separate in the same fractions as exosomes, and some Cav-1 is pulled down by CD63 and CD81, but like Rab11a, most of the protein is not present in vesicles marked by these tetraspanins.

2. *'A question for the discussion: In the fly, from the following text: "In contrast, a YFP-Rab7 gene trap fusion protein (Dunst et al., 2015) primarily trafficked to acidic LELs (Figure 1D) and marked very few puncta in the AG lumen."*

it appears there are mainly rab11 exosomes and not rab7 exosomes. In humans, instead, the authors suggest it's the other way around and the Rab11 exosomes are only stress induced.

Is this a difference between the two systems? Or do SCs have some basal stress levels because, perhaps, they are highly secretory?'

Response: We had mentioned that high levels of secretion might be involved in increased Rab11-exosome production in secondary cells under normal physiological conditions, but we have now added that the stress associated with secretion might be an explanation (**lines 566-8**)

3. *'Page 6 - the abbreviation "ILV" should be spelled out the first time it is used.'*

Response: Thank you for pointing this out – we have now added this in **line**

103.

4. *'Fig 4A - the authors conclude that the HCT116 cells "maintain growth factor signaling" when transferred from complete medium to serum free medium supplemented with insulin, however they do not show pS6 or p4EBP levels in cells in complete medium as a comparison. This is needed to claim the cells maintain growth factor signaling. (Otherwise, the levels of pS6 or p4EBP shown in Fig 4A could be only 1% of complete medium, but a long enough exposure of the blot will give a signal.)'*

Response: Sorry, we should have included this. It is now shown in **Appendix Fig S4A.**

5. *'Relating to the torin treatment of HCT116, the authors conclude that "Secretion of all exosome markers was significantly reduced (Figure S5D', S5E'), suggesting a general shut-down in exosome release." Indeed the EV number drops from 0.89 to 0.5, which is a reduction but not a general shut-down. Furthermore, in Fig S5D' one sees that Syn1 and Tsg101 levels in the EV isolation are ok. (Presumably because this is what is being used to normalize the EV loading) But nonetheless, these exosomal markers are present, also suggesting there is not a general shut-down in exosome release. Instead, the exosomal cargo seems to change because CD81, CD63 and Rab11a levels are dropping. I wonder whether the interpretation of the data is correct?'*

Response: **Ref#2** is correct that there is not a complete shut-down, although as discussed in **ref#2, pt.1-1** above, 'EV particle number' is measuring more than EVs; we should have discussed this point more accurately. We have reworded the statement about **Figs S5D', S5E'** (now **Figs EV4A, EV4B**) to reflect the fact that exosome markers are reduced, but not completely lost (**lines 319-325**). Syn-1 and Tsg101 in EVs are reduced on a per cell basis after Torin1 treatment, but less than other exosome markers (**Fig EV4A**). Without other tools to separate exosome subtypes, it is difficult to make any

firm conclusions from this. It may, of course reflect further diversity in exosome populations under these conditions.

6. *'Fig 5a - why are the error bars on the lower graph larger, yet the significance of the difference is *** compared to the * in the upper graph?'*

Response: Sorry, this was miscalculated in the top panel (now **Fig 6A**) and has been changed.

7. *'Fig 5a, upper panel - the lower part of the error bars appears to be missing. Or, if the error bars are only shown in one direction, this inconsistent across panels.'*

Response: Thank you, we have now changed this in new **Fig 6A**.

8. *'X-axis labeling Fig 5B has an error (the 2nd to last set should be without ERKi).'*

Response: Thank you for pointing this out. We have now changed this and altered the style of the bar charts to mirror other figure panels (now **Fig 7B**).

9. *'Fig 4F quantification - data points (dots) on the Rab11a and Cav-1 bars must be missing because all the dots shown are lower than the average.'*

Response: Thank you. This was because we had left a gap along the y axis to include the full span of the data, and this removed the position of some of the data points. The position of the gap has now been changed (now **Appendix Fig S5F**).

10. *'The order in which the panels are referenced in the text does not always match the order in the figures. This makes it a bit difficult to follow.'*

Response: We have looked at this in the revised manuscript, and made a number of changes to address the issue.

11. *'Some main-figure bar graphs are missing the overlaid individual data dots (e.g. Fig 5E, 6A)'*

Thank you, these have now been added, eg **Fig 6E, 7A, 7B**.

2nd Editorial Decision

31st Jan 2020

Thank you for submitting your revised manuscript for consideration by The EMBO Journal. Please accept our sincere apologies for the unusual delay in the processing of your revised article due to protracted reviewer input. Your amended study was sent back to the three referees for re-evaluation, and we have received comments from all of them, which I enclose below.

As you will see the referee finds that their concerns have been sufficiently addressed and they are now broadly in favour of publication.

Thus, we are pleased to inform you that your manuscript has been accepted in principle for publication in The EMBO Journal, pending some minor issues related to formatting and data representation as listed below, which need to be adjusted at re-submission.

REFeree REPORTS:

Referee #1:

The authors have properly answered the concerns raised on the previous version and the article can be published.

Referee #2:

The authors have nicely addressed the issues raised in my original review.

Referee #3:

This reviewer is satisfied by the response to my many concerns.
This reviewer appreciates the time it has taken to revise the manuscript.

2nd Revision - authors' response

9th Feb 2020

The authors performed the requested editorial changes.

3rd Editorial Decision

10th Feb 2020

Thank you for submitting the revised version of your manuscript. I have now evaluated your amended manuscript and concluded that the remaining minor concerns have been sufficiently addressed.

Thus, I am pleased to inform you that your manuscript has been accepted for publication in the EMBO Journal.

Corresponding Author Name:

Journal Submitted to:

Manuscript Number: